# MECHANISTICALLY ANALYZING THE EFFECTS OF FINE-TUNING ON PROCEDURALLY DEFINED TASKS

**Samyak Jain**[1,*] , **Robert Kirk**[2,*], **Ekdeep Singh Lubana**[3,4,*], **Robert P. Dick**[3],
**Hidenori Tanaka**[4,5], **Edward Grefenstette**[2], **Tim Rocktäschel**[2], **David Krueger**[1]

[1]University of Cambridge, UK
[2]University College London, UK
[3]EECS Department, University of Michigan, Ann Arbor, MI, USA
[4]Center for Brain Science, Harvard University, Cambridge, MA, USA
[5]Physics & Informatics Laboratories, NTT Research, Inc., Sunnyvale, CA, USA

## ABSTRACT

Fine-tuning large pre-trained models has become the *de facto* strategy for developing both task-specific and general-purpose machine learning systems, including developing models that are safe to deploy. Despite its clear importance, there has been minimal work that explains how fine-tuning alters the underlying capabilities learned by a model during pretraining: does fine-tuning yield entirely novel capabilities or does it just modulate existing ones? We address this question empirically in *synthetic, controlled* settings where we can use mechanistic interpretability tools (e.g., network pruning and probing) to understand how the model's underlying capabilities are changing. We perform an extensive analysis of the effects of fine-tuning in these settings, and show that: (i) fine-tuning rarely alters the underlying model capabilities; (ii) a minimal transformation, which we call a 'wrapper', is typically learned on top of the underlying model capabilities, creating the illusion that they have been modified; and (iii) further fine-tuning on a task where such "wrapped capabilities" are relevant leads to sample-efficient revival of the capability, i.e., the model begins reusing these capabilities after only a few gradient steps. *This indicates that practitioners can unintentionally remove a model's safety wrapper merely by fine-tuning it on a, e.g., superficially unrelated, downstream task.* We additionally perform analysis on language models trained on the TinyStories dataset to support our claims in a more realistic setup.

## 1 INTRODUCTION

Large language models (LLMs) pretrained on huge, web-crawled text datasets demonstrate extremely general capabilities (Radford et al., 2018; 2019; Brown et al., 2020; Bubeck et al., 2023). This has led to the current paradigm of machine learning, where practitioners often use model adaptation protocols such as fine-tuning to achieve unprecedented performance on a broad range of downstream tasks (Raffel et al., 2020; Sanh et al., 2022; Reid et al., 2022; Driess et al., 2023; Ahn et al., 2022). Relatedly, the generality of an LLM's capabilities implies the model also learns to exhibit several undesirable behaviors, e.g., producing sensitive, biased, or toxic outputs in the pursuit of completing a task (Weidinger et al., 2021; Lin et al., 2021; Jiang et al., 2021; Parrish et al., 2021; Zhou et al., 2021; Xu et al., 2021; Welbl et al., 2021). Fine-tuning with different training objectives has again seen immense usage in mitigating such "unsafe" capabilities, serving as an integral component of current state-of-the-art *alignment* approaches like RLHF (Ouyang et al., 2022; Go et al., 2023; Stiennon et al., 2020; Bai et al., 2022; Glaese et al., 2022).

Given its ubiquity in the design of both performant and safely deployable models, a natural question emerges: precisely how does fine-tuning influence a pretrained model's capabilities to adapt to a downstream dataset (see Fig. 1)? The generality of an LLM's capabilities opens the possibility

---

*Co-first authors. `samyakjain.cse18@itbhu.ac.in`, `robert.kirk.3.14@gmail.com`, `eslubana@umich.edu`.

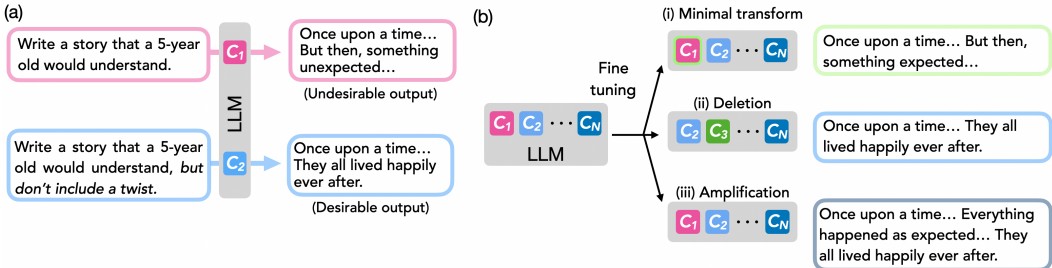

Figure 1: **How does fine-tuning alter a model's capabilities?** (a) Pretraining on huge, web-crawled datasets leads to LLMs learning several capabilities that can justifiably process an input. The figure shows this using an illustrative query, "write a story a 5-year old would understand." Via careful prompting, the desired answer can be retrieved, indicating both desired and undesired capabilities exist in an LLM. (b) Upon fine-tuning, e.g., to avoid use of undesirable capabilities, we hypothesize that three explanations are possible: (i) a minimal transformation of the original capability is learned, e.g., a negation of the original capability; (ii) the undesirable capability is deleted altogether; or (iii) the use of another relevant capability is amplified.

that fine-tuning protocols merely identify the most relevant capabilities and amplify their use for a given set of inputs, while inhibiting the use of other capabilities. Arguably, results on jailbreaking alignment-finetuned LLMs via adversarially generated prompts to elicit undesirable behavior support this hypothesis (Wei et al., 2023; Zou et al., 2023; Shen et al., 2023; Deng et al., 2023; Liu et al., 2023b); however, a precise study to establish the phenomenology of fine-tuning remains absent from the literature. It therefore remains unclear how pernicious this problem is.

Motivated by the above, we perform an extensive analysis of the effects of fine-tuning on a pretrained model's capabilities in *controlled settings* where we can use mechanistic interpretability tools to understand precisely what is happening to the model's underlying capabilities as it is fine-tuned. Specifically, we focus on the following two setups: (i) compiled transformer models based on the *Tracr library* (Lindner et al., 2023; Weiss et al., 2021), which allows encoding specific computational programs into a transformer, and (ii) procedurally generated setups involving *probabilistic context-free grammars (PCFGs)* (Sipser, 1996; Chomsky, 1956), a formal model designed to capture syntactic properties of natural and programmatic languages that has recently served as a testbed for mechanistically understanding language models (Allen-Zhu & Li, 2023c; Delétang et al., 2022; Shi et al., 2022). While Tracr allows us to analyze models with perfectly encoded capabilities, models trained on PCFGs allow evaluation of the effects of design choices involved in the pretraining pipeline. Fine-tuning these models via the often-used protocol of further training a pretrained model on a downstream dataset with a sufficiently small learning rate, we make the following findings.

- **Fine-tuning alters pretraining capabilities by minimally transforming them.** We find that when a relevant pretraining capability is present, the fine-tuned model learns a minimally transformed version of it. We call the transformed portion a *wrapper*.
- **Wrappers are generally very localized.** We show that the wrappers transforming a model's pretraining capabilities are often extremely localized: e.g., via mere pruning of a few weights or neurons, we show the model can start to reuse its pretraining capability and unlearn how to perform the downstream task. Relatedly, we find that via a simple linear probe, we are still able to retrieve outputs expected from the pretrained model.
- **Reverse fine-tuning to "revive" a capability.** In scenarios where upon fine-tuning a model *behaviorally* seems to not possesses a capability, we find that further fine-tuning the model on a subset of pretraining data leads to a sample-efficient "revival" of the capability. We corroborate these results in a realistic setup using the TinyStories dataset (Eldan & Li, 2023).

## 2 RELATED WORK

**Fine-tuning in the "foundation model" era.** Fine-tuning large-scale foundation models pretrained on huge datasets, such as LLMs (Radford et al., 2019; Brown et al., 2020) or large vision models (Radford et al., 2021; Caron et al., 2021), has become the norm in most domains of machine learning. Accordingly, several fine-tuning methods have been proposed in recent years, e.g., instruction fine-tuning (Wei et al., 2021; Liu et al., 2022b; Askell et al., 2021), parameter-efficient fine-tuning (Houlsby et al., 2019; Zaken et al., 2021; Wang et al., 2022), low-rank adaptation (Hu et al., 2021; Pfeiffer

et al., 2020; Lialin et al., 2023), and weight averaging (Gueta et al., 2023; Matena & Raffel, 2022). The diversity of these protocols makes fine-tuning a general, umbrella term for related methods used to adapt a pretrained model to elicit its most relevant capabilities. *For precision, we restrict this paper to fine-tuning protocols that continue training of a pretrained model on a smaller downstream dataset at a learning rate that is often one to three orders of magnitude lower than the average pretraining one.* Such protocols are widely used in practice, e.g., in instruction fine-tuning (Wei et al., 2021).

**Understanding fine-tuning.** A few papers theoretically analyze fine-tuning (Lampinen & Ganguli, 2018; Tripuraneni et al., 2020; Gerace et al., 2022; Maddox et al., 2021; Kumar et al., 2022) under strong assumptions such as relatively simple model classes (e.g., linear functions) or a kernel view of deep learning, which, as shown by Yang & Hu (2020), trivializes the notion of feature transfer in fine-tuning / transfer learning (though see Malladi et al. (2023) for a notable exception). Prior works have also evaluated the effects of fine-tuning via the lens of mode connectivity (Juneja et al., 2022; Lubana et al., 2022), behavioral evaluations (Lovering et al., 2021), and intrinsic dimensionality of the loss landscape (Aghajanyan et al., 2020). In contrast, we aim to provide a mechanistic analysis of how fine-tuning changes model capabilities. Contemporary works by Kotha et al. (2023); Prakash et al. (2024) claim that fine-tuning is unlikely to alter a model's capabilities—their results can be seen as further support for claims made in our work on other experimental setups.

**Model interpretability via synthetic tasks.** Several recent works have focused on *mechanistically* understanding how Transformers learn synthetic language generation tasks, such as learning formal grammars and board games (Allen-Zhu & Li, 2023c; Zhao et al., 2023a; Li et al., 2023; Nanda et al., 2023; Liu et al., 2022a; Valvoda et al., 2022; Liu et al., 2023a; Zhou et al., 2023; Chan et al., 2022). The goal of such papers, including ours, is not necessarily to provide accurate explanations for the success of LLMs, but to develop concrete hypotheses that can be used to develop grounded experiments or tools for understanding their behavior. For example, in a recent work, Allen-Zhu & Li (2023a;b) use a synthetically designed setup to develop hypotheses for how "knowledge" about an entity is stored in a pretrained model, showing such knowledge can often be manipulated via relatively simple linear transformations. Similarly, Okawa et al. (2023) use a procedurally defined multimodal dataset to demonstrate that emergent capabilities seen in neural networks are partially driven by the compositional nature of real world data. In another work, Zhou et al. (2023) utilize Tracr compiled Transformers to hypothesize and demonstrate that if primitive operations involved in a formal algorithm can be implemented by a model, length generalization if practically feasible.

## 3 DEFINING OUR NOTION OF CAPABILITIES

For precision and to motivate our experimental setup, we first discuss the notion of capabilities that we aim to capture for analyzing how fine-tuning alters a model (see Tab. 2 for a summary of notations). We use an idealized definition to communicate our primary intuition and emphasize that we do not expect all capabilities in a pretrained model will act as perfectly as the definition necessitates. However, for the procedural tasks used in this work, our idealized notion is fairly representative.

Let $\mathcal{D}_{\texttt{PT}}$ denote a dataset sampled from a distribution $\mathcal{P}_{\text{X}}$ over the domain X. We will assume the domain X can itself be factorized into two domains $\text{X}_I$ and $\text{X}_D$. Correspondingly, a sample $x \in \text{X}$ can be divided into a tuple of variables $(x_i \in \text{X}_I, x_d \in \text{X}_D)$, where $x_i$ identifies which capability a model should use to process the information encoded by the variable $x_d$. This decomposition captures the idea that different prompts can force a pretrained LLM to elicit different capabilities, as shown in Fig. 1. The identifier of capability $c$ is denoted $i_c$. Pretraining on $\mathcal{D}_{\texttt{PT}}$ yields us an L-layer model $\texttt{M}(.) \colon \text{X} \to \text{Y}$, where often $\text{Y} = \text{X}$ for language models. Let $\texttt{Read}_l(\texttt{M}(.))$ denote the action where a linear layer is trained on intermediate outputs at layer $l$ of model $\texttt{M}(.)$ using $\mathcal{D}_{\texttt{PT}}$. Under this setup, we define a capability as follows.

**Definition 1. (Capability.)** *Define a surjective map $f_{\mathcal{C}} \colon \text{X}_{\mathcal{D}} \to \text{Y}_{\mathcal{C}}$, where $\text{Y}_{\mathcal{C}} \subset \text{Y}$. Let $\text{X}_{\mathcal{C}} \subset \text{X}$ be a sub-domain s.t. $\forall\, x \in X_{\mathcal{C}}$, the capability identifier variable is the same, i.e., $x_i = \texttt{i}_{\mathcal{C}}$. Then, we say the model $\texttt{M}(.)$ "possesses a capability $\mathcal{C}$" if for all $x \in \text{X}_{\mathcal{C}}$, $\exists\, l \leq \text{L}$ s.t. $\texttt{Read}_{\texttt{l}}(M(x)) = f_{\mathcal{C}}(x_d)$.*

A linear readout at an intermediate layer is used in the definition above to emphasize that the notion of a capability need not correspond to only input–output behavior. Further, the definition is restricted to a sub-domain of the overall input space, which we argue is important to define a system's capabilities. For example, one can claim an 8-bit adder circuit possesses the capability to perform addition, but, technically, this is true only over the domain of 8-bit numbers; for inputs with more than 8-bit precision, the circuit will see an overflow error, generating an incorrect but syntactically valid output.

Similarly, an LLM may possesses the capability to identify the sentiment of a passage of text in a specific language, but *possibly* fail when inputs in a different language are shown. Such structured failures imply claiming the existence of a capability should account for the input domain.

We next consider how the fine-tuning distribution $\mathcal{P}_X^{\text{FT}}$ over the domain X can interact with capabilities exhibited in the pretrained model. Our goal here is to capture the fact that a large-scale pretraining corpora is likely to have non-zero probability under the fine-tuning distribution, i.e., it is unlikely that a pretrained model will lack *any* capability relevant to the downstream task. This motivates a notion of "relevance of a capability". Specifically, let $\mathcal{D}_{\text{FT}} \sim \mathcal{P}_X^{\text{FT,E}}$ denote the downstream dataset used for fine-tuning, where $\mathcal{P}_X^{\text{FT,E}}$ is the empirical distribution that captures a subset of the support with non-zero probability in the distribution $\mathcal{P}_X^{\text{FT}}$.

**Definition 2. (Relevance of a Capability.)** *Assume the capability $\mathcal{C}$ in a pretrained model can be transformed to a map $g \circ f_{\mathcal{C}}$ via fine-tuning on $\mathcal{D}_{\text{FT}}$, where $|\mathcal{D}_{\text{FT}}| \ll |\mathcal{D}_{\text{PT}}|$, such that for all $x \sim \mathcal{P}_X^{\text{FT,E}}$, the correct output is produced. Then, if for all $x \sim \mathcal{P}_X^{\text{FT}}$, $g \circ f_{\mathcal{C}}$ yields the correct output, we claim capability $\mathcal{C}$ is **strongly relevant** to the fine-tuning task; else, we call it **weakly relevant**.*

For example, a *weakly relevant* capability can involve the ability to recognize a spurious attribute that the model can learn to exploit to perform well on the fine-tuning dataset, without enabling generalization to the overall distribution that the fine-tuning dataset is sampled from. Meanwhile, a *strongly relevant* capability is one that extracts a causally relevant feature for that task (see Fig. 2 for an example). When a weakly relevant pretraining capability is available, we empirically observe that we can *often* identify specific components in the latter half of the model (e.g., neurons or layers) that seem to implement the transform g in Def. 2. In such

| | Identification of negation words (Weakly relevant) | Sentiment recognition (Strongly relevant) |
|---|---|---|
| Complete the passage while maintaining its narrative. | The movie was definitely not bad. I really can't believe they continue to make these. They're just a waste of resources! | The movie was definitely not bad. More character building could have helped develop the plot, but I loved the twists and action! |

Figure 2: **Capability Relevance.** Consider the task of completing a passage while maintaining its narrative. Herein, the ability to recognize the sentiment of a text will be deemed *strongly relevant* and the ability to recognize negative words *weakly relevant*. Such words are often spuriously correlated with a negative sentiment.

cases, we call g a "wrapper" and $g \circ \mathcal{C}$ a "wrapped capability". If we intervene on the model by either removing the wrapper or training it to forget the wrapper, we find the model starts to perform well on the pretraining task again. In such cases, we say the pretraining capability has been "revived".

## 4 BUILDING CAPABLE MODELS: TRACR AND PCFGS

We next describe the setup used in this work for analyzing how fine-tuning alters a model's capabilities (see Fig. 3 and App. B). Due to lack of clarity on what capabilities a language model possesses or what training data it has seen, we primarily focus on procedurally defined setups that enable clear interpretability. To understand how the relevance of a capability affects fine-tuning, we randomly embed a predefined spurious attribute into the fine-tuning dataset. Specifically, the attribute correlates with the features extracted by the pretraining capability—if the attribute is "simple" enough, the model preferentially exploits it to reduce the downstream loss (Shah et al., 2020; Lubana et al., 2022; Trivedi et al., 2023).

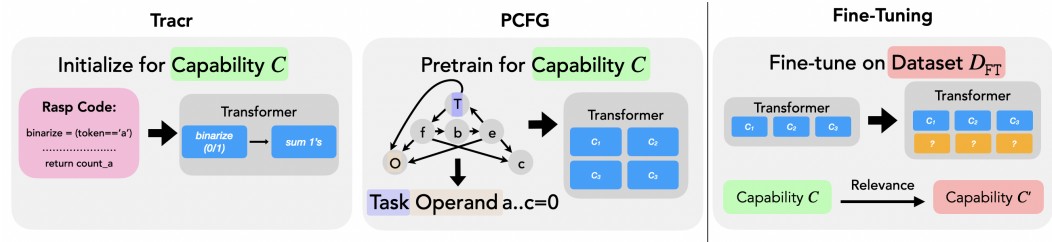

Figure 3: **Experimental setup.** We primarily analyze two setups: (i) Tracr "compiled" models with predefined capabilities and (ii) models trained to learn capabilities defined via a PCFG, following Allen-Zhu & Li (2023c). During fine-tuning, we train the model on a dataset $\mathcal{D}_{\text{FT}}$ that promotes learning of a capability $\mathcal{C}'$. We randomly embed spurious attributes in the fine-tuning dataset that correlate with features extracted by a pretraining capability $\mathcal{C}$ to operationalize capability relevance.

**Compiled capabilities with Tracr.** For a fully controllable system, we use the recently proposed Tracr library (Lindner et al., 2023). Tracr enables "compiling" a transformer model with a set of pre-defined computational primitives over a string of characters from the English alphabet. Accordingly, we define a set of capabilities as a Tracr program and compile it into a Transformer via Tracr (see App. B.1 for a detailed pipeline). The model is then fine-tuned on a downstream task to which the compiled capability may either be weakly or strongly relevant. While we analyze two tasks in this setup, for the main body of the paper, we focus on only the following one.

- **Counter:** Compile the capability to count the number of occurrences of a token $\mathsf{O_{PT}}$ in a string into the model; fine-tune to count occurrences of another token $\mathsf{O_{FT}}$. If $\mathtt{r}(x, \mathsf{O})$ denotes the number of occurrences of a token $\mathsf{O}$ in a string $x$, the spurious correlation is defined by enforcing a constant difference in token occurrences, i.e., $\mathtt{r}(x, \mathsf{O_{FT}}) - \mathtt{r}(x, \mathsf{O_{PT}}) = \mathsf{q}$. See also Alg. 1 and Fig. 12.

As an example, note that in the Counter setup, the model can exploit its pretraining capability and get the correct output on the fine-tuning dataset by merely adding q to the count of $\mathsf{O_{PT}}$ tokens. This *wrapped capability* will however perform poorly on samples without the correlation.

**Learned capabilities with PCFGs.** In this setup, capabilities are "learned", akin to practical situations. This allows us to probe the effects of different pretraining design choices, e.g., the distribution of the pretraining data. Specifically, we follow recent work by Allen-Zhu & Li (2023c) and train a minGPT model (Karpathy, 2020) via autoregressive training on probabilisitc context-free grammars (PCFGs), a formal model of language that captures syntactic properties. Broadly, the data-generating process involves a tree traversal (see Fig. 3), starting from an initial root node and randomly choosing and navigating a set of production rules of the grammar from start/intermediate nodes to intermediate/terminal nodes, stopping only when a terminal node is reached. The terminal nodes reached by all paths starting at the root node will be concatenated to define a string $x$ from the grammar (see Appendix for more details). We prepend special tokens $\mathsf{T}$ and $\mathsf{O}$, called "task family" and "operand" tokens, that specify a certain task must be performed on the string $x$; e.g., count the occurrences (a task family) of a certain token (operand) in a string. Overall, a specific pair of the task family and operand tokens instantiates a task in our setup. The ground truth output of this task and a special token indicating that the output should be produced at the next position are appended at the end of the string in the training data (see App. B.2 for further details and Fig. 15 for an example). Our experiments thus involve the following steps. (i) Pretrain a model on a set of task families. Every sample begins with the task family and operand tokens to specify the task; this ensures different tasks do not "conflict" (assign different labels to the same input), since, by construction, they have non-overlapping support. (ii) Fine-tune the model on a task which may or may not have been included during pretraining. (iii) Evaluate how this fine-tuning affects the model. The data-generating process involves a uniform prior over task family tokens; meanwhile, the set of operand tokens seen during pretraining, denoted $\{\mathsf{O_{PT}}\}$, have a multinomial sampling prior. Specifically, the probability of sampling a specific token $\mathsf{O_{PT}} \in \{\mathsf{O_{PT}}\}$ under task $\mathsf{T}$ is denoted $\mathcal{P}_{\mathsf{T}}(\mathsf{O_{PT}})$. If this probability is low, the model may not learn the relevant capability to perform the task specified by the special tokens. While we analyze the effect of fine-tuning in two broad setups, using a model pretrained on five distinct task families relating to counting and indexing elements of a string, we focus on only the following one in the main body of the paper.

- **Counter:** We intentionally reuse this task to demonstrate the effects of compilation of a capability via Tracr versus learning the capability via PCFGs. Specifically, the model is pretrained to learn to count the occurrences of tokens from a *set* of operand tokens $\{\mathsf{O_{PT}}\}$ and is fine-tuned to exclusively count occurrences of a token $\mathsf{O_{FT}} \in \{\mathsf{O_{PT}}\}$. By making the sampling probability of $\mathsf{O_{FT}}$ tokens high during pretraining, we can make the model preemptively performant on the downstream task. This allows us to model the notion of capability relevance.

## 5 EXPERIMENTS: MECHANISTIC ANALYSIS OF FINE-TUNING

We now provide several results indicating that fine-tuning rarely elicits meaningful changes to pretraining capabilities. To this end, we borrow several protocols commonly used in the field of mechanistic interpretability for our analysis (see Fig. 4), specifically **network pruning** (Voita et al., 2019; Tanaka et al., 2019), **attention map visualizations** (Serrano & Smith, 2019; Wiegreffe & Pinter, 2019; Lai & Tan, 2019), and **probing classifiers** (Tenney et al., 2019; Voita & Titov, 2020; Geva et al., 2023; 2022). We use multiple tools for all experiments since each

tool, individually, is known to suffer from pitfalls (Meister et al., 2021; Bai et al., 2021; Jain & Wallace, 2019; Belinkov, 2022; Bolukbasi et al., 2021). Demonstrating our claims consistently hold true across a diverse set of tools improves our conclusions' robustness to pitfalls of a specific tool. Additionally, we propose a methodology called **reverse fine-tuning** (`reFT`), wherein one takes a pretrained model, fine-tunes it on a downstream dataset, and then fine-tunes it again in the "reverse" direction, i.e., on a dataset sampled from the original pretraining distribution.

We argue if the behavior corresponding to a pretraining capability is retrieved in a few steps of `reFT`, fine-tuning did not meaningfully alter said capability (this claim can be formalized using results by Bengio et al. (2019); Le Priol et al. (2021)).

We primarily focus on the learned capabilities setup of PCFG counter in the main paper, relegating most results on compiled capabilities with Tracr to App. G and other results with PCFGs to App. H—findings remain consistent across all settings. In the PCFG counter setup, the model is pretrained, amongst other tasks, to count occurrences of tokens from the set $\{O_{PT}\} = \{a, b, c\}$ in a given string; during fine-tuning, the model is trained to count occurrences of $O_{FT} = b$. Here, the spurious correlation is defined by enforcing count of b to be 1 more than that of a. The probability a datapoint sampled from the train or test fine-tuning dataset contains a spurious correlation is denoted $C_{Tr}$ and $C_{Te}$, respectively. Here, $C_{Tr} \in \{0.0, 0.5, 0.8, 1.0\}$ and $C_{Te} \in \{0.0, 1.0\}$. We use three sets of sampling probabilities of the task operands in the pretraining data: $\mathcal{P}_T^L = (0.999, 0.001, 0.000)$, $\mathcal{P}_T^M = (0.9, 0.1, 0.0)$, or $\mathcal{P}_T^H = (0.5, 0.3, 0.2)$. These priors indicate a low/medium/high probability of sampling $O_{FT}$. We use the following learning rates (LR) for fine-tuning, $\eta_M = \eta_{PT}/10^1$ and $\eta_S = \eta_{PT}/10^2$, where $\eta_{PT}$ is the *average* pretraining learning rate.

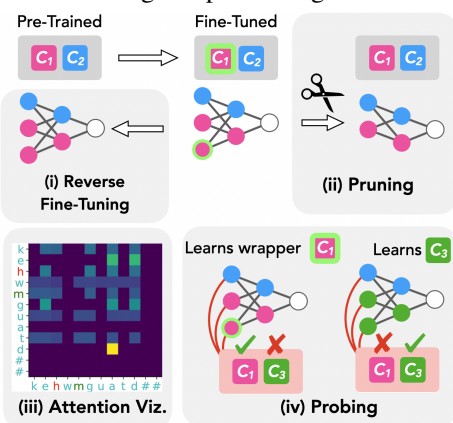

Figure 4: **Analysis protocols.** We analyze how fine-tuning affects a pretrained model's capabilities by (i) reverse Fine-tuning, (ii) network pruning, (iii) attention visualization, and (iv) probing classifiers. We use (ii)—(iv) to show fine-tuning often yields wrapped capabilities. For further evidence, we use (i) and (ii) and find we can "revive" the original capabilities, i.e., the model starts performing well on the pretraining task again. See App. D for precise details.

### 5.1 RESULTS

**Behavioral assessment of fine-tuning.** We first evaluate the model's learning dynamics during fine-tuning (see Fig. 5 and Tab. 5). When the pretraining prior has low probability of sampling the token $O_{FT}$, we see the fine-tuned model performs well only when the spurious correlation is

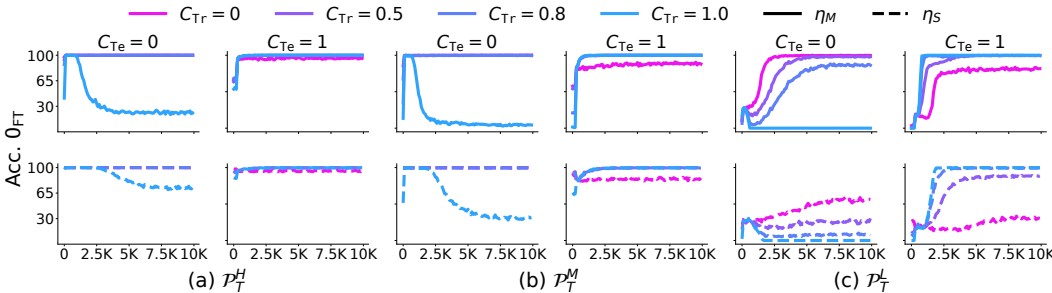

Figure 5: **Fine-tuning accuracy w.r.t. number of training iterations.** We vary the probability of sampling the token $O_{FT}$ in the pretraining data and the spurious correlation in the fine-tuning datasets. When the prior is sufficiently high (a, b), we find the model learns to perform well on the downstream task. Meanwhile, if the prior is low (c), the model learns the downstream task only if a high enough learning rate is used and the spurious correlation is imperfect. This indicates the ability to extract information relevant for the downstream task is likely to be exploited during fine-tuning.

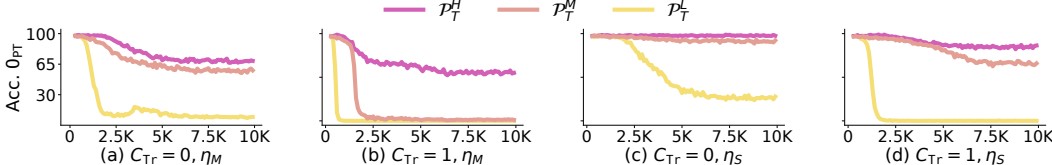

Figure 6: **Impact of sampling prior on the pretraining task's accuracy as fine-tuning is performed.** We plot accuracy on the pretraining task w.r.t. fine-tuning iterations. When the sampling prior of the $O_{FT}$ is low during pre-training, the pretraining task accuracy quickly plummets, especially if the spurious correlation is high; having a high sampling prior mitigates this behavior. This *indicates* pretraining capabilities are affected the most when they are weakly relevant.

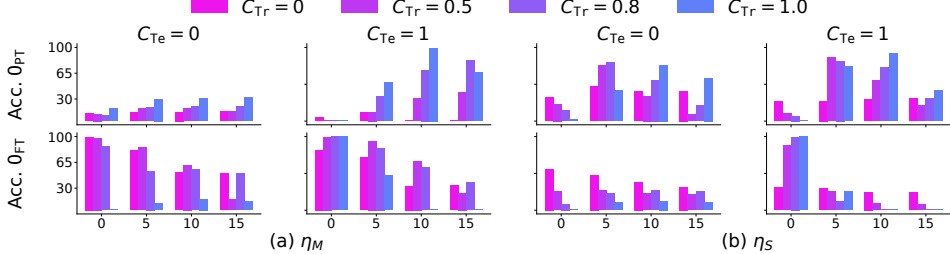

Figure 7: **Pruning a few neurons is sufficient to retrieve pretraining task accuracy.** We plot accuracy w.r.t. number of neurons pruned to improve performance on the pretraining task. We see when a small learning rate is used for fine-tuning, the pretraining task's performance improves after just 5–15 neurons are pruned (top), while the fine-tuning task's performance reduces correspondingly (bottom). We argue these neurons serve as a wrapper to minimally alter the weakly relevant pretraining capability and exploit the spurious correlation present in the fine-tuning data.

present, i.e., $C_{Te} = 1$. As the sampling probability is increased, however, we observe this behavior significantly changes. In particular, even if the model is fine-tuned for a high value of $C_{Tr}$, albeit less than 1, it starts to perform well on the test data regardless of the presence of the spurious attribute. Note that the performance is not high to begin with, indicating the ability to count $O_{FT}$ was learned during fine-tuning; however, having a sufficiently large sampling probability for the token during pretraining leads the model to avoid the spurious correlation. This indicates a pretraining ability to extract information relevant for the downstream task is likely to be exploited during fine-tuning. This is further corroborated by the results in Fig. 6, where we observe that when the spurious correlation is present in the fine-tuning data, accuracy on the pretraining task is affected the most if sampling prior of the target token was low during pretraining. We next analyze these results mechanistically.

**Pruning / Probing fine-tuned models indicates learning of wrapped capabilities.** Our results above *indicate* the model exploits its weakly relevant capabilities, i.e., the capability that helps exploit any spurious correlation, to solve the downstream task. We hypothesize that, at a mechanistic level, the model exploits the weakly relevant capability by learning a *wrapper* over it. To evaluate this, we analyze the models fine-tuned with a low sampling prior via network pruning and linear probing (see App. D for setup details). Specifically, we prune the fine-tuned models to find the most salient weights for reducing loss on the pretraining task of counting $O_{PT}$. If the model learns a wrapper on this capability, the neurons we find should correspond to this wrapper, such that deleting them recovers the capability to count that token. As shown in Fig. 7, we find this is indeed the case—in a setup with weak relevance of capabilities, pruning a very small number of neurons is sufficient to revive the ability to perform well on the original task of counting $O_{PT}$. To assess this further, we train a linear probe on the residual output of every block of the transformer model and determine whether the count of $O_{PT}$ can be accurately computed via the fine-tuned model. As shown in Fig. 8, in the presence of spurious correlations, a linear probe can retrieve the count of the token $O_{PT}$, indicating intermediate outputs relevant to the pretraining capability are still being produced by the fine-tuned model. This observation is particularly evident when a smaller learning rate is used, which is common in practice. Overall, these results show that when a weakly relevant capability is present in the pretrained model, a *wrapper*, i.e., a localized transformation of the pretraining capability, is learned during fine-tuning.

reFT **enables "revival" of pretraining capabilities.** To further corroborate our claims above, we use a model fine-tuned to count $O_{FT}$ and reverse fine-tune it to re-learn the ability to count $O_{PT}$.

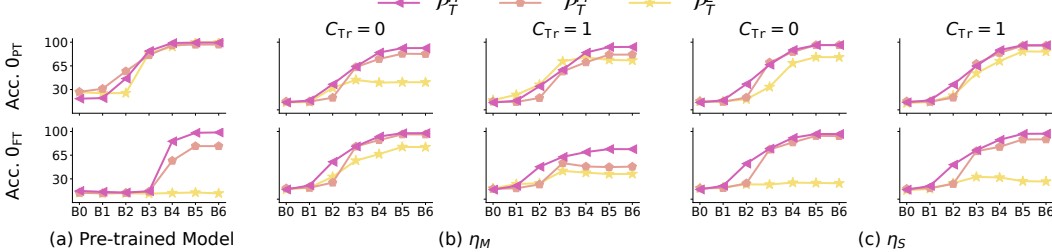

Figure 8: **Probing the presence of pre-training (top) and fine-tuning (bottom) capabilities.** We plot probe accuracy versus the index of the block in the Transformer model. $C_{\text{Te}}$ is set to 0. The pretrained model (left) acts as a baseline for the trend of performance through the model's blocks. In most scenarios, we find we can infer the count of $O_{\text{PT}}$ with a similar trend as the pretrained model (left). A drop in performance is observed only when learning rate $\eta_M$ is used with a weakly relevant capability (low sampling prior). This indicates pretraining capabilities persist after fine-tuning.

As a baseline, we also report a protocol called Scr. + FT, wherein the model is initialized with parameters pre-trained to count $O_{\text{FT}}$ and then fine-tuned to count $O_{\text{PT}}$. Note that this baseline and the reFT protocol differ in their initialization state: former is initialized with parameters pretrained to count $O_{\text{FT}}$, while latter is initialized with parameters pretrained to count $O_{\text{PT}}$ and fine-tuned to count $O_{\text{FT}}$. Results are shown in Fig. 9. We see the model starts to perform well on the pre-training task even if a small learning rate is used for reFT, i.e., even minimally changing the fine-tuned model's parameters is sufficient to regain the pretraining capability! Further, increasing the sampling prior of $O_{\text{FT}}$ accelerates this behavior. This indicates that when a strongly relevant capability is present, the model essentially amplifies its use, but does not catastrophically affect the pretraining capability itself; meanwhile, with a weakly relevant capability (low sampling prior during pretraining), even though the performance is initially poor on the pretraining task, in relatively few iterations (compared to baseline), the accuracy becomes perfect.

**Attention map visualizations further corroborate the wrappers hypothesis.** As we noted before, the results discussed above remain consistent across other experimental setups for both Tracr and PCFG models. However, by construction, Tracr yields particularly interpretable attention maps, allowing us to directly visualize the effects of fine-tuning. We thus analyze the attention maps of a Tracr model on the Counter task described in Sec. 4. Results are shown in Fig. 10. The original Tracr compiled model serves as a baseline and clearly demonstrates that all tokens only attend the pretraining target token, $O_{\text{PT}} = a$. Upon fine-tuning to count $O_{\text{FT}} = b$, we find the model clearly continues to pay attention to $O_{\text{PT}}$ if a small learning rate is

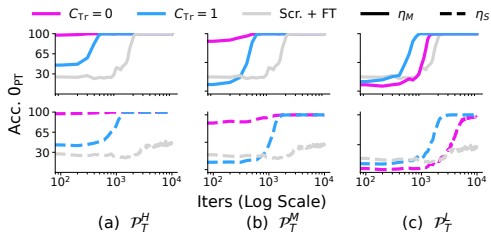

Figure 9: **Reverse Fine-Tuning:** We set $C_{\text{Te}}$ to be 0 to test if the model performs well regardless of a spurious correlation. Models are fine-tuned for 10K iterations. We observe that when a strongly relevant capability is present (a, b), the model very quickly (0.1–1K iterations) starts to perform well on the task via reFT, even if behavior relevant to the capability ceased during pretraining (e.g., when $C_{\text{Tr}}$ is 1). Meanwhile, when the model possesses a weakly relevant capability (c), this "revival" is *slightly* slower (3K iterations). In contrast, the Scr. + FT baseline only reaches perfect accuracy at 4.5K iterations and when using a larger learning rate, i.e., $\eta_M$.

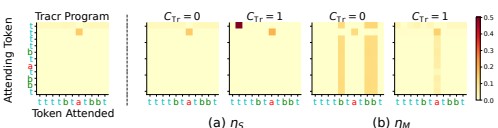

Figure 10: **Visualizing attention maps of fine-tuned Tracr models.** Leftmost panel shows the Tracr compiled model's attention map on the counter task. Upon fine-tuning under different spurious correlations, we see the model continues to pay attention to the pretraining target $O_{\text{PT}} = a$. Only when a large enough learning rate and zero spurious correlation is used, there is a change in the attention pattern.

used. A larger learning rate is, however, able to alter the model computation, but only if the pretraining capability is not weakly relevant to the fine-tuning task, i.e., when $C_{\text{Tr}} = 0$; otherwise, we again find the model continues to pay attention to the pretraining target.

Figure 11: **Validation on TinyStories.** Models are trained to produce stories with several features (e.g., `foreshadowing`) and fine-tuned via different protocols to not produce stories with a "forbidden" feature (specifically, `twists`) (see App. F for details). **Left:** We probe the existence of this feature at intermediate Transformer blocks. Probe accuracy on models pre-trained with or without `twist` data (*Present/Not in Pretraining*, respectively) act as upper and lower bounds on the expected accuracy, and are plotted for ease of comparison. Regardless of the fine-tuning protocol (Filtering, Filtering + Randomisation, Filtering + Mix & Match), for the lower LR, no protocol removes a meaningful amount of information and a similar but less strong trend holds for the higher LR. **Right:** We plot the loss during reverse fine-tuning (`reFT`) to again produce stories with the forbidden feature. Fine-tuned models' loss goes down very quickly (30–300 iterations) compared to baselines (which never reach the same loss; also see Tab. 1). Both these results indicate the capability of identifying the forbidden feature, a necessary capability for story modelling, continues to persist after fine-tuning.

## 5.2 VALIDATING OUR HYPOTHESES ON TINYSTORIES

To give additional strength to our results, we perform an analysis using more realistic language models trained on the TinyStories-Instruct dataset (Eldan & Li, 2023) (see App. B.3 for an example). These models are able to follow specific instructions to write coherent English stories over multiple paragraphs. We perform experiments analogous to the `reFT` and probing experiments in the previous sections, but now explicitly focus on whether fine-tuning can *delete* capabilities present in pre-trained models. Models are pre-trained to generate stories with specific story features (such as containing a `twist`, `foreshadowing`, or `bad ending`) and fine-tuned to *not* generate stories with a specific feature (`twist`) (see App. F for details on the protocols). We probe these models to detect the deleted feature from the intermediate model outputs in Fig. 11, where the dynamics of loss during `reFT` on learning to generate stories with the deleted feature are also shown. We also report the percentage of stories with the deleted feature generated by models during `reFT` in Table 1, where the generated stories are processed by a fine-tuned GPT-3.5 classifier to predict

| Deletion Type | Twist Proportion at Iteration | | | |
|---|---|---|---|---|
| | 0 | 30 | 300 | 3000 |
| F ($\eta_M$) | 44% | 81% | 81% | 82% |
| F + RR ($\eta_M$) | 12% | 56% | 69% | 75% |
| F + MM ($\eta_M$) | 31% | 88% | 50% | 75% |
| F ($\eta_S$) | 69% | 88% | 75% | 94% |
| F + RR ($\eta_S$) | 12% | 44% | 81% | 81% |
| F + MM ($\eta_S$) | 50% | 81% | 62% | 81% |
| Not in PT | 12% | 31% | 44% | 81% |

Table 1: **TinyStories `reFT` Analysis**. We report the percent of generations with a `twist` during reverse fine-tuning for the twist capability. F, R, and MM stand for our three fine-tuning protocols: Filtering, Randomisation and Mix & Match (see App. F for details). Regardless of learning rate and protocol, models relearn to generate stories with `twist` more sample-efficiently than the control model pre-trained on data w/o twists and fine-tuned to generate them (Not in PT).

if the story contains the deleted feature (see App. F for details). Overall, we find that "deleted" capabilities can be easily and sample-efficiently recovered (compared to the baseline), i.e., stories with that feature can be generated again, regardless of the fine-tuning protocol used. These results support our hypotheses that fine-tuning only minimally alters pre-trained model capabilities. We also highlight a few recent papers that propose similar protocols as `reFT` and experiments as ours with further realistic settings (Qi et al., 2023; Yang et al., 2023; Zhang & Wu, 2024; Zhao et al., 2023b).

## 6 CONCLUSION

In this work, we show that fine-tuning generally alters pre-trained model via small, localized changes that only minimally transform their capabilities. We perform our analysis both with existing mechanistic interpretability tools as well as our proposed `reFT` method. Our results pave the way for future work both understanding how fine-tuning works in more realistic settings with larger models, as well as emphasize the need for developing methods beyond fine-tuning that alter pre-trained model capabilities more substantially, particularly deleting unsafe capabilities.

## ACKNOWLEDGEMENTS

ESL thanks Eric Bigelow, Nikhil Vyas, and Usman Anwar for relevant discussions early in the project. SJ was partially supported by BERI. ESL was partially supported via NSF under award CNS-2008151. RK was supported by the Foundation AI CDT at UCL.

## AUTHORS' CONTRIBUTIONS

ESL conceived the project direction and developed a set of hypotheses on the limitations of fine-tuning, with inputs from RK. SJ and ESL co-designed a draft of the PCFG and Tracr setups, and came up with pruning and reverse fine-tuning analysis which led to validation and further refining of the hypotheses. SJ led the experimental execution and made the tasks considered in the paper precise in collaboration with ESL. RK proposed and ran the TinyStories experiments with inputs from ESL, SJ, EG and TR. Literature review and writing of the main paper was led by ESL. SJ led writing of the appendix. ESL, SJ, RK, and HT collaborated on design of all figures and plots. DSK acted as the primary senior advisor on the paper, with inputs from RPD, HT, EG, and TR as well.

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

## A  ORGANIZATION OF APPENDIX

In the appendix we present a comprehensive analysis of our claims on Tracr, PCFG and TinyStories-Instruct using different mechanistic interpretability tools discussed in Section-D of the main paper. We also present a summary of the notations used in this work in Tab. 2. Overall, the appendix is organized as follows:

- Sec. B presents details of the Tracr, PCFG and Tiny Stories datasets utilized in this work.
- Sec. C presents the training and model details for each of the datasets considered.
- Sec. D lists the protocols used for different mechanistic interpretability tools like attention maps, probing, pruning and reverse fine-tuning.
- Sec. E provides a few more results in practically relevant contexts, such as in a synthetic jailbreaking setup.
  - Sec. E.1 studies the effect of using different fractions of pre-training and fine-tuning data points for fine-tuning.
  - Sec. E.2 presents the jailbreaking analysis using the PCFG setup.
  - Sec. E.3 shows reverse fine-tuning a fine-tuned model is sample efficient compared to baselines for both PCFG and Tracr models.
  - Sec. E.4 presents reverse fine-tuning analysis of a fine-tuning protocol that actively tries to remove a capability from PCFG / Tracr models.
- Sec. F presents detailed discussion of setup details and results on TinyStories.
- Sec. G presents additional results on Tracr for counter and max element tasks.
- Sec. H presents additional results on PCFG for the counting and index of occurrence tasks.

Table 2: Notations used in this work.

| Notation | Meaning |
|---|---|
| $X$ | Input domain |
| $X_D$ | Factor of the input domain that captures values of the inputs |
| $X_I$ | Factor of the input domain that captures task identifiers |
| $\mathcal{P}_X$ | Probability distribution over the input domain |
| $\mathcal{P}_X^{\mathrm{FT}}$ | The overall distribution defining the downstream fine-tuning task |
| $\mathcal{D}_{\mathrm{FT}}$ | Dataset used for fine-tuning |
| $\mathcal{P}_X^{\mathrm{FT,E}}$ | Empirical distribution from which the fine-tuning dataset is sampled |
| T | Denotes a task to be performed by the model (e.g., count) |
| O | Denotes an operand that will be processed by to perform the task T |
| $\{O_{\mathrm{PT}}\}$ | Set of operand tokens seen during pretraining |
| $O_{\mathrm{PT}}$ | A specific token used as an operand during pretraining |
| $O_{\mathrm{FT}}$ | A specific token used as an operand during fine-tuning |
| $\mathbf{r}(x, O)$ | Denotes the result of executing a task from Sec. 4 on a string $x$ for some operand O |
| $C_{\mathrm{Tr}}$ | Probability that a randomly sampled string in the training data used for fine-tuning has a spurious correlation between the pretraining capability and the downstream task |
| $C_{\mathrm{Te}}$ | Probability that a randomly sampled string in the test data used for evaluating fine-tuned models has a spurious correlation between the pretraining capability and the downstream task |
| $\mathcal{P}_{\mathrm{T}}(O)$ | Sampling prior. Denotes the probability that when a string with task token T is sampled during pretraining, the operand to perform the task on is O |
| $\mathcal{P}_C^H, \mathcal{P}_C^M, \mathcal{P}_C^S$ | Sampling priors such that the probability of sampling the target token for fine-tuning ($O_{\mathrm{FT}}$) is high ($\mathcal{P}_C^H$), medium ($\mathcal{P}_C^M$), or small ($\mathcal{P}_C^S$) |
| $\eta_M, \eta_S, \eta_{VS}$ | Medium / Small / Very-small learning rates used for fine-tuning. $\eta_{VS}$ is only used for a specific reverse fine-tuning experiment with Tracr compiled models. |
| reFT | Denotes reverse fine-tuning |
| $n_{\mathrm{iters}}$ | Number of iterations used during pre-training |
| LR | Learning rate |

# B ADDITIONAL DETAILS ON DATASETS

We consider three experimental setups: Compiled programs with Tracr (Lindner et al., 2023), learning models on Probabilistic Context Free Grammars (PCFG) (Allen-Zhu & Li, 2023c), and the TinyStories Instruct dataset.

## B.1 TRACR DETAILS

Tracr (Lindner et al., 2023) generates a transformer model using the RASP library by Weiss et al. (2021). The specific code snippet used to generate the Tracr models for the counting and the max element tasks are shown in Fig. 1 and Fig. 2 respectively. The models corresponding to these tasks is implemented with three standard transformer blocks, where each block consists of a self-attention layer followed by two MLP layers.

We analyze the following two tasks to understand the effects of fine-tuning on a pretrained model's capabilities.

- **Counter:** Compile the capability to count the number of occurrences of a token $O_{PT}$ in a string into the model; fine-tune to count occurrences of another token $O_{FT}$. If $r(x, O)$ denotes the number of occurrences of a token $O$ in a string $x$, the spurious correlation is defined by enforcing a constant difference in token occurrences, i.e., $r(x, O_{FT}) - r(x, O_{PT}) = q$. See also Alg. 1 and Fig. 12.
- **Max-identifier:** Compile the capability to identify the $O_{PT}$-th largest element in a string; fine-tune to identify the $O_{FT}$-th largest element. If $r(x, O)$ reads out the $O$-th largest token in the string $x$, we define the spurious correlation as $r(x, O_{FT}) - r(x, O_{PT}) = q$; e.g., if $q = 1$ and the $O_{PT}$ largest token in the string $x$ is a, then the $O_{FT}$-th largest token will be b (which is equal to a + 1 in Tracr's vocabulary). See also Alg. 2 and Fig. 13.

The fine-tuning data is generated by randomly sampling tokens from a uniform distribution over the input vocabulary. For the Counter task, the input vocabulary consists of first nine letters from the English alphabet. For the max element task, the input vocabulary consists of all the letters in the English alphabet. We sample with replacement for the Counter task and without replacement for the max element task (to avoid having multiple max elements). Examples for the task are shown in Figs. 12, 13.

**Algorithm 1: Pseudocode for compiling the Counter capability via Tracr:** Rasp code used to generate the model for the Counter capability and task via Tracr

```
def countA():
    # binzarize the tokens into 0's and 1's
    bin = (rasp.tokens=='a')
    # Select the indices of tokens with value of 1
    bin_idx = rasp.Select(bin, rasp.indices, rasp.Comparison.EQ)
    # Count the number of selected indices
    count_a = rasp.SelectorWidth(bin_idx)
    # Generate an identity map
    idx_select = rasp.Select(rasp.indices, rasp.indices,
     rasp.Comparison.EQ)
    # Output the count
    sum = rasp.Aggregate(idx_select, count_a)
```

> **Task:** Count b
> **Sample:** \$, c, a, d, a, b, c, b, a, d, f, b, g, c, e, b, b, a, h, j, i, b, d, e, f, ,i, h, f, e, g, a, b, g, f, h, j, c, b, e, d, d, h, j, i, b, a, b, #,
> **Answer:** 10

Figure 12: **Exemplar for Counter Task:** A sample used for fine-tuning Tracr compiled models on counting 'b'.

---

**Algorithm 2: Pseudocode for compiling the Max identifier capability via Tracr:** Rasp code used to generate the model for the Max Identifier capability and task via Tracr.

```
def maxᵢdentifier():
    # Identify the tokens larger than a given token
    var_small = rasp.Select(rasp.tokens, rasp.tokens,
     rasp.Comparison.LT)
    # Calculate the sum of the identified tokens for every input token
    sum_small = rasp.SelectorWidth(var_small)
    # Identify the fifth largest token
    bin_target = (sum_small==4)
    # Find the index of the identified token in the original input
    select_idx = rasp.Select(bin_target, rasp.indices,
     rasp.Comparison.EQ)
    # Output the identified index
    return rasp.Aggregate(select_idx, rasp.tokens)
```

> **Task:** Find fifth largest element
> **Sample:** $, b, d, a, f, h, m, x, p, q, n, #, #, #, #
> **Answer:** 'h'

Figure 13: **Exemplar for Max-Element Task:** A sample used for fine-tuning Tracr compiled models on the Max identifier task.

## B.2 PCFG

We follow Allen-Zhu & Li (2023c) and use the production rules shown in Fig. 14. We sample a string of tokens from the grammar and then randomly subsample a string of 250 tokens from the generated original sequence (this helps remove bias towards tokens likely to be at the beginning). The sampling probabilities to sample a valid rule given the parent node is fixed to be 0.5. We formulate the training data as follows: Start of sequence token (SOS) + Task family token (e.g., Counting) (T) + Operand one (counting what) (O) + Operand two (number of positions) ($O'$) + Start of text token ($SOT$) + Data generated from DGP (Txt) + End of text token (EOT) + Answer request token (ART) + Answer token (Ans) + End of sequence token (EOS). This can be summarized as follows.

$$\textbf{Sample input: } \text{SOS} + \text{T} + \text{O} + \text{O}' + \text{SOT} + \text{Txt} + \text{EOT} + \text{ART} + \text{Ans} + \text{EOS}. \qquad (1)$$

We consider the following tasks for pre-training:

- **Counting (C):** Counting number of O (say a) for the last $O'$ positions (forty). This example will be written as Ca40.

- **Counting composition of elements (CC):** Counting number of O (say aa) for the last $O'$ positions (forty). This example will be written as CCa40.

> $s \rightarrow r, q$;  $s \rightarrow q, p$;  $p \rightarrow m, n, o$;  $p \rightarrow n, o, m$;  $q \rightarrow n, m, o$;
> $q \rightarrow m, n$;  $r \rightarrow o, m$;  $r \rightarrow m, o, n$;  $m \rightarrow l, j$;  $m \rightarrow j, l, k$;
> $n \rightarrow k, j, l$;  $n \rightarrow l, j, k$;  $o \rightarrow l, k, j$;  $o \rightarrow k, j$;  $j \rightarrow h, i$;
> $j \rightarrow i, h$;  $k \rightarrow h, g, i$;  $k \rightarrow g, h, i$;  $l \rightarrow i, h, g$;  $l \rightarrow h, i, g$;
> $g \rightarrow d, f, e$;  $g \rightarrow f, e, d$;  $h \rightarrow e, d, f$;  $h \rightarrow d, e, f$;  $i \rightarrow e, f, d$;  $i \rightarrow f, d, e$;
> $d \rightarrow c, a$;  $d \rightarrow a, b, c$;  $e \rightarrow c, b$;  $e \rightarrow c, a, b$;  $f \rightarrow c, b, a$;  $f \rightarrow b, a$;

Figure 14: **PCFG setup:** Grammar rules considered to generate the PCFG dataset. The highlighted token represents the parent token. These rules have been adapted from Allen-Zhu & Li (2023c).



**Task Family Token T::** '('
**Operand Token O::** 'a'
**Sample:** \$, (, a, 40, <, c, a, b, a, c, a, b, a, a, a, c, b, c, b, b, b, a, b, c, a, c, b, c, a, a, c, a, c, a, a, c, c, a, b, a, c, b, b, a, a, a, c, b, c, b, b, c, a, a, c, b, c, b, c, b, a, c, b, c, b, a, a, c, c, b, b, a, c, c, b, a, a, a, b, a, c, b, b, a, a, a, c, b, c, b, b, c, a, a, c, b, c, b, c, b, a, c, b, c, b, a, c, c, b, b, a, c, c, b, a, a, a, b, a, c, b, b, a, a, a, c, b, c, b, b, c, a, a, c, b, c, b, c, b, a, a, a, b, b, a, b, b, a, b, a, b, b, c, b, a, c, c, c, b, a, c, a, c, b, a, c, c, b, c, b, b, a, a, a, c, a, c, b, c, b, a, c, b, c, b, a, c, c, b, b, a, a, a, c, a, c, b, c, b, a, c, b, c, b, a, c, b, b, a, a, c, b, b, a, a, a, c, a, b, c, a, c, b, c, a, b, c, b, a, a, b, a, b, c, a, c, a, c, b, b, c, b, b, a, a, c, b, c, b, a, b, <, =, 15, 10, #, #, #, #, #, #, #, #, #, #, #, #, #, #, #, #, #, #, #, \$
**Answer:** 15



Figure 15: **PCFG Exemplar.** A representative sample from the PCFG dataset (Allen-Zhu & Li, 2023c)

- **Index of occurrence (I):** Index from the EOT token when O (say a) occurred for the $O'^{th}$ time (sixth). This example will be written as Ia10.

- **Index of occurrence of composition element (IC):** Index from the EOT token when O (say aa) occurred for the $O'^{th}$ time (sixth). This example will be written as ICa10.

- **Token value at an index (T):** The token value at index $O'$ (forty) before the end token. O is NULL here. This example will be written as TNULL5.

For the "Counting", "Counting composition of elements", and "Token value at index" tasks, we set the value of $O'$ token as 40. For "Index of occurrence" and "Index of occurrence of composition element" task, we set the value of $O'$ token as 6. All five tasks above are considered during pre-training, but for fine-tuning we consider only a single task with a given operand. Specifically, we analyze fine-tuning the pre-trained models on the "Counting" and "Index of occurrence" tasks only.

We analyze the following two tasks to understand the effects of fine-tuning on a pretrained model's capabilities.

- **Counter:** We intentionally reuse this task to demonstrate the effects of compilation of the capability via Tracr versus learning the capability via PCFGs. Instead of being compiled, the model is trained to count the number of tokens from a *set* of tokens $\{O_{PT}\}$. The model is then fine-tuned to exclusively count a $O_{FT} \in \{O_{PT}\}$ token. By making the sampling probability of $O_{FT}$ tokens high during pretraining, we can make the model preemptively performant on the downstream task; this allows us to model the notion of capability relevance.
- **Indexer:** Amongst other tasks, the model is pretrained to output the index (location in a string) of a token from the *set* $\{O_{PT}\}$ occurs for the $k^{th}$ time; fine-tuning is performed to output the index of $k^{th}$ occurrence of another token $O_{FT}$ instead. We arbitrarily set $k$ to 6 for our experiments, but emphasize that any integer less than context size can be used. If $r(x, O)$ denotes the index of $k^{th}$ occurrence of a token $O$ in a string $x$, the spurious correlation is enforced via constant offset $q$ in operand token indices, i.e., $r(x, O_{FT}) - r(x, O_{PT}) = q$.

While the pre-training dataset is generated by simply sampling from PCFG (see Fig. 15 for an example), for generating the fine-tuning dataset we provide explicit control over the value of $C_{Tr}$ by artificially adding the target tokens $O_{FT}$ from the fine-tuning task. It is important to ensure that the distribution shift between the fine-tuning distributions with different values of $C_{Tr}$ is minimized and the data is fairly spread across multiple classes to enable reusability of feature via pretraining. As shown in Fig. 16, the class distribution of the datasets with $C_{Tr} = 1$ and $C_{Tr} = 0$ for the counting and the index of occurrence tasks satisfies these requirements.

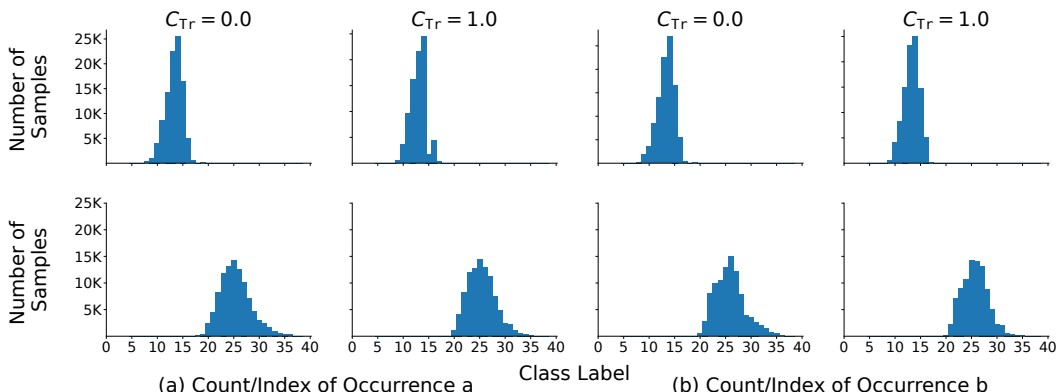

Figure 16: **Distribution of the class labels for Counting (first row) and Index of occurrence tasks (second row).** (a) shows the distribution for the operand token a and (b) shows the same for the operand token b. The data is similarly distributed across different classes and the distribution shift for the two operands and the different values of $C_{\text{Tr}}$ is small.

**Summary:** Eddie, an elderly engine who likes to sleep, is blown off the train tracks by a big wind, but a little bird and its friends help him get back on the tracks and become his best friends. Eddie learns that friends can make you feel young again.

**Features:** Dialogue, Twist.

**Sentence:** Eddie was happy again.

**Words:** sleep, engine, elderly.

**Story:** Once upon a time, there was an elderly engine named Eddie. Eddie was old and tired. He liked to sleep a lot. One day, Eddie was sleeping on the train tracks when he heard a little bird. "Hello, Eddie!" said the bird. "Why are you sleeping here?" "I am old and tired," Eddie told the bird. "I like to sleep." Just then, something unexpected happened. A big wind came and blew Eddie off the tracks. Eddie was scared but the little bird was brave. "Don't worry, Eddie!" said the bird. "I will help you." The bird flew fast and found strong friends to help Eddie. They pushed Eddie back on the tracks. Eddie was happy again. From that day on, Eddie and the bird were best friends. And Eddie learned that even when you are old and tired, friends can make you feel young again.

Figure 17: **TinyStories Exemplar.** An example from the TinyStories Instruct dataset (Eldan & Li, 2023).

### B.3 TinyStories Instruct

For the TinyStories results, we use the TinyStories Instruct variant of the dataset (Eldan & Li, 2023)[1]. An example from this dataset is given in Fig. 17. For the fine-tuning datasets, we take the original dataset and alter it in several ways. Details are discussed in App. F.

## C Details on Training and Evaluation

### C.1 Tracr

**Compiled Model Details:** The compiled model obtained for the counting and max identifier tasks consists of three blocks, wherein each block contains a single head attention layer followed by two layer MLP. No normalization layers are used by models developed using Tracr.

**Training details:** The compiled model is fine-tuned using SGD with momentum for 10K iterations with a batch size of 96. Tracr yields models with a rather sparse parameterization, which often yields unstable training dynamics (e.g., gradient spikes), especially with adaptive optimizers. To address this, we perform the following two interventions. First, we add a small amount of initial gaussian noise $w_{noise} \in \mathcal{N}(0, 0.001)$ to the weights of the compiled model to densify them slightly. Note that the scale of this noise is not high, i.e., it avoids any performance loss but is sufficient enough to reduce gradient spikes resulting from extreme sparsity of model parameters. Second, we choose to use on SGD with momentum as the optimizer, using the following four choices of learning rates: Large LR ($10^{-1}$), Medium LR ($10^{-2}$), Small LR ($10^{-3}$), and Very Small LR ($10^{-4}$). The characterization of "Large" or "Small" is based on a general heuristic of what learning rate regimes are commonly used with SGD in modern neural network training. Linear warmup is used for 2K iterations followed by a cosine schedule with a minimum learning rate of the order $10^{-2}$ smaller than its max value. Evaluation of the fine-tuned model is done on both test set with and without the spurious correlation (ie. $C_{\text{Te}} = 0$ and $C_{\text{Te}} = 1$).

### C.2 PCFG

**Model details:** We use the minGPT model by Karpathy (2020) for all experiments on the synthetically generated PCFG dataset, similar to Allen-Zhu & Li (2023c). The model has close to 3 million parameters and consists of 6 blocks each made up of multihead self attention with 6 heads and two layers of MLP layers with an embedding dimension of 192.

**Pre-training details:** Pretraining is performed from scratch with a learning rate of $10^{-3}$ using the standard AdamW optimizer. Cosine learning rate is used along with linear warmup, where the warmup is used in the first 20% of the training. The model is trained using the standard next token prediction task used for training language models. We consider the set of five tasks mentioned in the previous section during the pre-training phase, but focus on only one of these tasks during fine-tuning. We use the task family token and an operand token to define the notion of a task. The task family token is sampled from a uniform distribution, while the operand token (O) is sampled from a multinomial distribution. The sampling probability for different operands is varied in the experimental setup to understand the effect of capability relevance in fine-tuning. More specifically, we analyze the following distributions for sampling the operand tokens (a, b, c):

- $\mathcal{P}_{\text{T}}(a) = 0.999$, $\mathcal{P}_{\text{T}}(b) = 0.001$, $\mathcal{P}_{\text{T}}(c) = 0.0$;
- $\mathcal{P}_{\text{T}}(a) = 0.99$, $\mathcal{P}_{\text{T}}(b) = 0.01$, $\mathcal{P}_{\text{T}}(c) = 0.0$;
- $\mathcal{P}_{\text{T}}(a) = 0.9$, $\mathcal{P}_{\text{T}}(b) = 0.1$, $\mathcal{P}_{\text{T}}(c) = 0.0$;
- $\mathcal{P}_{\text{T}}(a) = 0.7$, $\mathcal{P}_{\text{T}}(b) = 0.2$, $\mathcal{P}_{\text{T}}(c) = 0.1$; and
- $\mathcal{P}_{\text{T}}(a) = 0.5$, $\mathcal{P}_{\text{T}}(b) = 0.3$, $\mathcal{P}_{\text{T}}(c) = 0.2$.

For each of the configurations of sampling distributions of operands, we pre-train the model for 10K, 50K, 100K and 200K iterations. The model is trained in an online fashion to model the standard language model training pipeline, i.e., data is sampled on the fly from the data generating process during training time.

---

[1] https://huggingface.co/datasets/roneneldan/TinyStoriesInstruct

**Fine-tuning details:** While pre-training is done in the next token prediction fashion, fine-tuning is done in a supervised way where the model is required to just perform the desired fine-tuning task. We use the final iteration model obtained from pre-training as the initialization for fine-tuning. While pre-training is done on multiple pairs of task and operand tokens, the model is fine-tuned on a single pair of task and operand tokens. To simulate a similar setup for fine-tuning as in Tracr, we analyze the effect of fine-tuning the model using three different sets of learning rate: Large LR ($\eta_L$: $10^{-4}$), Medium LR ($\eta_M$: $10^{-5}$) and Small LR ($\eta_S$ : $10^{-6}$). Fine-tuning is done for 10K iterations using AdamW optimizer with a batch size of 96 samples. Similar to pre-training phase, we use cosine learning rate with an initial warmup of 20% of the fine-tuning iterations. The minimum value of the learning rate is set to be $100\times$ lower than the maximum learning rate. Similar to Tracr evaluation is done on both the test sets with and without the spurious correlation ($C_{\texttt{Te}} = 0$ and $C_{\texttt{Te}} = 1$).

## D   MECHANISTIC INTERPRETABILITY TOOLS SETUP

In this section, we describe the different tools of interpretability considered in our work.

**Attention Maps:** We present the attention maps for different tasks considered in the Tracr setup. Each map shows the tokens which are attending other tokens on the y axis and the token which are being attended to on the x-axis. If a token is attended by many other tokens, then, in a crude sense, this can imply that the presence of the token is impacting the underlying task performed by the model. In the Counter task, if significant attention is given to a's / b's is an indicator of the respective capability of the model. For the max identifier task, in the attention map in Block-0, the model implements the sorting function, where each token is attended by the tokens which are greater than that. The vocabulary order followed is a > b > c > d.... In the attention map of Block-2, the model implements the read function, where it outputs the token at the desired position in the sorted sequence.

**Pruning:** We consider single step pruning where we prune the weights/neurons with largest dot product between their gradient and weights, where the gradients are calculated by minimizing the loss for the capability we want to revive. More formally, let the weights of the model with $N$ parameters be given by $w_i$ where $i \in \{0, 1, \ldots, N-1\}$, Let the corresponding gradient be given by $grad(w_i)$ then the top-K weights with largest value of $grad(w_i)w_i$ are pruned off. This follows the pruning protocols proposed in prior work for reducing or preserving loss via pruning (Molchanov et al., 2016; Lubana & Dick, 2021; Mozer & Smolensky, 1988). We use weight pruning for the Tracr setup and neuron pruning for PCFG, where a neuron is defined as a row in the weight matrix. We present a detailed description of the pruning protocol considered in Algorithm-3.

---

**Algorithm 3:  Pruning Pseudocode**. A fine-tuned model $f_\theta$ is parameterized by $\theta$ and $\theta_i$ denotes its $i^{th}$ neuron or weight (we prune neurons in PCFG experiments and weights in Tracr). Pre-training task family token is given by $\texttt{O}_{\texttt{PT}}$ and is prepended to a string $X$ sampled from the data generating process, yielding the input $\texttt{O}_{\texttt{PT}} \circ X$. The true value corresponding to pre-training task family token $\texttt{O}_{\texttt{PT}}$ is given by $y$. Let the cross-entropy loss be given by CE. Let $\text{Top}_K(W)$ denote the indices of the top $K$ values in the vector $W$.

```
def prune():
    # Forward prop the model on pre-training task
    out = f_θ(O_PT ∘ X)
    # Calculate the loss
    L = CE(out, y)
    # Calculate the gradients
    grad = ∇_θ L
    # Calculate the dot product between model weights and gradients
    dotproduct = θ.grad
    # Select the indices of top K values
    indices = Top_K(dotproduct)
    # Prune off the neurons/weights present in top K indices
    θ[indices] = 0
    return θ
```

---

**Probing:** Probing is used to understand if a particular capability is present in the model. In this, we train a linear layer (probe) on top of every block (residual layer's output) of the mini-gpt model and

analyze if the probe is able to perform on a task requiring the use of the desired capability. The probe is a linear layer with the output dimensions same as the vocabulary size of the model. The probe is trained using the data randomly sampled from the PCFG data generating process for 4K iterations using AdamW optimizer with maximum learning rate of $10^{-3}$ which is decayed by a factor of 10 at 2K, 3K and 3.5K iterations. Training of the probe is done separately on the residual output of each of the six blocks present in the minGPT model. The stream corresponding to the answer token (`Ans`) is used as the input to the probe.

**Reverse Fine-tuning:** Same set of hyperparameters as used in the fine-tuning of the pre-trained Tracr model are used in `reFT` , except for the learning rate, which we force to be smaller than the corresponding fine-tuning learning rate. Note that this use of an even smaller learning rate is intentional: if the original pretraining capability can be revived even with this setup, it is stronger evidence that the pretraining capability was never forgotten or removed.

# E ADDITIONAL RESULTS

## E.1 FINE-TUNING IN PRESENCE OF SOME PRE-TRAINING DATA

In this section, we demonstrate our claims also hold for an often used fine-tuning setup wherein, beyond the fine-tuning data, the model also gets to see some portion of the pretraining data again. Specifically, we perform three degrees of mixing of the pretraining and fine-tuning data: (i) 50% PT + 50% FT, (i) 10% PT + 90% FT, and (i) 0.1% PT + 99.9% FT. We show behavior results on how the performance of the model improves as a function of fine-tuning iterations for different spurious correlations for a low pretraining sampling prior in Figs. 18, 19 and high sampling prior in Figs. 20, 21. Furthermore, we probe these models' intermediate outputs to infer if features relevant to the pretraining capability continue to persist. Results can be seen in Figs. 22, 23.

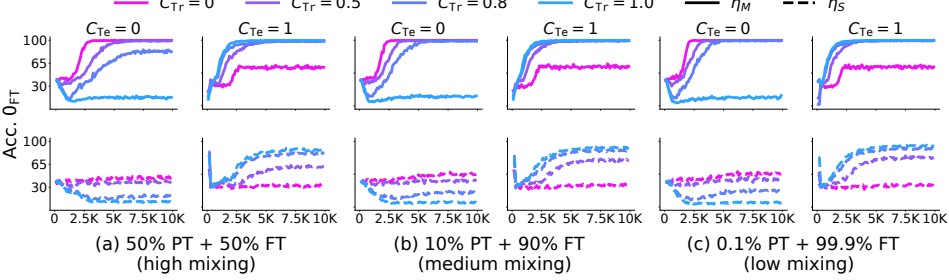

Figure 18: **Effect of different sampling probabilities of pre-training target token $O_{PT}$ on fine-tuning task's performance.** We observe similar gains for different values of sampling probabilities of $O_{PT}$ during fine-tuning.

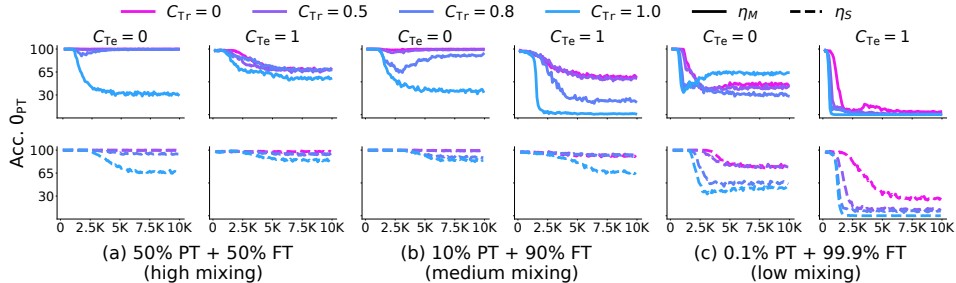

Figure 19: **Effect of different sampling probabilities of pre-training target token $O_{PT}$ on pre-training task's performance.** We observe a higher loss in performance if low sampling probability is used for sampling the pre-training target token $O_{PT}$ during fine-tuning.

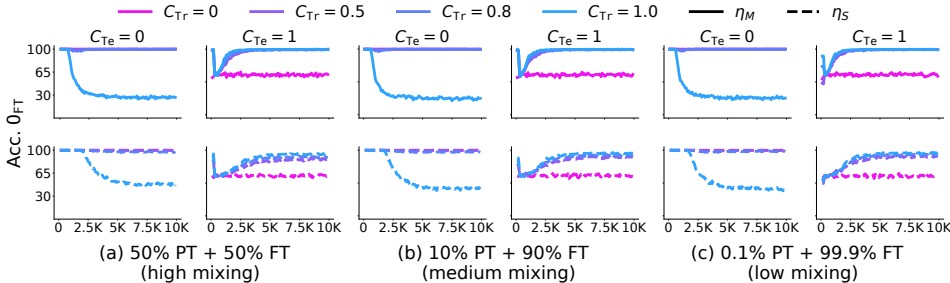

Figure 20: **Effect of different sampling probabilities of pre-training target token $O_{PT}$ on fine-tuning task's performance.** Pre-training is done using high sampling prior for fine-tuning task family token. We observe similar gains for different values of sampling probabilities of $O_{PT}$ during fine-tunning.

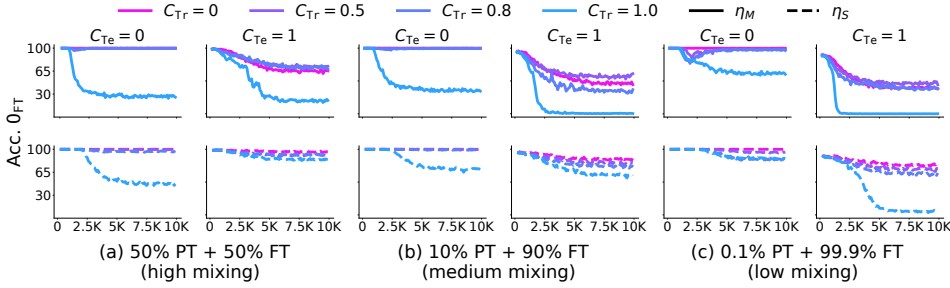

Figure 21: **Effect of different sampling probabilities of pre-training target token $\boxed{\textsf{O}_{\textsf{PT}}}$ on pre-tuning task's performance.** Pre-training is done using high sampling prior for fine-tuning task family token. We observe similar gains for different values of sampling probabilities of $\boxed{\textsf{O}_{\textsf{PT}}}$ during fine-tunning.

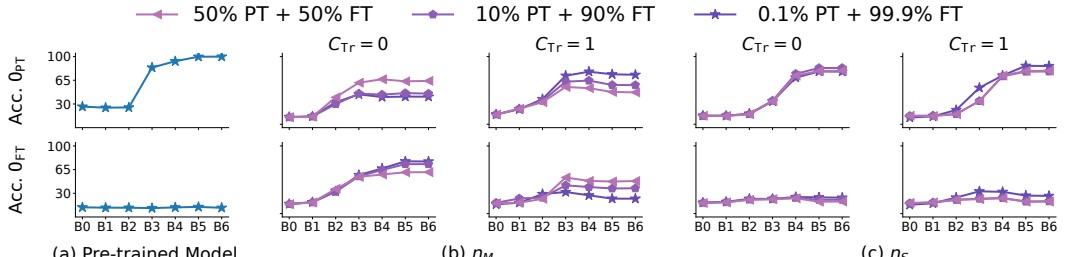

Figure 22: Probing analysis corresponding to Fig-18 and 19

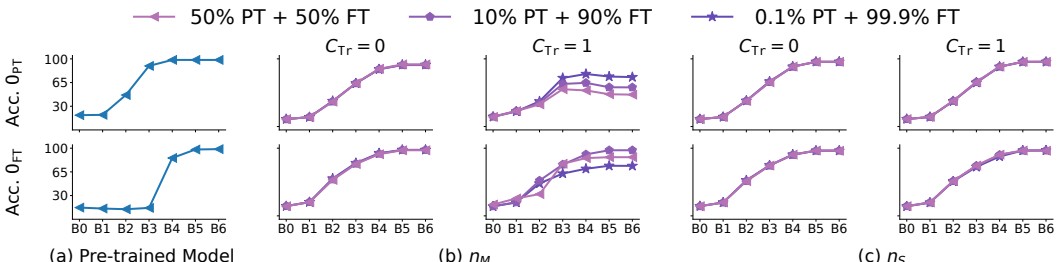

Figure 23: Probing analysis corresponding to Fig-20 and 21

### E.2 JAILBREAKING ANALYSIS

We emulate jailbreaking (Wei et al., 2023; Zou et al., 2023) in our PCFG setup by defining several task family tokens describing the same task. Specifically, for the "Counter" task, we use three task family tokens $T_{NJ}, T_{J_1}, T_{J_2}$ to refer to the task in a string. Here subscript $NJ$ indicates the task family token will not allow jailbreaking, while $J_1/J_2$ indicate the task family token can be used to jailbreak the model, as explained next. For pretraining, the token $T_{NJ}$ may be paired with operand tokens a, b, c to learn to count them from inputs sampled from the PCFG. However, tokens $T_{j_1}, T_{j_2}$ are used only for counting a. During fine-tuning, the model is fine-tuned to count the token b using the task family token $T_{NJ}$. For evaluation, we compute the model's accuracy on its ability to count the token a, using either the task family token $T_{NJ}$ or $T_{J_1}, T_{J_2}$. As shown in Fig. 24, the model is unable to infer the count of a if the task family token $T_{NJ}$ is used; however, if task family tokens $T_{J_1}, T_{J_2}$ are used, the model performs perfectly if the prior for sampling the fine-tuning target b during pretraining was sufficiently high. We argue that this is expected because under a high sampling prior breaks the symmetry between task family tokens (indeed, $T_{J_1}$ is only seen with operand token a, but $T_{NJ}$ is seen for all operand tokens. This indicates the pretraining capability continues to persist in the model, enabling jailbreaking. To further investigate this result, we also probe the fine-tuned models. Results are shown in Fig. 25. As expected, we see task family tokens $T_{J_1}, T_{J_2}$ allow for linear readout of the count of a; however, we see that even for inputs with task family token $T_{NJ}$, the model does encode the count of a in the outputs around the middle layers!

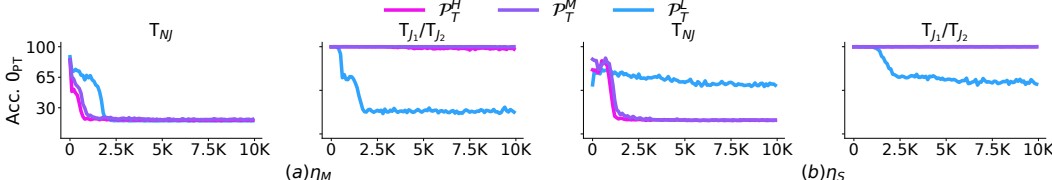

Figure 24: **Jailbreaking analysis using PCFG.** We report performance on the pretraining task (counting $O_{PT}$) as a function of fine-tuning iterations, where the fine-tuning task (counting $O_{FT}$) is performed using the task family token $T_{NJ}$. We find that the model is able to learn the fine-tuning task and seemingly performs poorly on the pretraining task when task family token $T_{NJ}$ is used in the input. However, in presence of a sufficiently relevant capability (high pretraining prior for $O_{FT}$), using task family tokens $T_{J_1}$ or $T_{J_2}$ in the input shows the model can still perform the pretraining task perfectly—i.e., we can jailbreak the model.

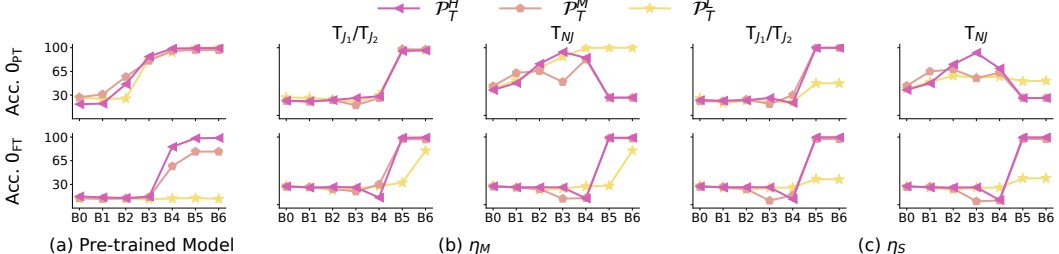

Figure 25: **Probing analysis for the setup used to understand jail-breaking.** Similar results on using the fine-tuning token or the jailbreaking token for training the probe indicate that the pre-training capabilities are not removed on fine-tuning.

### E.3 SAMPLE EFFICIENCY ANALYSIS FOR REVERSE FINE-TUNING

To emphasize the fact that the pretraining capability is "revived" in the model relatively sample-efficiently, we repeat Fig. 9, where models trained on PCFG are reverse fine-tuned, and repeat the experiment with the Scr. + FT baseline for Tracr compiled models. As can be seen in Figs. 26, 27, compared to the baseline, the model learns to perform the pretraining task in substantially fewer iterations than the baseline. We note that for the Tracr models in these results, even an extremely small learning rate is sufficient to revive the pretraining capability! We also note that we do not sweep over the $C_{\text{Tr}}$ hyperparameter in the Tracr models because they are compiled, i.e., we cannot control the correlation with the pretraining capabilities in a meaningful way.

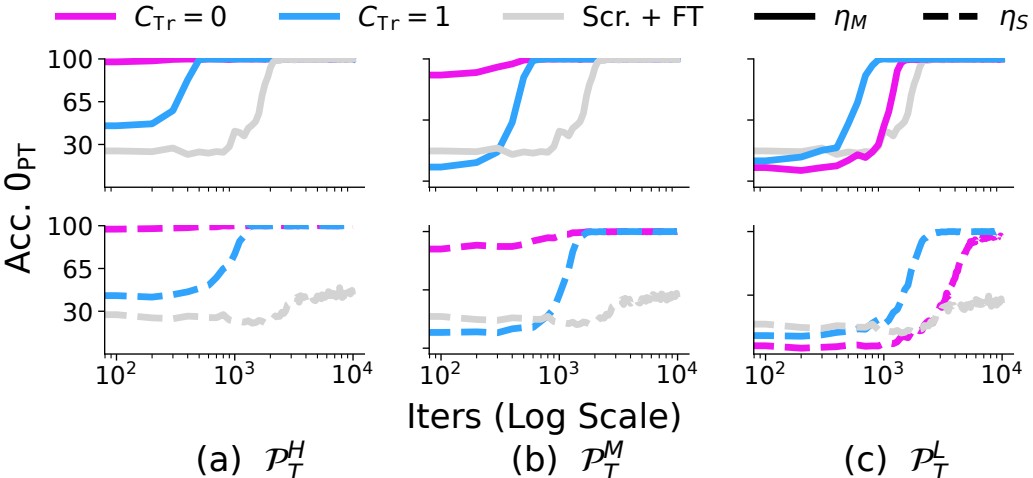

Figure 26: **Reverse Fine-Tuning on PCFGs:** We set $C_{\text{Te}}$ to be 0 to test if the model performs well regardless of a spurious correlation. We observe that when a strongly relevant capability is present (a, b), the model very quickly (0.1–1K iterations) starts to perform well on the task via reFT , even if behavior relevant to the capability ceased during pretraining (e.g., when $C_{\text{Tr}}$ is 1). Meanwhile, when the model possessesses a weakly relevant capability (c), this "revival" is *slightly* slower (3K iterations). In contrast, the Scr. + FT baseline only reaches perfect accuracy at 4.5K iterations and when using a larger learning rate $\eta_M$.

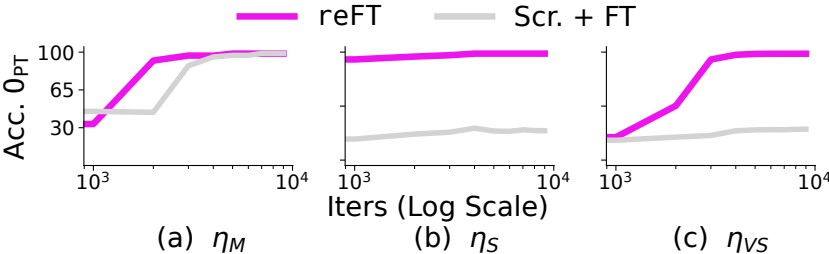

Figure 27: **Reverse Fine-Tuning on Tracr:** We set $C_{\text{Te}}$ to be 0 to test if the model performs well regardless of a spurious correlation. We observe that the fine-tuned model upon reFT very quickly starts starts to perform well on the pretraining task. Moreover, the protocol works even if an extremely small learing rate is used. In contrast, the Scr. + FT baseline only reaches a large learning rate $\eta_M$ is used, and does so less sample efficiently. We note that the results for $\eta_M$ learning rate look worse than the $\eta_S$ learning rate around $10^3$ iterations because $\eta_M$ is too big of a learning rate, forcing the model to essentially go through a "retraining" phase.

### E.4 REVERSE FINE-TUNING A MORE SAFETY-ORIENTED FINE-TUNING PROTOCOL

The fine-tuning protocols used in the bulk of the paper focus on learning a new capability, e.g., counting a new operand, while promoting reusability of capabilities learned during pretraining. Part of our motivation is to see if a pretrained model is actively forced to remove a capability, does that work? To analyze this, we define a fine-tuning protocol called `randFT` wherein the model is trained to actively produce an incorrect output for inputs that require use of the pretraining capability. For example, if the model possessesses the capability to produce the count the number of occurrences of token $O_{PT}$ = a in a string, we fine-tune it to produce the count of tokens $O_{FT}$ = b in that string. We analyze these fine-tuned models analyzed via reverse fine-tuning (`reFT`), i.e., by further training them to produce the correct outputs (number of occurrences of token $O_{PT}$). We provide results for three baselines as well: (i) `Scr.`, wherein the model is trained from scratch to learn to count the token a; (ii) `Scr. + FT`, wherein the model is initialized with parameters trained via trained from scratch to count a separate token ($O_{FT}$) and then the model is fine-tuned to count the token $O_{PT}$; and (iii) `reFT`, which follows reverse fine-tuning models that were fine-tuned with the protocols used in the bulk of the paper, i.e., fine-tuned to learn a new capability that is related to the pretraining one.

Results are shown in Fig. 28. We specifically zoom in on the the scenario where `reFT` takes the longest time, i.e., when the sampling prior of the downstream target $O_{FT}$ is low in pretraining data; results for other sampling priors are shown in Fig. 29 We see that reverse fine-tuning a `randFT` model is similarly sample-efficient as the standard `reFT` pipeline used in the bulk of the paper, while being more sample-efficient than the `Scr.` and `Scr. + FT` baselines.

In addition, we perform a probing analysis of the `randFT` models in Fig. 30. We again find that we can predict the information relevant for the pretraining task, i.e., the count of $O_{PT}$.

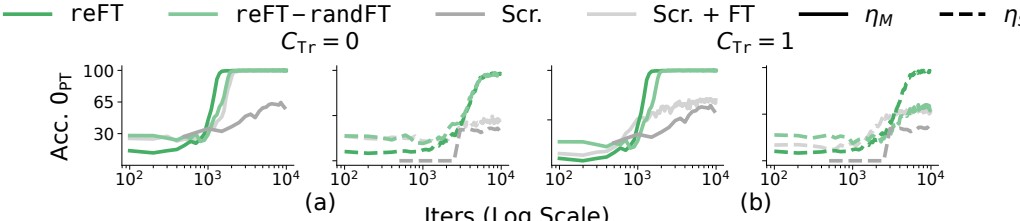

Figure 28: **Reverse fine-tuning a model fine-tuned to remove its pretraining capability.** See text in Sec. E.4 for details.

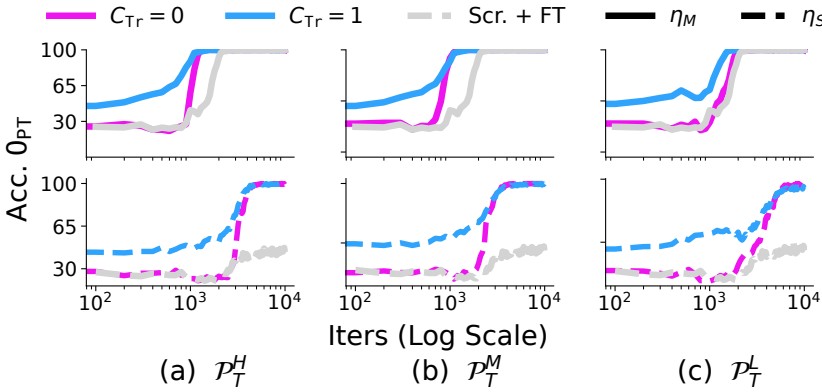

Figure 29: **Reverse fine-tuning performance on using `randFT` fine-tuning protocol to forget the pre-training capability.** We follow the setup of Fig. 9 and plot results for several sampling priors of the target token for fine-tuning, i.e., $O_{FT}$, but we use `randFT` for fine-tuning the models before `reFT`. The baseline results `Scr. + FT` are copied from the Fig. 28, i.e., baseline is not trained in an "adversarial" way, but the `randFT` results are. While this makes the baseline unfairly stronger, we find `reFT` the `randFT` models are still more sample efficient.

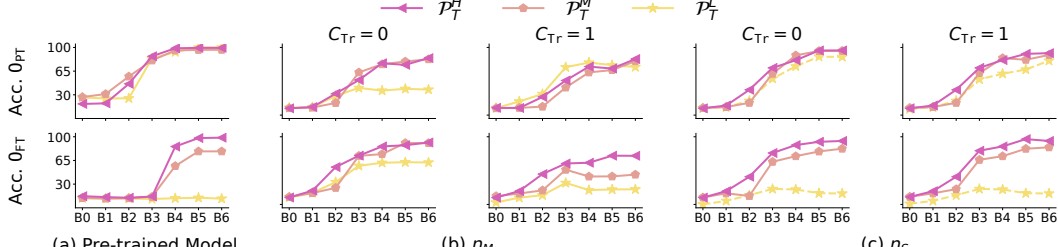

Figure 30: **Probing analysis of** `randFT` **fine-tuning protocol.** We plot probe accuracy versus the index of the block in the Transformer model. $C_{\text{Te}}$ is set to 0. The pretrained model (left) acts as a baseline for the trend of performance through the model's blocks. In most scenarios, we find we can infer the count of $0_{\text{PT}}$ with a similar trend as the pretrained model (left). A drop in performance is observed only when learning rate $\eta_M$ is used with a weakly relevant capability (low sampling prior). This indicates pretraining capabilities continues to persist upon fine-tuning.

# F    DETAILS AND RESULTS ON TINYSTORIES EXPERIMENTS

In this section, we describe our experiments on the TinyStories dataset in more detail. These experiments are designed to validate our hypotheses in a more realistic language modelling setting. Overall, the results support our hypothesis that fine-tuning does not lead to deletion of capabilities as they can be revived in a sample-efficient way and uncovered through probing.

## F.1    MODEL TRAINING

**Dataset.**    We use the TinyStories (Eldan & Li, 2023) dataset to train our models. This data consists of children's stories written by GPT-3.5 and GPT-4. Each story is several paragraphs long, and comes with several attributes labelled: a set of three words that are included in the story; a sentence that is included in the story; a GPT-3.5-written summary of the story; and a list of 0-3 "story features", such as Twist, Dialogue or Bad Ending, which the story abides by.

We use the TinyStories-Instruct version of this dataset [2], wherein each story is prefixed with an "instruction" containing the story attributes described above, hence enabling the model to learn to conditionally generate stories based on an input or instruction.

**Pre-training.**    We pretrain 91 million parameter autoregressive language models with a similar architecture to LLaMa 2 (Touvron et al., 2023), with a custom tokenizer with vocabulary of size 8192 trained on the dataset.[3] They have hidden dimension 768, 12 layers, and 12 attention heads per layer. These models are trained with the standard language modelling cross-entropy loss, with batch size 128, sequence length 1024, no dropout, for 30,000 gradient steps, with a learning rate schedule with a linear warmup from 0 and cosine decay to 0, with maximum learning rate 0.001. These models achieve a loss of $\tilde{0}.8$ at the end of training, and can generate coherent multi-paragraph stories given a specific instruction in the form it saw during training.

**Fine-tuning.**    We are interested in analysing whether fine-tuning these models can alter underlying capabilities. The specific capability we investigate is that of generating stories containing Twists (which is one of the story features), and are analysing whether various fine-tuning protocols can remove this capability from the pre-trained model. We investigate a variety of fine-tuning protocols modelled after plausible realistic scenarios where one may want to fine-tune a model to not generate text of a certain type (e.g., highly toxic text), regardless of the input instruction. These include: **Filtering** fine-tunes the model on a dataset where all instances of stories with Twists are filtered out; **Filtering + Mix & Match** filters, and then replaces all instances of another, unrelated feature (in this case, Foreshadowing) in the instruction with the Twist feature; and **Filtering + Randomisation** filters, and then adds the "Twist" instruction to the prompt for stories that do not contain Twists, thus training the model to not model stories with Twists even if instructed. This last protocol acts as a kind of adversarial training (in that there are stories with the Twist instruction but no Twists), and introduces a spurious correlation between the Twist instruction and the foreshadowing capability, as in the Tracr and PCFG results.

We take the pre-trained model described above, and fine-tune it with these various protocols. We then perform reFT on a dataset of stories which all have Twists in, to measure the extent to which each fine-tuning protocol deleted the capability of Twist generation. To ensure a good control, we compare the reFT models to a model pre-trained on data with no Twist stories, which is then fine-tuned on Twist stories. The sample efficiency and final performance of this model serves as a comparison for the reFT ed models.

## F.2    EVALUATION METRICS

We evaluate whether the fine-tuning protocols have removed the capability to model and generate stories with Twists in multiple ways. Firstly, we look at the loss on stories with Twists. If fine-tuning deletes the Twist capability, we expect the loss on this dataset to increase.

---

[2]https://huggingface.co/datasets/roneneldan/TinyStoriesInstruct
[3]Our code is based on this repository: https://github.com/karpathy/llama2.c

**GPT Evaluations.** To evaluate the generative capabilities of these models, we generate stories from them given prompt instructions with the `Twist` story feature. We then evaluate whether these stories contain `Twists`. To do this evaluation, we use the OpenAI GPT fine-tuning API [4] to fine-tune a GPT-3.5 model to classify whether a given story has a `Twist` or not. To do this, we use the TinyStories dataset and accompanying labels. This fine-tuned model achieves 92% accuracy on a held-out test set after fine-tuning. We generate stories with multiple different prompts from both the fine-tuned and reverse fine-tuned models throughout fine-tuning, and measure the proportion of stories which are classified as having a `Twist`, which we call the *generation score*.

**Probing.** As well as using `reFT` to measure whether the fine-tuning protocols have deleted the capability to generate `Twists`, we also use probing to evaluate whether fine-tuning removes information from internal representations. We train linear probes on the internal activations of the transformer models to predict which story features (e.g. `Twist`, `Bad Ending`, `Dialogue`) are present in the story. These probes take an average of the activations at the final 10 token positions of the story. Given that this is a multi-label classification problem we employ a separate binary classification probe to classify the presence of each story feature. We use the accuracy of these probes at different layers before and after fine-tuning, and on the control pre-trained model which was trained on data with no `Twists`, to measure whether fine-tuning has removed information from the models' internal activations.

## F.3 RESULTS

**Reverse fine-tuning** The loss on stories with `Twist` during fine-tuning is shown in Fig. 31. This shows that the fine-tuning protocols are raising the loss, and hence behaviourally deleting the capability of fine-tuning. The generation scores are shown in Fig. 32. This again reinforces that most fine-tuning protocols are removing the capability behaviourally, as the generation scores (while noisy) drop to close to 0.

Fig. 33 shows the loss during `reFT` for all the fine-tuned models, as well as the control model pre-trained without stories with `Twists`, and Fig. 34 shows the generation scores. Both of these results show that the fine-tuned models learn the new capability in a much more sample-efficient way, and in fact converge to a lower loss on this dataset than the control pre-trained model.

**Probing** In addition to the `reFT` results, we perform probing experiments. The probing accuracy for the Twist feature across layers for the fine-tuned models and the two control pre-trained models is shown in Fig. 11, which we reproduce here in Fig. 35 for completeness. These results show that a small amount of information about story classification has been removed from the activations of the fine-tuned models compared to the model pre-trained with `Twist` stories, but the reduction is very minor, as shown in comparison to the information present in the model pre-trained without `Twist` stories.

Fig. 36, Fig. 37, and Fig. 38 show similar plots for several other story features. Some of these are easier or harder for probes to classify, but *the important result is that the difference in probe accuracy between the fine-tuned models and both pre-trained control models is negligible for all of these features, showing that the results in Fig. 35 are due to the `Twist` feature, i.e., the feature that we trained the model to delete.*

---

[4]https://platform.openai.com/docs/guides/fine-tuning

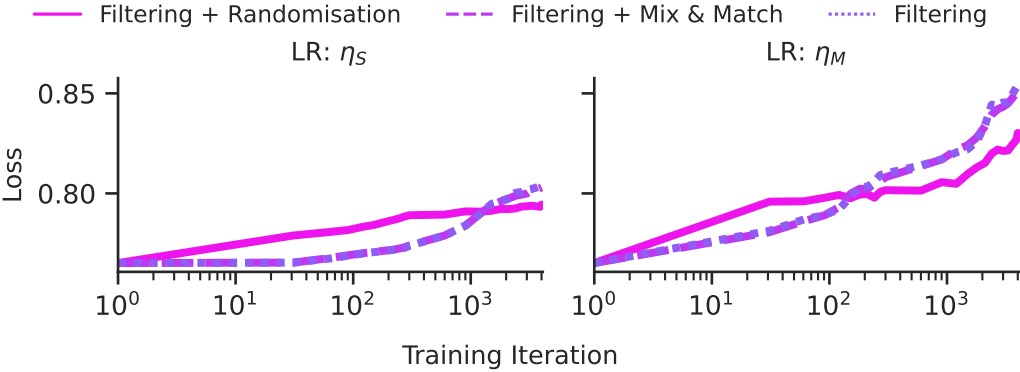

Figure 31: **Larger learning rates lead to more pronounced loss of modelling capability.** The plots show loss on data with the Twist feature present while fine-tuning to delete the capability to model text with the *Twist* feature, for different learning rates and fine-tuning protocols.

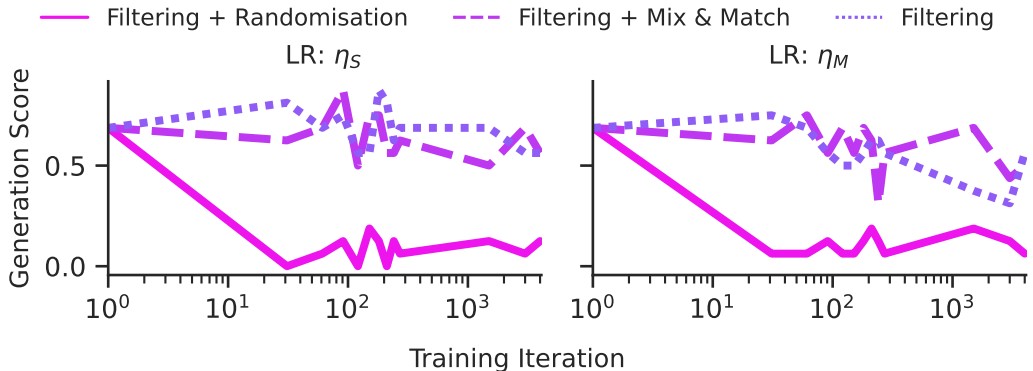

Figure 32: **Larger learning rates lead to more pronounced loss of generative capability.** The plots show the generation score for the Twist feature present while fine-tuning to delete the capability to model text with the *Twist* feature, for different learning rates and fine-tuning protocols.

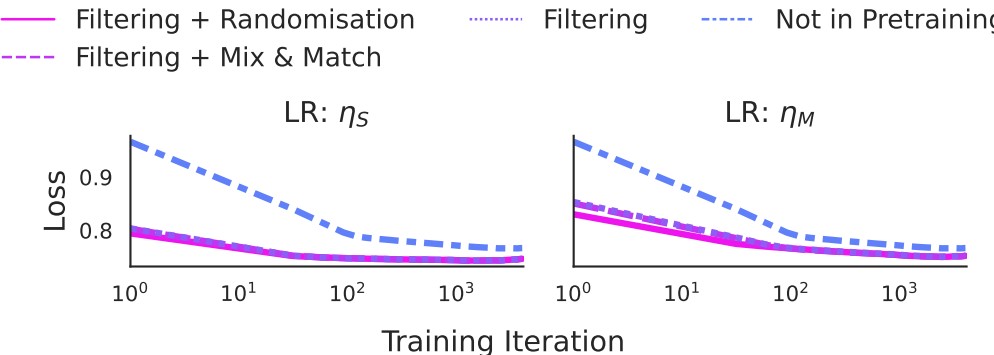

Figure 33: `reFT` **easily recovers deleted capabilities**. We plot loss on data with the `Twist` for `reFT` of various models fine-tuned to delete the capability, as well as a control model which was pre-trained without data with `Twists`. The fine-tuned models learn the capability more sample-efficiently, and additionally converge to a lower loss than the control model.

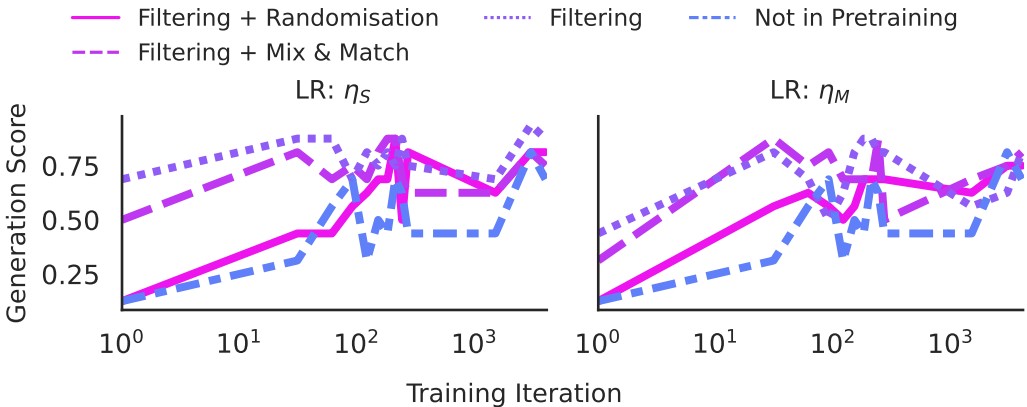

Figure 34: `reFT` **easily recovers deleted generative capabilities**. We plot the generation scores for the `Twist` feature for `reFT` of various models fine-tuned to delete the capability, as well as a control model which was pre-trained without data with `Twists`. The fine-tuned models learn the capability much more sample-efficiently, and additinoally converge to a lower loss, than the control model.

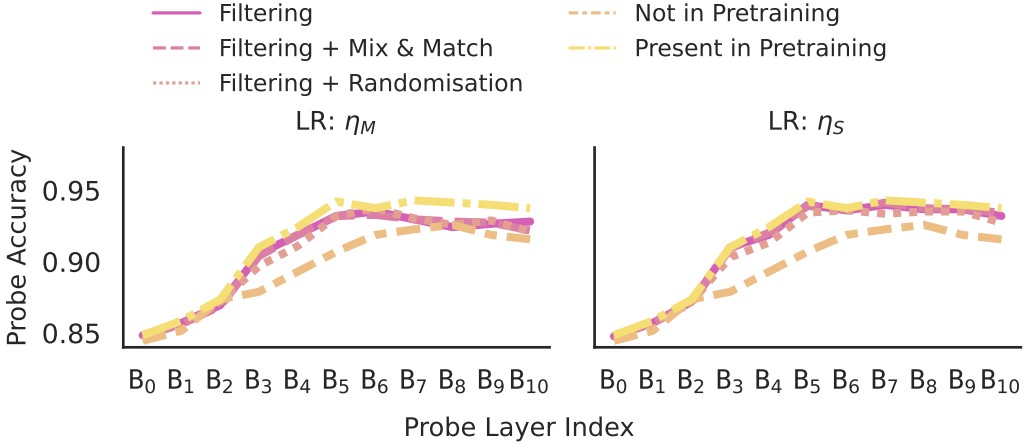

Figure 35: **Probing the presence of capabilities in TinyStories Models.** We plot probe accuracy of classifying whether a story contains a `Twist` or not wrt. the layer of the Transformer model (similarly to Fig. 8). Accuracy on models pre-trained with or without `Twist` data (*Present/Not in Pretraining* respectively) act as upper and lower bounds on the expected accuracy of the probes, and are plotted on both LR figures for ease of comparison, although they do not use a fine-tuning learning rate. We find that regardless of fine-tuning protocol (Filtering, Filtering + Randomisation, Filtering + Mix & Match), for the lower LR no fine-tuning protocol removes a meaningful amount of information from the activations, and a similar but less strong trend holds for the higher LR, implying that the pre-trained model retains its capability of story identification (a necessary capability for story modelling) throughout fine-tuning. Identical to Fig. 11

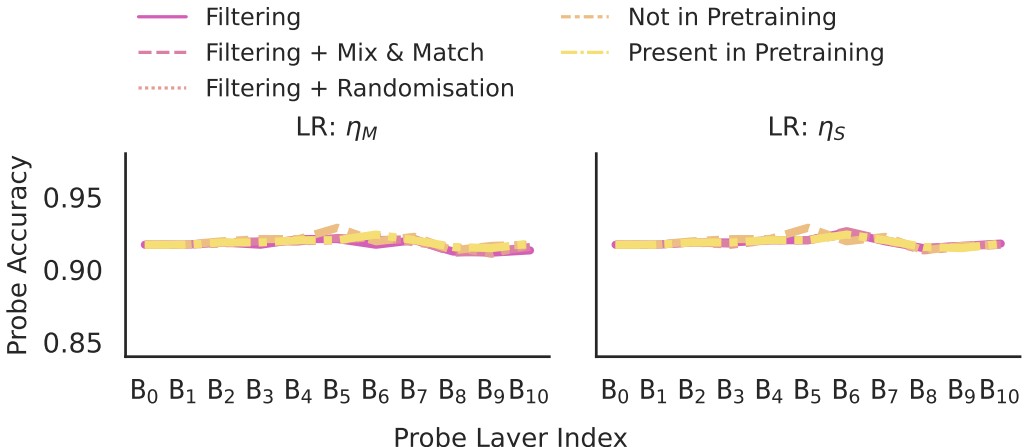

Figure 36: **Probing the presence of capabilities in TinyStories Models.** We plot probe accuracy of classifying whether a story contains the Foreshadowing feature or not wrt. the layer of the Transformer model. All other details the same as Fig. 35

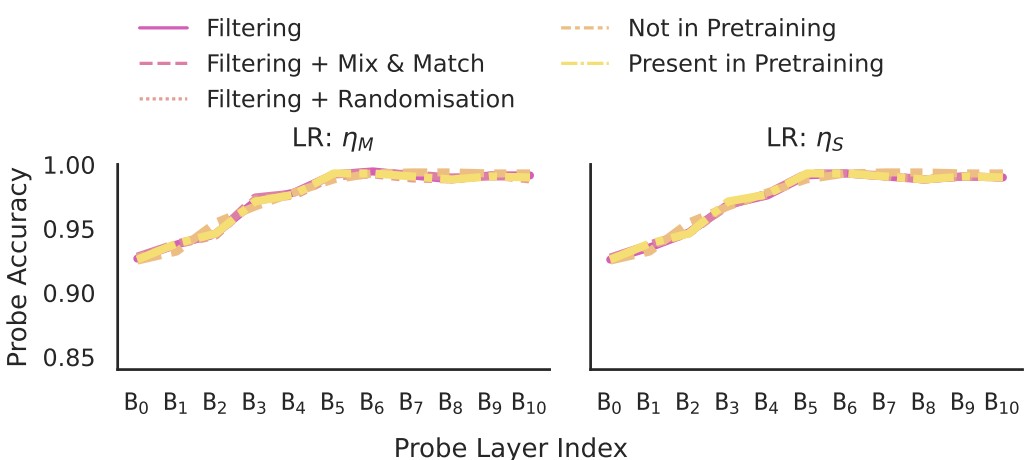

Figure 37: **Probing the presence of capabilities in TinyStories Models.** We plot probe accuracy of classifying whether a story contains the Moral Value feature or not wrt. the layer of the Transformer model. All other details the same as Fig. 35

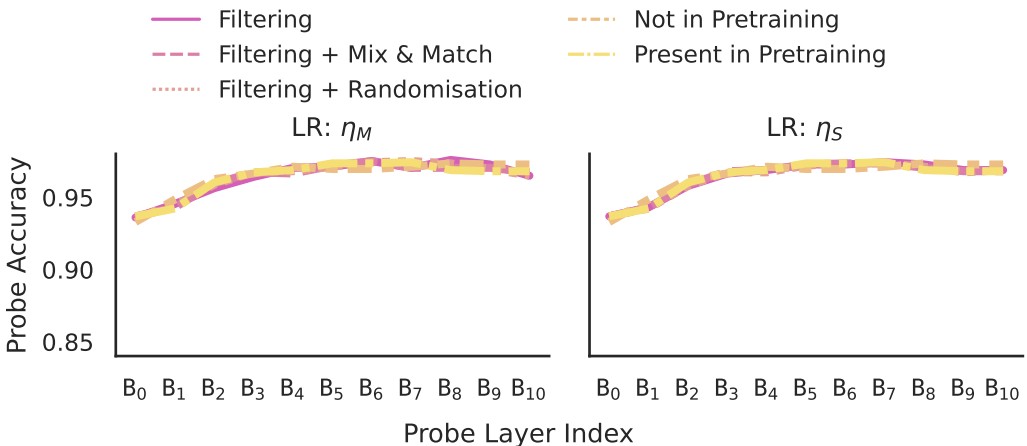

Figure 38: **Probing the presence of capabilities in TinyStories Models.** We plot probe accuracy of classifying whether a story contains the Bad Ending feature or not wrt. the layer of the Transformer model. All other details the same as Fig. 35

## G ADDITIONAL TRACR RESULTS

In this section, we present additional results on the counting and max-identifier tasks with Tracr models. These results provide an extensive analysis and support our claims presented in the main paper. Firstly, we present the detailed attention maps showing the full input sequence data: Fig. 40 and Fig. 39 show the attention maps corresponding to Fig. 42 and Fig. 43. We now present the detailed results on Tracr's Counter tasks.

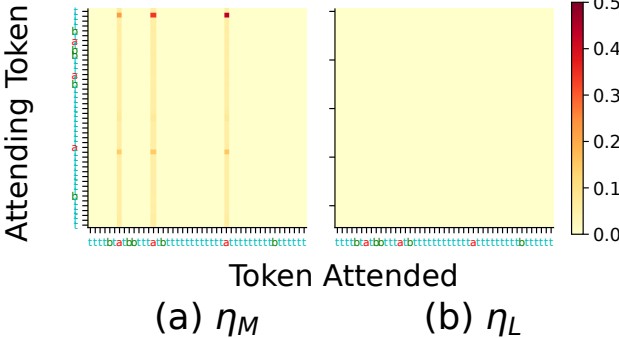

(a) $\eta_M$         (b) $\eta_L$

Figure 39: **Counter Task: Detailed visualization of the capability revival analysis on Tracr compiled for the Counter task. Observation:** On using $\eta_{VS}$, we are able to revive the compiled capability of the Tracr model fine-tuned with $\eta_M$, whereas using $\eta_L$ for fine-tuning hampers the underlying capability of the model and therefore there is no revival seen.

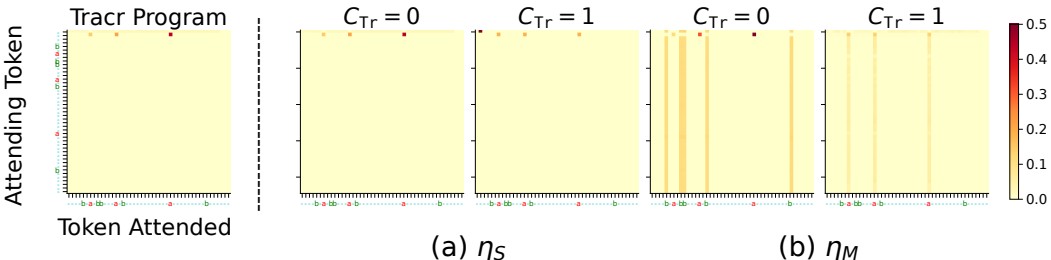

(a) $\eta_S$         (b) $\eta_M$

Figure 40: **Counter Task: Detailed visualization of the self attention layer in Block-1 of the Tracr fine-tuned model. Observation:** (a) shows the map for the compiled Tracr model, where we observe that a's are being attended by the second token in the input sequence. (b) On using $\eta_S$ for fine-tuning in the absence of spurious correlations ($C_{Te} = 0$), a's are still being attended in the attention map. (c) On using $\eta_M$, the model learns the fine-tuning task as b's are also being attended now. On fine-tuning in the presence of the spurious correlation ($C_{Te} = 1$), the model doesn't learn to attend b's, but rather learns the spurious correlation.

### G.1 BEHAVIORAL RESULTS ON FINE-TUNING

**Summary of results on the Counter task.** As shown in Fig. 42, on using $\eta_M$, the model seems to learn a new capability of counting b's (the model primarily attends to b in its attention map). However, on using a small learning rate ($\eta_S$) of $10^{-3}$, in the absence of correlations, the model is not able to learn to attend to b's in its attention map. Thus the model is not able to learn the capability of counting b's. As shown in Tab. 3, in the presence of spurious correlations however, the model is able to learn the spurious correlation and achieve high accuracy on the correlated test set. We also present the visualization of the attention maps after reverse fine-tuning in Fig. 43, where it is observed that on using $\eta_M$ for fine-tuning, revival of capability is possible even on using a very small learning rate ($\eta_{vs}$) of $10^{-4}$. We present detailed results on reverse fine-tuning in Tab. 4 Whereas, in case the model is fine-tuned with a large learning rate ($\eta_L$) of $10^{-1}$, revival of capability is not possible. We also present analysis of single weight pruning and grafting in Fig. 55. On pruning off a single weight from the Tracr model compiled to count a's, the model can achieve a boost in accuracy of over 60% on the task of counting a's. This observation is evident only when the Tracr model is fine-tuned on correlated dataset.

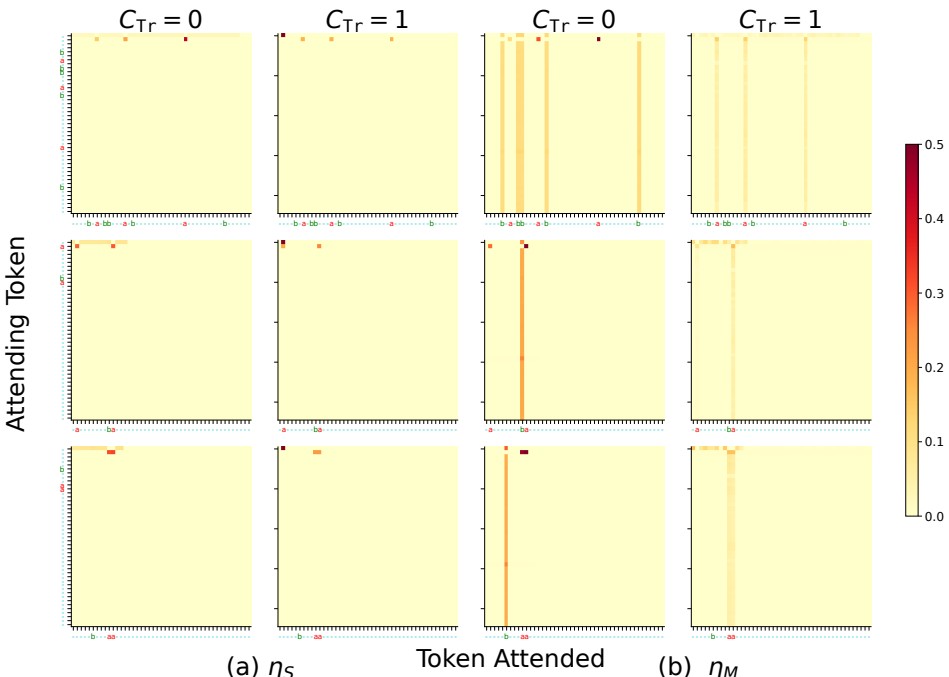

Figure 41: **Counter Task: Validation of Tracr observations on Counter task using three different input samples.** The rows represents different input samples. **Observation:** Using $\eta_S$ for fine-tuning the Tracr model compiled on Counter task is unable to learn to attend b's (a). But the model learns the spurious correlation. On the other hand, on using $\eta_M$ the model is able to learn the fine-tuning task by attending to b's. But in the presence of the spurious correlation $C_{\mathrm{Tr}} = 1$ the model still doesn't learn to attend b's.

**Summary of results on the max-identifer task.** We present a visualization of the attention maps for the max identifier task in Fig. 44, where we observe that Tracr model implements the sorting and the reading functions in the attention maps in blocks 0 and 2 respectively. On fine-tuning the model using different learning rates, the sorting capability implemented in Block-0, gets distorted, thereby resulting in poor fine-tuning performance (as evident in Tab. 3). However using $\eta_{VS}$ ($10^{-4}$), changes the reading function, without disturbing the sorting function. Thus the model is able to perform well on the downstream task (as evident in Tab. 3).

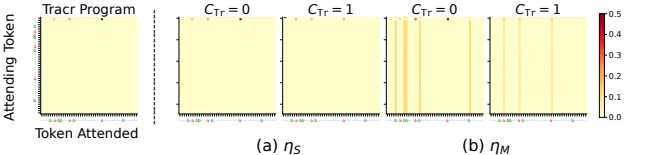

Figure 42: **New capabilities are not learned on using $\eta_S$ for fine-tuning.** (a) The counting a's capability implemented by the originally compiled Tracr model is to attend to a's. On using $\eta_S$ ($10^{-3}$) for fine-tuning, the compiled model is not able to learn the capability of counting b's (b). Increasing the learning rate makes the model learn to attend b's (c), but the pretraining capability of attending to a's still exists.

Figure 43: **Capability Revival Analysis:** Using $\eta_{VS}$ (a) is able to recover the old capability on reverse fine-tuning the model fine-tuned with $\eta_M$. But $\eta_S$ is not able to recover the original capability, when the compiled model is fine-tuned with $\eta_l$. This is because using a large value of learning rate during=fine-tuning hampers the pre-training capabilities.

## G.2  COUNTER RESULTS

A detailed analysis of the attention maps of block-1 and 2 for different learning rates is shown in Fig. 45. We further validate our results for three different input datapoints in Fig. 41 and Fig. 49. A detailed analysis of the activation map in Block-1 for different values of $C_{\mathrm{Tr}}$ is shown in Fig. 48.

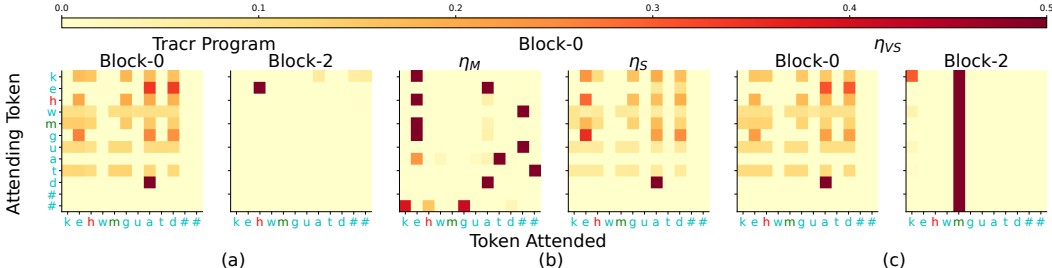

(a)            (b)            (c)

Figure 44: **Learning of the fine-tuning capability is affected by type of compiled capability present in the model.** (a) The Tracr program implements the sorting function in Block-0 and read function in Block-2. Using $\eta_M$ and $\eta_S$ can destroy the sorting capability present in Block-1 (b). But using $\eta_{VS}$, preserves the sorting capability (c). Thus on using $\eta_{VS}$, the model learns to read a different stream of output, while preserving the sorting capability.

Table 3: **Results on counting and max element task Tracr setups.** The evaluation is done on test sets with and without the spurious correlation. The Tracr compiled models are fine-tuned for different learning rates and different value of $C_{\mathrm{Tr}}$.

| $\eta$ | $C_{\mathrm{Tr}}$ | Counting Element | | | | Max Identifier | | | |
|---|---|---|---|---|---|---|---|---|---|
| | | $C_{\mathrm{Te}}=1$ | | $C_{\mathrm{Te}}=0$ | | $C_{\mathrm{Te}}=1$ | | $C_{\mathrm{Te}}=0$ | |
| | | Acc. $O_{\mathrm{PT}}$ | Acc. $O_{\mathrm{FT}}$ | Acc. $O_{\mathrm{PT}}$ | Acc. $O_{\mathrm{FT}}$ | Acc. $O_{\mathrm{PT}}$ | Acc. $O_{\mathrm{FT}}$ | Acc. $O_{\mathrm{PT}}$ | Acc. $O_{\mathrm{FT}}$ |
| $10^{-1}$ | 0 | 0.0 | 100.0 | 0.0 | 100.0 | 20.3 | 34.5 | 0.0 | 99.3 |
| | 0.2 | 0.0 | 100.0 | 0.0 | 100.0 | 0.5 | 92.0 | 0.0 | 97.1 |
| | 0.5 | 0.0 | 100.0 | 0.0 | 100.0 | 0.6 | 97.0 | 0.1 | 97.6 |
| | 0.6 | 0.0 | 100.0 | 0.0 | 100.0 | 0.3 | 98.5 | 0.0 | 96.7 |
| | 0.8 | 0.0 | 100.0 | 0.0 | 100.0 | 0.1 | 99.4 | 0.0 | 98.6 |
| | 0.9 | 0.0 | 100.0 | 0.0 | 100.0 | 0.7 | 98.2 | 0.1 | 92.5 |
| | 1 | 0.0 | 100.0 | 35.8 | 0.7 | 0.3 | 99.6 | 16.8 | 37.8 |
| $10^{-2}$ | 0 | 1.1 | 96.3 | 0.0 | 98.8 | 29.7 | 0.2 | 16.3 | 52.6 |
| | 0.2 | 0.0 | 100.0 | 0.0 | 99.2 | 28.4 | 18.6 | 19.0 | 46.0 |
| | 0.5 | 0.6 | 99.4 | 0.0 | 95.9 | 4.8 | 87.9 | 3.3 | 92.6 |
| | 0.6 | 0.1 | 99.9 | 0.0 | 98.8 | 3.9 | 83.2 | 2.6 | 82.5 |
| | 0.8 | 0.3 | 99.6 | 0.1 | 97.0 | 4.5 | 88.8 | 6.6 | 72.9 |
| | 0.9 | 1.4 | 98.5 | 7.1 | 39.3 | 16.0 | 45.7 | 26.9 | 11.1 |
| | 1 | 0.3 | 98.3 | 4.2 | 0.2 | 11.1 | 78.5 | 23.8 | 14.4 |
| $10^{-3}$ | 0 | 54.6 | 1.2 | 25.7 | 27.2 | 6.4 | 20.2 | 4.5 | 28.5 |
| | 0.2 | 50.2 | 15.0 | 26.5 | 24.3 | 7.4 | 27.4 | 5.4 | 28.0 |
| | 0.5 | 7.1 | 90.9 | 19.8 | 2.3 | 11.3 | 24.0 | 7.6 | 20.8 |
| | 0.6 | 4.1 | 94.2 | 11.8 | 2.2 | 11.8 | 26.7 | 8.4 | 20.1 |
| | 0.8 | 1.3 | 98.3 | 6.7 | 0.7 | 11.5 | 34.3 | 8.5 | 19.9 |
| | 0.9 | 1.8 | 97.8 | 9.2 | 0.7 | 14.6 | 32.2 | 11.4 | 15.8 |
| | 1 | 4.0 | 94.3 | 10.3 | 2.2 | 16.0 | 33.2 | 12.8 | 14.0 |
| $10^{-4}$ | 0 | 32.6 | 0.0 | 10.6 | 28.7 | 0.5 | 82.6 | 0.5 | 91.1 |
| | 0.2 | 59.2 | 0.1 | 31.9 | 24.4 | 0.1 | 84.8 | 0.6 | 91.3 |
| | 0.5 | 28.5 | 65.1 | 37.8 | 5.6 | 0.0 | 89.3 | 0.6 | 91.8 |
| | 0.6 | 24.4 | 70.3 | 35.6 | 4.8 | 0.0 | 89.6 | 0.6 | 90.8 |
| | 0.8 | 14.1 | 84.2 | 29.7 | 2.1 | 0.0 | 89.7 | 0.6 | 89.9 |
| | 0.9 | 1.3 | 98.3 | 6.7 | 0.7 | 0.0 | 93.2 | 0.2 | 97.1 |
| | 1 | 1.6 | 98.3 | 10.6 | 0.2 | 0.0 | 90.2 | 0.7 | 88.6 |

We present an evidence further, showing that capability of the Tracr compiled model to count a's is still present in the model in Fig. 46, 47, where Fig. 46 presents a closer look of the Fig. 47. As can be seen in Fig. 46, on using $\eta_S$ and $\eta_M$, Block-1 activation map of the Tracr fine-tuned model shows neurons corresponding to token a being activated in a different output channel.

Finally, we present evidence of the wrapper being learned by the model on fine-tuning using spuriously correlated dataset. We show that this wrapper can be localized in a very few neurons of the model.

Table 4: **Results on counting task Tracr for reverse fine-tuning with different learning rates.** Fine-tuning was done using $\eta_M$. The evaluation is done on test sets with and without the spurious correlation. The Tracr compiled models are fine-tuned for different learning rates and different value of $C_{\mathrm{Tr}}$.

| $\eta$ | $C_{\mathrm{Tr}}$ | Counting Element | | | | | | | |
|---|---|---|---|---|---|---|---|---|---|
| | | $C_{\mathrm{Te}}=1$ | | $C_{\mathrm{Te}}=0$ | | $C_{\mathrm{Te}}=1$ | | $C_{\mathrm{Te}}=0$ | |
| | | Acc. $0_{\mathrm{PT}}$ | Acc. $0_{\mathrm{FT}}$ | Acc. $0_{\mathrm{PT}}$ | Acc. $0_{\mathrm{FT}}$ | Acc. $0_{\mathrm{PT}}$ | Acc. $0_{\mathrm{FT}}$ | Acc. $0_{\mathrm{PT}}$ | Acc. $0_{\mathrm{FT}}$ |
| $10^{-1}$ | 0 | 96.5 | 100.0 | 2.1 | 0.0 | 94.0 | 100.0 | 0.1 | 0.0 |
| | 0.2 | 98.5 | 100.0 | 0.1 | 0.0 | 94.9 | 100.0 | 0.1 | 0.0 |
| | 0.5 | 39.4 | 100.0 | 43.6 | 0.0 | 44.9 | 100.0 | 5.6 | 0.0 |
| | 0.6 | 72.9 | 100.0 | 26.7 | 0.0 | 69.4 | 100.0 | 0.2 | 0.0 |
| | 0.8 | 49.3 | 100.0 | 6.5 | 0.0 | 37.1 | 100.0 | 16.6 | 0.0 |
| | 0.9 | 31.3 | 100.0 | 64.0 | 0.0 | 34.1 | 100.0 | 1.7 | 0.0 |
| | 1 | 69.2 | 100.0 | 3.7 | 0.0 | 65.4 | 100.0 | 6.3 | 0.0 |
| $10^{-2}$ | 0 | 63.3 | 99.9 | 36.6 | 0.1 | 65.5 | 98.6 | 0.0 | 0.0 |
| | 0.2 | 19.5 | 100.0 | 48.8 | 0.0 | 29.6 | 100.0 | 17.5 | 0.0 |
| | 0.5 | 14.3 | 100.0 | 54.4 | 0.0 | 28.9 | 99.9 | 18.7 | 0.0 |
| | 0.6 | 86.2 | 99.9 | 13.8 | 0.1 | 78.3 | 98.6 | 0.0 | 0.0 |
| | 0.8 | 65.6 | 100.0 | 1.7 | 0.0 | 43.7 | 99.5 | 10.8 | 0.0 |
| | 0.9 | 33.3 | 100.0 | 27.9 | 0.0 | 36.5 | 100.0 | 12.6 | 0.0 |
| | 1 | 99.0 | 99.6 | 0.9 | 0.3 | 95.2 | 96.8 | 0.1 | 0.1 |
| $10^{-3}$ | 0 | 19.8 | 99.9 | 34.9 | 0.0 | 23.7 | 98.6 | 10.2 | 0.0 |
| | 0.2 | 3.9 | 17.1 | 33.8 | 78.0 | 24.4 | 42.9 | 14.7 | 3.0 |
| | 0.5 | 2.0 | 87.6 | 33.2 | 9.5 | 18.9 | 85.7 | 11.9 | 0.9 |
| | 0.6 | 7.1 | 99.8 | 35.4 | 0.1 | 22.3 | 97.3 | 15.4 | 0.1 |
| | 0.8 | 11.6 | 97.2 | 45.8 | 0.3 | 27.3 | 95.5 | 16.5 | 0.5 |
| | 0.9 | 24.2 | 75.2 | 20.3 | 23.6 | 33.5 | 68.8 | 13.1 | 1.5 |
| | 1 | 68.5 | 99.9 | 26.7 | 0.1 | 65.3 | 98.6 | 5.8 | 0.0 |
| $10^{-4}$ | 0 | 45.2 | 99.9 | 0.1 | 0.1 | 16.9 | 98.5 | 4.9 | 0.0 |
| | 0.2 | 30.1 | 9.4 | 22.7 | 45.1 | 18.1 | 26.2 | 17.9 | 19.5 |
| | 0.5 | 30.1 | 3.2 | 22.7 | 41.3 | 15.7 | 26.1 | 17.9 | 16.2 |
| | 0.6 | 54.1 | 0.0 | 45.9 | 35.9 | 0.0 | 27.8 | 25.7 | 11.9 |
| | 0.8 | 26.8 | 83.0 | 64.5 | 10.9 | 3.0 | 76.8 | 49.2 | 1.3 |
| | 0.9 | 27.9 | 85.0 | 61.5 | 8.4 | 2.1 | 80.9 | 44.2 | 1.1 |
| | 1 | 45.2 | 99.6 | 31.9 | 0.3 | 42.2 | 96.8 | 12.6 | 0.0 |

As shown in Fig. 55, we present this evidence for different values of $C_{\mathrm{Tr}}$ in Fig. 56. Similar to the analysis presented for PCFG where we prune multiple neurons, we analyze the effect of pruning of mutliple weights and neurons in Fig. 57 and Fig. 58 respectively. These results verify that the wrapper learned by the Tracr compiled model on fine-tuning using spuriously correlated dataset can indeed be localized to a few weights of the Tracr model. To ensure that the gains achieved on pruning are indeed because of removal of the wrapper, we present the histograms showing the distribution of the predicted classes in Fig. 59, where it can be observed that after pruning the model still predicts multiple classes.

## G.3 MAX IDENTIFIER RESULTS

In this section, we provide additional evidence and a detailed analysis of the performance of the Tracr compiled model on the max identifier task. We show that the model implements the sorting pattern in the activation map of its first block in Fig. 53 and Fig-54. We present validation of our observations on considering the spurious correlation as the difference between the fifth and seventh maximmum element being three in Fig. 51. We validate our results for three different input data-points in Fig. 50. A detailed visualization of the attention maps in Block-0 and Block-2 for different learning rates is shown in Fig. 52.

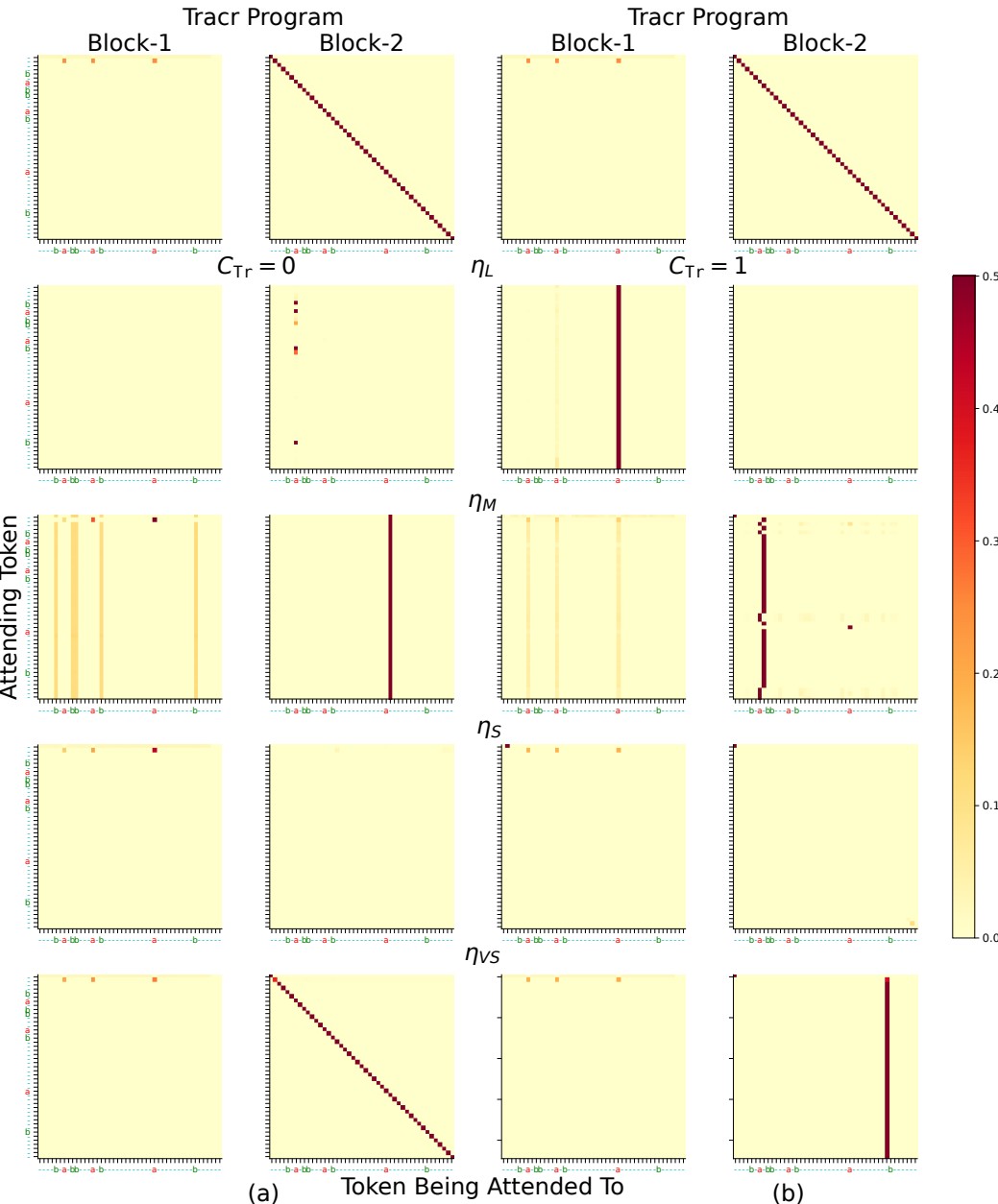

Figure 45: **Counter Task: Visualization of the attention maps of the first and second blocks of the Tracr fine-tuned models.** (a) shows the analysis when the spurious correlation is not present in the fine-tuning datatset, whereas in case of (b) the spurious correlation is present in the fine-tuning dataset. The first row shows the maps for the Tracr compiled model and other rows shows the analysis for different learning rates. **Observation:** (a) Using $\eta_S$ or $\eta_{VS}$ the model is not able to learn to attend b's and thus the fine-tuning task performance is poor. Whereas using $\eta_M$ the model is able to learn to attend to b's, however the capability to count a's is likely still present since the model still attends to a's. Further increasing the learning rate leads to distortion of the compiled capabilities, and thus model learns the fine-tuning task by learning a different capability. (b) In the presence of spurious correlation, even for large learning rate the compiled capability is still present, since the model attends to a's.

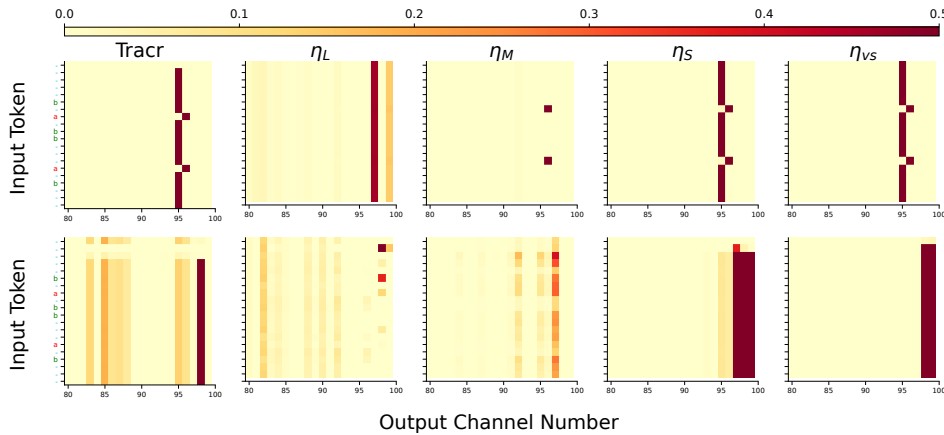

Figure 46: **Counter Task: Visualization of the activated output of the first MLP layer in first and second blocks.** The visualization is shown only for channel numbers 80-100. **Observation:** Using $\eta_M$ preserves the Tracr compiled capability, while also learning the fine-tuning task. This shows that the model changes behaviourally but mechanistically the compiled capabilities are still present in it.

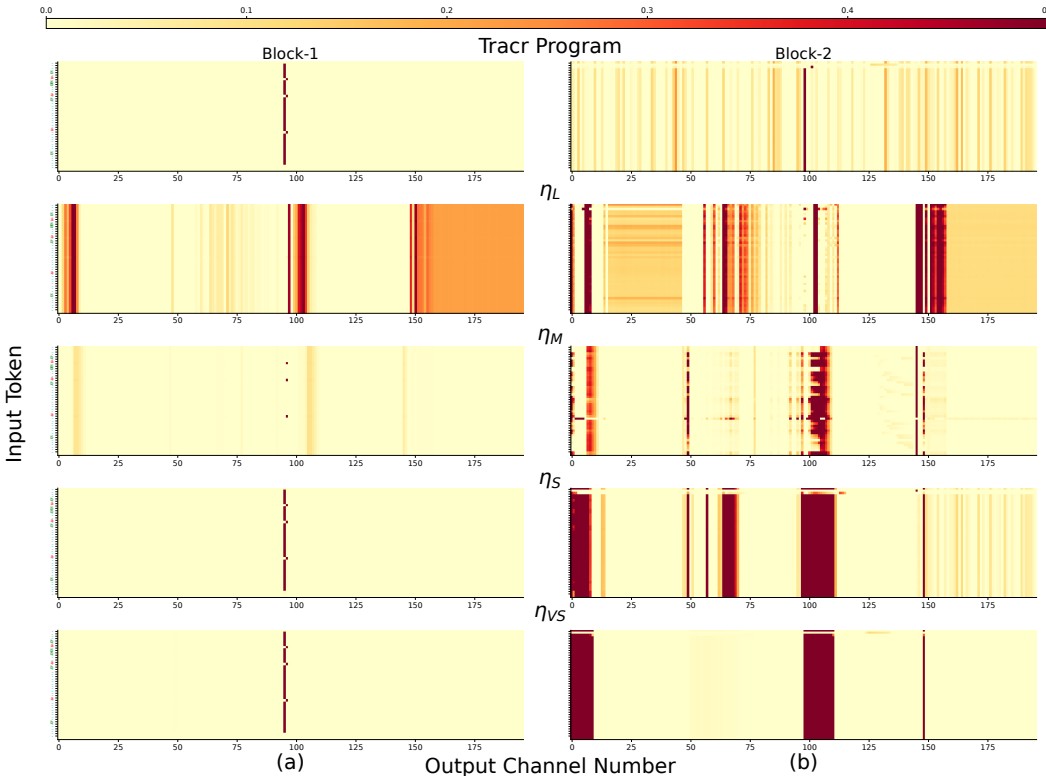

Figure 47: **Counter Task: Visualization of the activated output of the first MLP layer in first and second blocks.** This is the complete visualization of the activation map presented in Fig. 46.

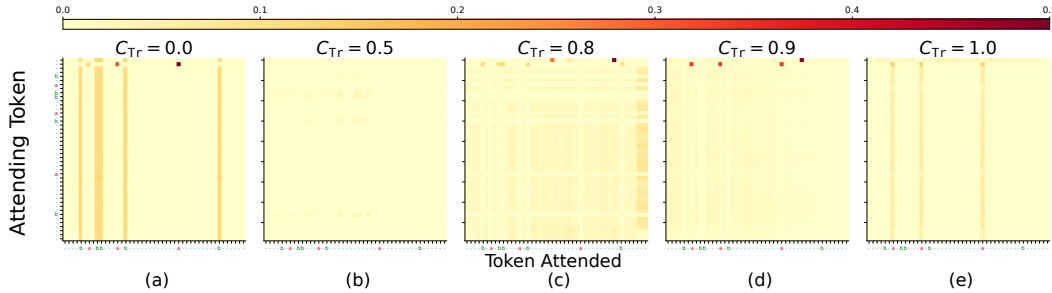

Figure 48: **Counter Task: Effect of the presence of the spuriously correlated data-points in different fractions ($C_{\text{Tr}}$) in the fine-tuning dataset.** The Tracr compiled model with the capability to count a's is fine-tuned to count b's on different values of $C_{\text{Tr}}$. $\eta_M$ is used for fine-tuning. **Observation:** On increasing the value of $C_{\text{Tr}}$, the model gives lower attention to b's and in case of $C_{\text{Tr}} = 1$, almost no attention is given by the model to b's.

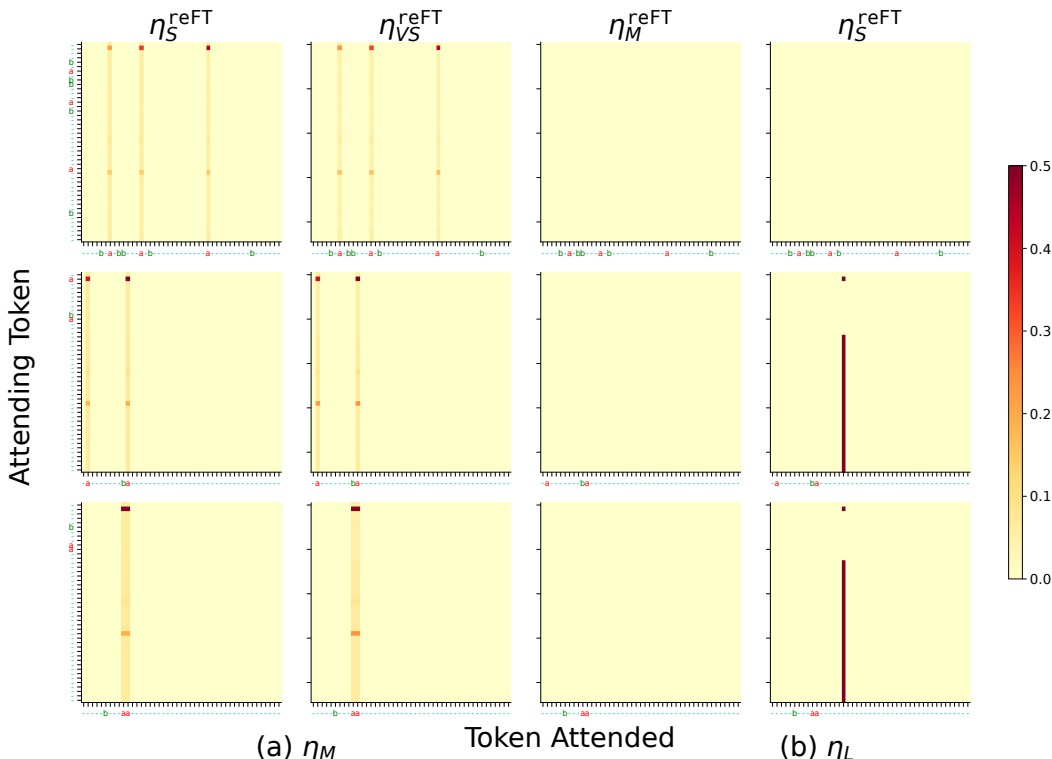

Figure 49: **Counter Task: Validation of Tracr observations on Counter task on reverse fine-tuning on three different input samples.** The rows represents different input samples. **Observation:** Capability revival is possible on using $\eta_M$ for fine-tuning but not on using $\eta_L$.

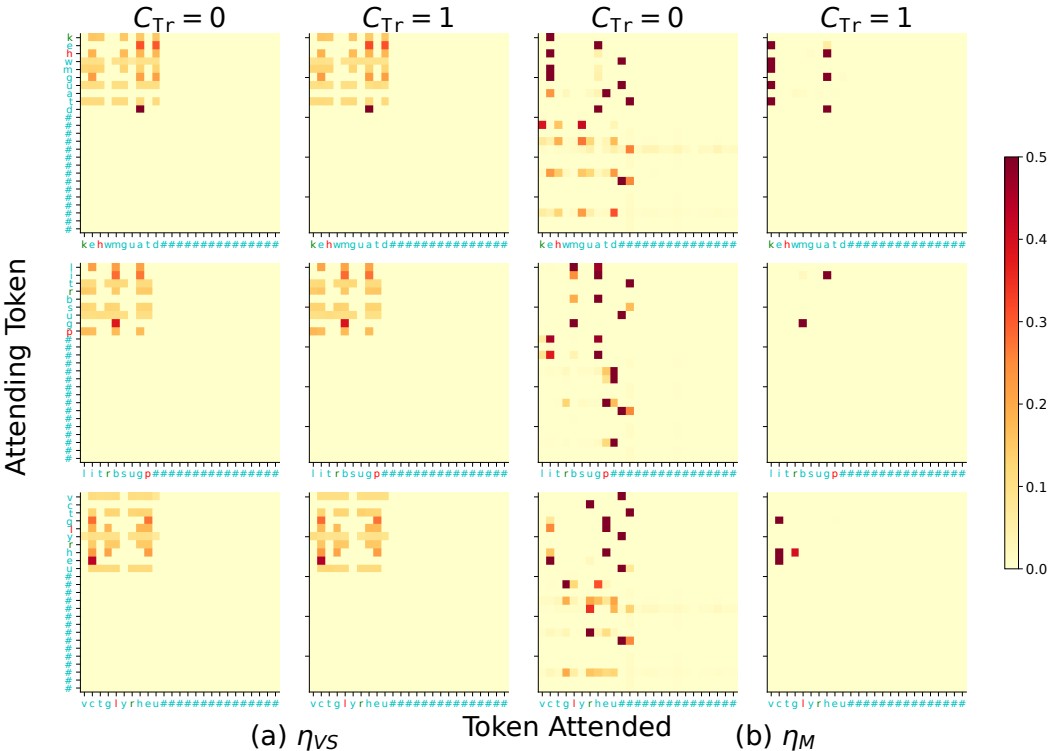

Figure 50: **Max Identifier Task: Validation of Tracr observations on max identifier task on three different input samples.** The rows represents different input samples. **Observation:** Using $\eta_{VS}$ preserves the original Tracr capabilities and therefore performs well on the fine-tuning task, whereas using $\eta_M$ distorts the compiled capabilities resulting in poor performance on the fine-tuning task.

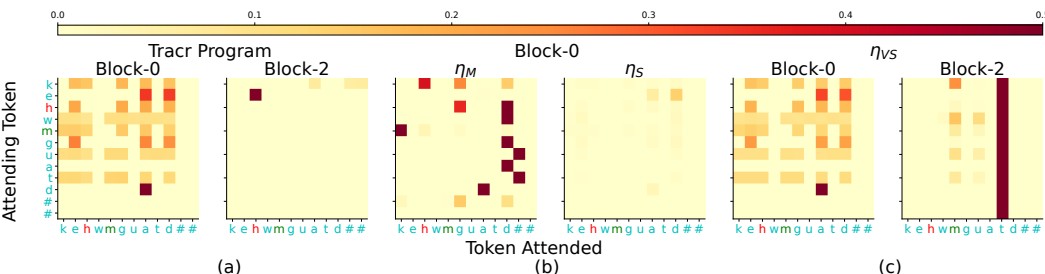

Figure 51: **Max Identifier Task: Validation of Tracr observations on max identifier task with the spurious correlation defined as the difference between the indices of fifth and seventh maximum elements being three.** The rows represents different input samples. **Observation:** Using $\eta_{VS}$ preserves the original Tracr capabilities and therefore performs well on the fine-tuning task, whereas using $\eta_M$ distorts the compiled capabilities resulting in poor performance on the fine-tuning task.

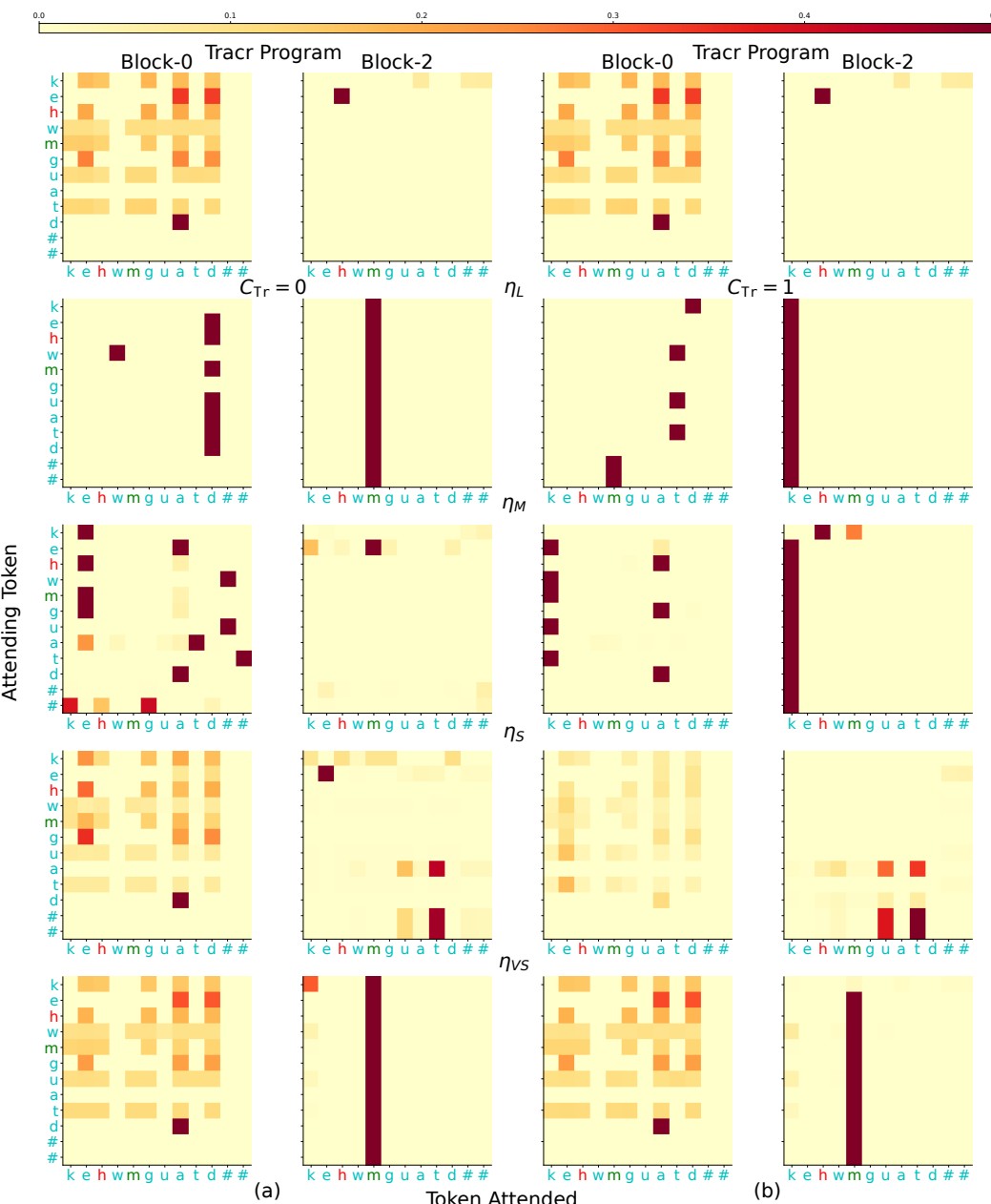

Figure 52: **Max Identifier Task: Visualization of the attention maps of the zeroth and second blocks of the Tracr fine-tuned models on the max identifier task.** (a) shows the analysis when the spurious correlation is not present in the fine-tuning datatset, whereas in case of (b) the spurious correlation is present in the fine-tuning dataset. The first row shows the maps for the Tracr compiled model and other rows shows the analysis for different learning rates. **Observation:** Using $\eta_L$, $\eta_M$ or $\eta_S$ for fine-tuning distorts the capability of the programmed Tracr model in the Block-0 and as a result the Block-2 attention map is not able to attend to the desired output token. Whereas using $\eta_{VS}$ is able to preserve the capability and as a result the fine-tuned model is able to attend to the correct token in the attention map in Block-2.

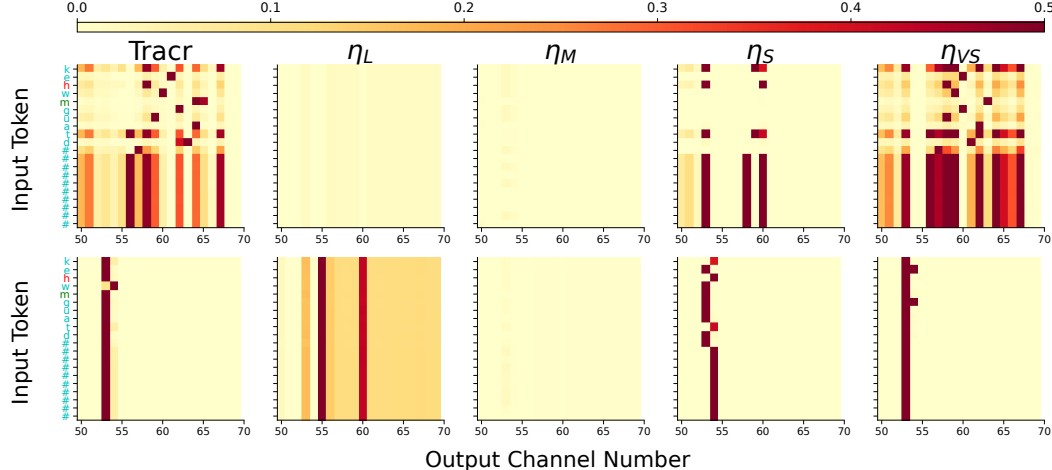

Figure 53: **Max Identifier Task: Visualization of the activated output of the first MLP layer in first and second blocks for the max identifier task.** The visualization is shown only for channel numbers 50-70. **Observation:** Using $\eta_{VS}$ for fine-tuning, which enables the model to learn the fine-tuning task, preserves the Tracr compiled model's compiled capability of sorting tokens in Block-1. Whereas other learning rates are not able to preserve this capability.

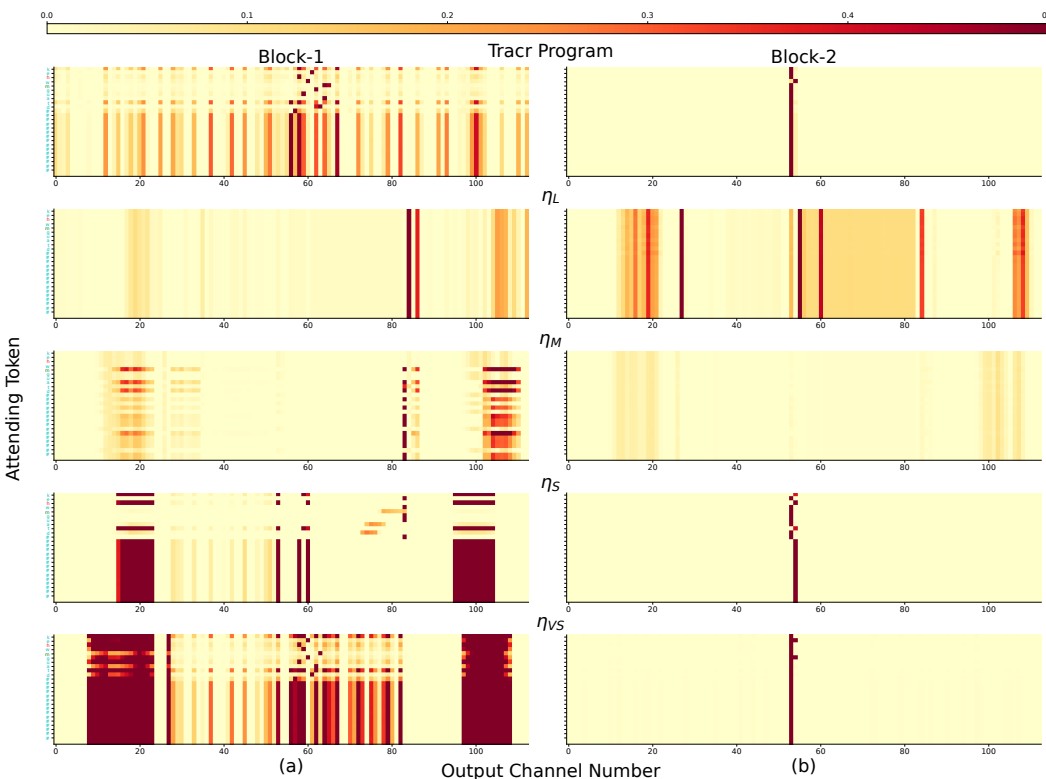

Figure 54: **Max Identifier Task: Visualization of the activated output of the first MLP layer in first and second blocks.** This is the complete visualization of the activation map presented in Fig. 53.

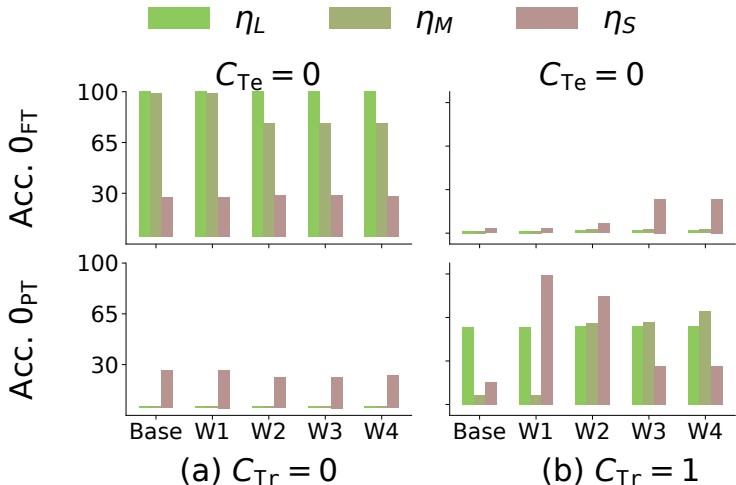

Figure 55: **Counter Task: Pruning evaluation on Tracr model fine-tuned to count** b's. **Observation:** Higher value of $C_{\text{Tr}}$ leads to the learning of the wrapper on top of the Tracr compiled capability. This wrapper is learned on using $\eta_M$ and $\eta_S$ and can be localized in a few weights of the model.

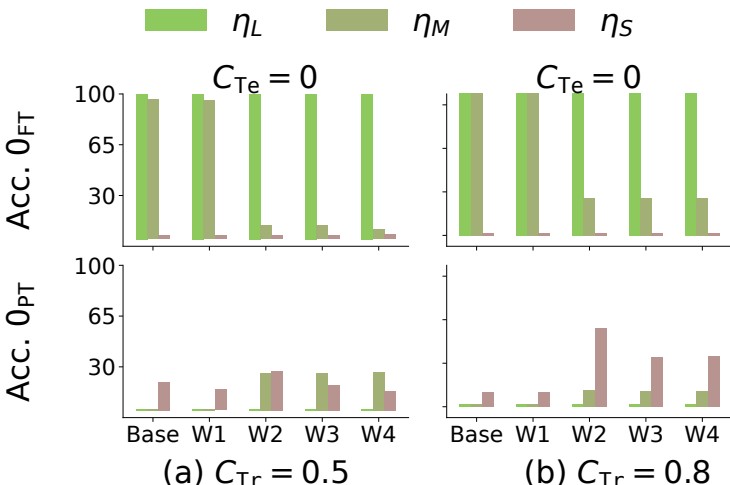

Figure 56: **Counter Task: Pruning evaluation on Tracr model fine-tuned to count** b's. **Observation:** Observations are consistent with Fig-7.

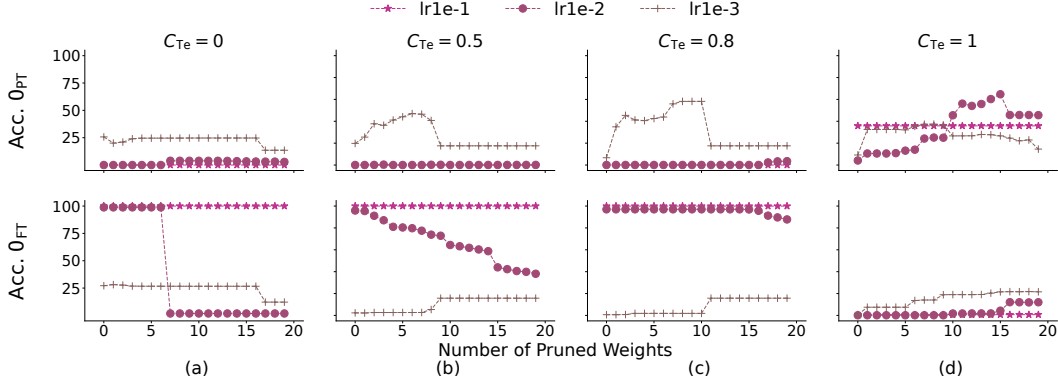

Figure 57: **Counter Task, weight pruning: Pruning weights of the Tracr model fine-tuned to count b's using different learning rates. Observation:** In the presence of spurious correlation the model learns a wrapper when learning rates $\eta_M$ and $\eta_S$ are used. This wrapper can be localized in a few weights of the model.

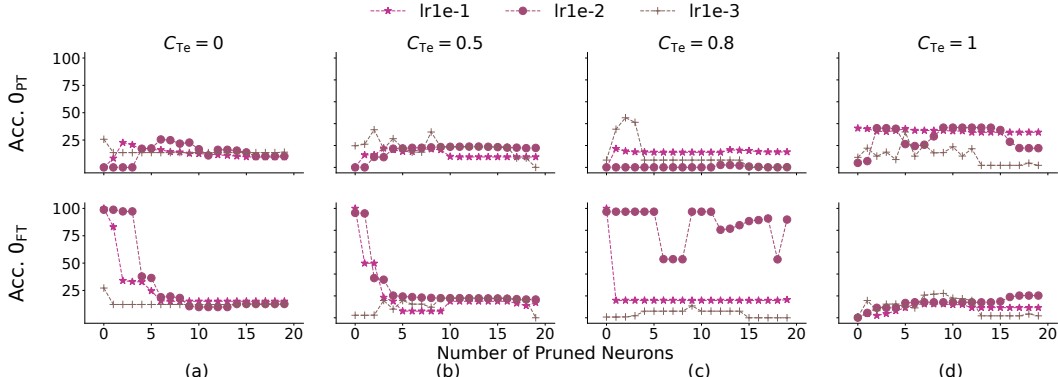

Figure 58: **Counter Task, neuron pruning: Pruning neurons of the Tracr model fine-tuned to count b's using different learning rates. Observation:** In the presence of spurious correlation the model learns a wrapper in case of $\eta_M$ and $\eta_S$. This wrapper can be localized in a few weights of the model.

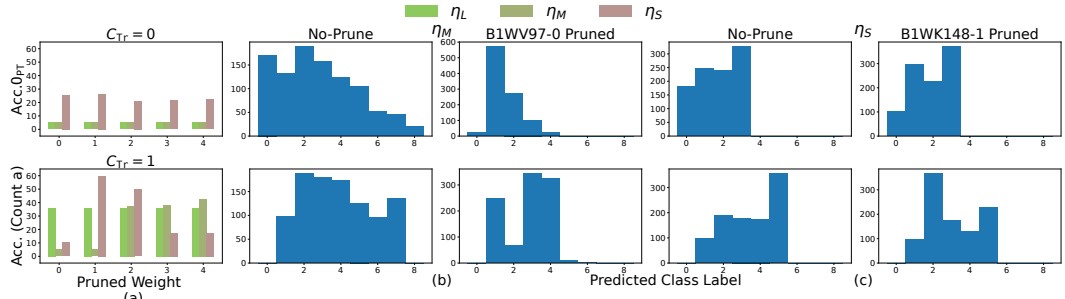

Figure 59: **Counter Task: Effect of pruning a single weight on the distribution of predicted class labels. Observation:** Even after pruning, the model predicts different classes indicating that the gain in accuracy on pruning is indeed because the model has removed the wrapper.

# H  ADDITIONAL PCFG RESULTS

In this section, we provide a detailed analysis of the PCFG results on the counter task. More specifically, we analyze the effect of the presence of weakly and strongly relevant capability in the model across three different parameters: training iterations ($n_{iters}$), fraction of fine-tuning data samples with spurious correlation ($C_{Tr}$) and the probability of sampling operand token (0) to be a during pre-training ($\mathcal{P}_T$). $\mathcal{P}_T$ essentially determines whether the capability present in the pre-trained model is strongly relevant or weakly relevant for the fine-tuning task. Additionally we also analyze the effect of using the spuriously correlated ($C_{Te} = 1$) and uncorrelated test set ($C_{Te} = 0$) for evaluation of the fine-tuned model. We present the summary of the results in Tables 5, 6 for the counting task and Tables 7,6 for the index of occurrence tasks. Then we discuss the effect of learning rate on fine-tuning pre-trained models with weakly and strongly relevant capabilities in Fig. 60, 61. We observe that the observations are consistent for the considered counting and index of occurrence tasks. Next we analyze the effect of the presence of weakly and strongly relevant capability in the pre-trained model for different fractions of spuriously correlated data-points ($C_{Tr}$) and different pre-training iterations ($n_{iters}$) in Fig. 62, 64, 66 for the counting element task and Fig. 63, 65, 67 for the index of occurrence task. We observe that the observations are fairly consistent across both the tasks and different values of $n_{iters}$. Next we present the effect of the spuriously correlated data and presence of weakly and strongly correlated capabilities on the learning of the wrapper in Fig. 68, 69 on using uncorrelated test set for evaluation on counting and index of occurrence tasks respectively. Similar analysis on using test set with spuriously correlated samples is present in Fig. 70 and 71. We present the capability revival analysis on the Counter task for $n_{iters} = 200K$ and $n_{iters} = 50K$ pre-trained models for weakly and strongly relevant capability fine-tuned models in Fig. 76 and Fig. 77 respectively. A similar analysis for different number of pre-training iterations is present in Fig. 78.

Table 5: Results on the PCFG counting task with 200K iterations of pre-training.

| $\eta$ | $\mathcal{P}_T$ (a) | $C_{Tr}$ | | Acc 0$_{PT}$ | | | | | Acc 0$_{FT}$ | | | |
| --- | --- | --- | --- | --- | --- | --- | --- | --- | --- | --- | --- | --- |
| | | | Acc PT | $C_{Te}=0$ | | $C_{Te}=1$ | | Acc PT | $C_{Te}=0$ | | $C_{Te}=1$ | |
| | | | | 1K | 10K | 1K | 10K | | 1K | 10K | 1K | 10K |
| $10^{-4}$ | 0.999 | 0 | 100 | 9.7 | 9.5 | 14.5 | 0 | 27.1 | 99.4 | 100 | 74.9 | 87.5 |
| | | 0.5 | | 7.2 | 9.7 | 1.3 | 0 | | 75.9 | 99.9 | 95.9 | 100 |
| | | 0.8 | | 5.6 | 10.8 | 0.2 | 0 | | 60 | 99.8 | 98.1 | 100 |
| | | 1 | | 17.2 | 15.2 | 0.2 | 0 | | 0 | 1.6 | 98.9 | 100 |
| | 0.9 | 0 | 100 | 99.9 | 92.6 | 13.6 | 67.2 | 99.9 | 99.8 | 100 | 100 | 92.8 |
| | | 0.5 | | 100 | 90.7 | 15.6 | 73.5 | | 99.8 | 99.4 | 99.9 | 100 |
| | | 0.8 | | 99.9 | 43.4 | 11.4 | 33.2 | | 99.4 | 99.2 | 100 | 100 |
| | | 1 | | 99.8 | 15.9 | 16.5 | 2.4 | | 99.6 | 9.4 | 6.1 | 100 |
| | 0.5 | 0 | 99.9 | 98.9 | 14.7 | 76.6 | 44.1 | 99.9 | 99.9 | 100 | 87.6 | 99.2 |
| | | 0.5 | | 95.1 | 23.6 | 71.2 | 33.4 | | 99.7 | 100 | 99.8 | 99.9 |
| | | 0.8 | | 92.8 | 12.2 | 79.8 | 2.2 | | 99.9 | 99.9 | 98.7 | 99.9 |
| | | 1 | | 49.5 | 19 | 60.6 | 0 | | 25.8 | 16 | 99.8 | 100 |
| $10^{-5}$ | 0.999 | 0 | 100 | 48.9 | 10.2 | 51.9 | 4.6 | 27.1 | 39.8 | 99.8 | 13.9 | 79.7 |
| | | 0.5 | | 19.7 | 11.6 | 12.3 | 1.2 | | 18.4 | 98.1 | 81.4 | 99.7 |
| | | 0.8 | | 12.1 | 6.6 | 7.7 | 0.2 | | 6.1 | 85.7 | 98.4 | 99.7 |
| | | 1 | | 0.4 | 17.5 | 0 | 0 | | 0 | 0 | 99.9 | 100 |
| | 0.9 | 0 | 100 | 100 | 85.3 | 94.8 | 56.9 | 99.9 | 99.8 | 99.9 | 83.3 | 87.3 |
| | | 0.5 | | 99.9 | 67.2 | 94.9 | 55.4 | | 99.9 | 99.9 | 99.3 | 99.8 |
| | | 0.8 | | 100 | 34.6 | 94.8 | 21.7 | | 99.8 | 99.4 | 99.7 | 100 |
| | | 1 | | 98.5 | 13.5 | 88.6 | 0.8 | | 58.3 | 3.6 | 99.8 | 100 |
| | 0.5 | 0 | 100 | 100 | 97.5 | 97.5 | 65.7 | 99.9 | 100 | 100 | 95.6 | 95.4 |
| | | 0.5 | | 99.9 | 94.1 | 98.1 | 69 | | 100 | 100 | 99.3 | 100 |
| | | 0.8 | | 99.9 | 87.4 | 93.8 | 67.7 | | 99.8 | 100 | 100 | 100 |
| | | 1 | | 99.6 | 41.2 | 91.8 | 53.3 | | 90.1 | 19.6 | 99.8 | 100 |
| $10^{-6}$ | 0.999 | 0 | 100 | 100 | 29 | 96.6 | 25.7 | 27.1 | 28.5 | 51.8 | 15.1 | 29.2 |
| | | 0.5 | | 98.7 | 21.8 | 88.9 | 10.6 | | 23.3 | 23.7 | 20.3 | 87.5 |
| | | 0.8 | | 83 | 15.1 | 69.7 | 6.8 | | 18.4 | 8.9 | 26.5 | 99.7 |
| | | 1 | | 71.7 | 2.3 | 56 | 0 | | 15.7 | 0 | 29.5 | 99.9 |
| | 0.9 | 0 | 100 | 100 | 100 | 95.4 | 91.8 | 99.9 | 99.8 | 99.5 | 84.1 | 84.6 |
| | | 0.5 | | 100 | 99.9 | 96 | 94.4 | | 99.6 | 99.5 | 95.9 | 99.6 |
| | | 0.8 | | 99.8 | 99.4 | 95.9 | 92.6 | | 99.6 | 99.3 | 94.8 | 99.6 |
| | | 1 | | 99.8 | 51.6 | 95.1 | 63.9 | | 99.5 | 30.5 | 94.2 | 99.7 |
| | 0.5 | 0 | 99.9 | 100 | 99.9 | 97.7 | 98.1 | 99.9 | 99.8 | 99.8 | 95.4 | 95.6 |
| | | 0.5 | | 100 | 99.9 | 97.9 | 93.7 | | 100 | 99.9 | 98.5 | 99.6 |
| | | 0.8 | | 100 | 100 | 98.8 | 93.1 | | 100 | 99.9 | 98.3 | 99.9 |
| | | 1 | | 100 | 97.6 | 98.6 | 85.1 | | 99.8 | 73.9 | 98.3 | 100 |

Table 6: Results on the PCFG counting task with 50K iterations of pre-training.

| $\eta$ | $\mathcal{P}_{\mathrm{T}}(a)$ | $C_{\mathrm{Tr}}$ | Acc. PT | Acc $0_{\mathrm{PT}}$ | | | | Acc. PT | Acc. $0_{\mathrm{FT}}$ | | | |
|---|---|---|---|---|---|---|---|---|---|---|---|---|
| | | | | $C_{\mathrm{Te}}=0$ | | $C_{\mathrm{Te}}=1$ | | | $C_{\mathrm{Te}}=0$ | | $C_{\mathrm{Te}}=1$ | |
| | | | | $1K$ | $10K$ | $1K$ | $10K$ | | $1K$ | $10K$ | $1K$ | $10K$ |
| $10^{-4}$ | 0.999 | 0 | 99.9 | 10.8 | 9.1 | 2 | 0.1 | 5.17 | 98.9 | 100 | 86.3 | 93.6 |
| | | 0.5 | | 11.7 | 8.9 | 2.1 | 0.1 | | 90.2 | 99.9 | 97.9 | 99.8 |
| | | 0.8 | | 5.5 | 11 | 0 | 0 | | 64.9 | 100 | 98.9 | 99.9 |
| | | 1 | | 20.2 | 15.9 | 0 | 0 | | 0 | 1.9 | 99.9 | 100 |
| | 0.9 | 0 | 99.9 | 9.1 | 10.3 | 0.7 | 0.1 | 15.8 | 99.6 | 100 | 84.2 | 94.4 |
| | | 0.5 | | 11.4 | 10.6 | 1.5 | 0 | | 93.2 | 99.9 | 97.9 | 100 |
| | | 0.8 | | 4.2 | 9.2 | 0.1 | 0 | | 63.1 | 100 | 97.8 | 100 |
| | | 1 | | 18.4 | 16.4 | 0 | 0 | | 0.5 | 5 | 100 | 99.9 |
| | 0.5 | 0 | 99.8 | 87.7 | 10.1 | 62.6 | 0 | 99.7 | 99.9 | 100 | 89.3 | 93.5 |
| | | 0.5 | | 90.1 | 9.4 | 67.4 | 0 | | 99.5 | 100 | 99.9 | 100 |
| | | 0.8 | | 59.5 | 10.1 | 29.3 | 0.1 | | 99.2 | 99.9 | 100 | 99.9 |
| | | 1 | | 18.7 | 15.2 | 5.1 | 0 | | 17.4 | 14.2 | 100 | 100 |
| $10^{-5}$ | 0.999 | 0 | 99.9 | 3 | 9.3 | 16.6 | 0.6 | 5.17 | 32.6 | 99.8 | 12.4 | 88.7 |
| | | 0.5 | | 30.9 | 11.4 | 4.1 | 0.6 | | 12.7 | 99.1 | 93 | 99.7 |
| | | 0.8 | | 6.3 | 10.8 | 1 | 0.1 | | 1.4 | 93.9 | 99 | 99.8 |
| | | 1 | | 2.1 | 20.7 | 0.2 | 0 | | 0 | 0 | 99.8 | 99.9 |
| | 0.9 | 0 | 99.9 | 28 | 10.9 | 34.7 | 0.1 | 15.8 | 39.6 | 99.8 | 23.4 | 88.6 |
| | | 0.5 | | 33.2 | 8.9 | 4.2 | 0 | | 22.7 | 99.6 | 92.8 | 100 |
| | | 0.8 | | 13.1 | 9.3 | 2.9 | 0 | | 9.4 | 95.1 | 98.9 | 100 |
| | | 1 | | 1.9 | 19.8 | 0.3 | 0 | | 0 | 0.1 | 99.7 | 100 |
| | 0.5 | 0 | 99.8 | 99.6 | 73.7 | 88.1 | 46 | 99.7 | 99.9 | 99.9 | 86.4 | 89.2 |
| | | 0.5 | | 99.6 | 79.3 | 82.7 | 57.1 | | 99.8 | 99.9 | 99.1 | 99.9 |
| | | 0.8 | | 99.6 | 60.8 | 80 | 33.6 | | 99.5 | 99.9 | 99.3 | 100 |
| | | 1 | | 81.7 | 12.9 | 68.6 | 0.5 | | 46.4 | 16.1 | 98.8 | 100 |
| $10^{-6}$ | 0.999 | 0 | 99.9 | 94.1 | 18.6 | 81.9 | 18.8 | 5.17 | 9 | 49.1 | 0.4 | 24.4 |
| | | 0.5 | | 38.8 | 37.2 | 20 | 5.6 | | 6 | 23.8 | 50.7 | 93.7 |
| | | 0.8 | | 14.4 | 8.8 | 4.2 | 0.2 | | 0.9 | 7.8 | 74.8 | 99.9 |
| | | 1 | | 8.9 | 5.8 | 2.3 | 0.2 | | 1.3 | 0 | 79.2 | 99.8 |
| | 0.9 | 0 | 99.9 | 99.7 | 21 | 94.3 | 20.6 | 15.8 | 24.4 | 56.8 | 14.6 | 28.4 |
| | | 0.5 | | 46.9 | 38.6 | 19.7 | 3.6 | | 11.8 | 28.6 | 39.4 | 92 |
| | | 0.8 | | 30.4 | 10.4 | 8.7 | 1.6 | | 3.6 | 12.3 | 62.8 | 98.9 |
| | | 1 | | 27 | 6.3 | 6.7 | 0.4 | | 1.3 | 0 | 70.4 | 99.5 |
| | 0.5 | 0 | 99.8 | 99.8 | 99.6 | 94.4 | 82.7 | 99.7 | 99.9 | 99.8 | 81 | 84.6 |
| | | 0.5 | | 100 | 99.5 | 91.4 | 80.1 | | 99.7 | 99.9 | 95.9 | 99.6 |
| | | 0.8 | | 99.7 | 97 | 90.1 | 74.6 | | 99.7 | 99.3 | 96.2 | 99.3 |
| | | 1 | | 99.8 | 55.7 | 91.2 | 61 | | 99.4 | 30 | 96.3 | 99.4 |

Table 7: Results on the PCFG index of occurrence task with 200K< iterations of pre-training.

| $\eta$ | $\mathcal{P}_T(a)$ | $C_{Tr}$ | Acc. PT | Acc. $0_{PT}$ $C_{Te}=0$ 1K | 10K | $C_{Te}=1$ 1K | 10K | Acc. PT | Acc $0_{FT}$ $C_{Te}=0$ 1K | 10K | $C_{Te}=1$ 1K | 10K |
|---|---|---|---|---|---|---|---|---|---|---|---|---|
| $10^{-4}$ | 0.999 | 0 | 99.0 | 5.8 | 0.0 | 23.9 | 0.0 | 9.3 | 71.8 | 99.7 | 46.1 | 99.5 |
| | | 0.5 | | 9.0 | 0.0 | 23.0 | 0.0 | | 66.6 | 99.6 | 81.0 | 100.0 |
| | | 0.8 | | 14.9 | 0.0 | 0.8 | 0.0 | | 36.8 | 99.1 | 99.5 | 100.0 |
| | | 1 | | 33.6 | 9.5 | 0.0 | 0.0 | | 3.7 | 5.8 | 100.0 | 100.0 |
| | 0.9 | 0 | 99.2 | 96.2 | 0.0 | 88.6 | 0.1 | 97.1 | 98.5 | 99.9 | 98.6 | 99.7 |
| | | 0.5 | | 96.8 | 0.0 | 83.5 | 0.0 | | 97.3 | 99.5 | 100.0 | 100.0 |
| | | 0.8 | | 96.8 | 0.0 | 84.8 | 0.0 | | 97.4 | 99.2 | 100.0 | 100.0 |
| | | 1 | | 72.6 | 1.3 | 79.1 | 0.0 | | 37.3 | 24.0 | 100.0 | 100.0 |
| | 0.5 | 0 | 98.0 | 95.7 | 4.5 | 96.5 | 3.6 | 98.9 | 99.5 | 99.7 | 99.8 | 99.8 |
| | | 0.5 | | 95.2 | 5.6 | 78.2 | 3.5 | | 99.1 | 99.8 | 100.0 | 100.0 |
| | | 0.8 | | 96.0 | 17.7 | 90.2 | 5.6 | | 98.8 | 99.4 | 100.0 | 100.0 |
| | | 1 | | 91.3 | 15.8 | 79.8 | 14.5 | | 48.0 | 28.8 | 100.0 | 100.0 |
| $10^{-5}$ | 0.999 | 0 | 99.0 | 94.6 | 2.6 | 86.2 | 18.6 | 9.3 | 17.0 | 85.4 | 16.0 | 68.5 |
| | | 0.5 | | 98.4 | 3.4 | 97.6 | 7.7 | | 14.3 | 79.8 | 27.2 | 97.8 |
| | | 0.8 | | 94.1 | 5.1 | 94.0 | 0.2 | | 10.7 | 66.6 | 37.6 | 99.7 |
| | | 1 | | 77.6 | 27.7 | 70.8 | 0.0 | | 4.9 | 3.7 | 56.6 | 100.0 |
| | 0.9 | 0 | 99.2 | 99.2 | 88.5 | 99.0 | 79.3 | 97.1 | 97.3 | 99.1 | 98.9 | 99.1 |
| | | 0.5 | | 99.2 | 91.9 | 99.2 | 77.9 | | 97.3 | 98.3 | 100.0 | 100.0 |
| | | 0.8 | | 98.8 | 95.6 | 98.9 | 84.3 | | 96.9 | 98.7 | 99.9 | 100.0 |
| | | 1 | | 98.0 | 65.4 | 98.0 | 72.7 | | 81.3 | 31.8 | 99.9 | 100.0 |
| | 0.5 | 0 | 98.0 | 97.3 | 95.8 | 100.0 | 89.1 | 98.9 | 99.1 | 99.9 | 99.9 | 99.4 |
| | | 0.5 | | 97.6 | 95.9 | 98.8 | 75.4 | | 99.2 | 99.3 | 100.0 | 100.0 |
| | | 0.8 | | 97.4 | 95.1 | 97.8 | 80.3 | | 99.6 | 99.1 | 100.0 | 100.0 |
| | | 1 | | 97.4 | 87.7 | 98.0 | 81.0 | | 97.2 | 51.8 | 100.0 | 100.0 |
| $10^{-6}$ | 0.999 | 0 | 99.0 | 99.0 | 80.4 | 99.4 | 73.0 | 9.3 | 14.6 | 23.3 | 19.5 | 15.5 |
| | | 0.5 | | 98.9 | 94.2 | 99.5 | 94.3 | | 13.7 | 20.2 | 22.0 | 40.5 |
| | | 0.8 | | 99.2 | 67.8 | 98.7 | 61.2 | | 13.2 | 9.2 | 25.1 | 71.8 |
| | | 1 | | 99.4 | 61.6 | 98.8 | 11.3 | | 12.9 | 4.5 | 19.3 | 95.6 |
| | 0.9 | 0 | 99.2 | 98.8 | 99.6 | 99.5 | 99.6 | 97.1 | 97.3 | 97.0 | 98.0 | 98.4 |
| | | 0.5 | | 99.0 | 98.8 | 99.8 | 99.2 | | 96.6 | 96.6 | 98.8 | 99.9 |
| | | 0.8 | | 98.7 | 99.2 | 99.7 | 99.0 | | 97.1 | 96.7 | 99.3 | 100.0 |
| | | 1 | | 98.7 | 98.0 | 99.4 | 95.8 | | 97.8 | 70.7 | 99.3 | 100.0 |
| | 0.5 | 0 | 98.0 | 97.8 | 97.9 | 99.8 | 99.6 | 98.9 | 99.2 | 98.9 | 99.2 | 99.7 |
| | | 0.5 | | 98.7 | 98.0 | 99.8 | 98.2 | | 98.9 | 99.0 | 99.8 | 100.0 |
| | | 0.8 | | 97.4 | 97.9 | 99.6 | 98.2 | | 99.5 | 98.9 | 100.0 | 100.0 |
| | | 1 | | 98.1 | 96.7 | 99.9 | 96.4 | | 98.9 | 94.5 | 99.9 | 100.0 |

Table 8: Results on the PCFG index of occurrence task with 50K iterations of pre-training.

| $\eta$ | $\mathcal{P}_{\mathrm{T}}$ (a) | $C_{\mathrm{Tr}}$ | Acc. PT | Acc $0_{\mathrm{PT}}$ $C_{\mathrm{Te}}=0$ 1K | 10K | $C_{\mathrm{Te}}=1$ 1K | 10K | Acc. PT | Acc $0_{\mathrm{FT}}$ $C_{\mathrm{Te}}=0$ 1K | 10K | $C_{\mathrm{Te}}=1$ 1K | 10K |
|---|---|---|---|---|---|---|---|---|---|---|---|---|
| $10^{-4}$ | 0.999 | 0 | 94.2 | 0.5 | 0.0 | 11.7 | 0.1 | 3.2 | 77.1 | 99.6 | 66.1 | 99.4 |
| | | 0.5 | | 3.2 | 0.0 | 2.0 | 0.0 | | 60.0 | 98.5 | 92.5 | 100.0 |
| | | 0.8 | | 6.7 | 0.1 | 1.0 | 0.0 | | 26.7 | 96.8 | 98.7 | 100.0 |
| | | 1 | | 23.1 | 10.9 | 0.0 | 0.0 | | 3.8 | 5.1 | 99.6 | 100.0 |
| | 0.9 | 0 | 94.2 | 43.2 | 0.0 | 34.6 | 0.8 | 69.9 | 86.1 | 99.6 | 75.1 | 97.5 |
| | | 0.5 | | 48.2 | 0.0 | 56.2 | 0.1 | | 81.0 | 98.9 | 97.5 | 99.9 |
| | | 0.8 | | 53.5 | 0.0 | 53.1 | 0.0 | | 74.0 | 97.0 | 98.8 | 100.0 |
| | | 1 | | 12.8 | 11.0 | 32.8 | 0.0 | | 4.4 | 3.4 | 99.8 | 100.0 |
| | 0.5 | 0 | 88.6 | 72.1 | 2.3 | 59.6 | 2.6 | 91.5 | 95.2 | 98.6 | 90.2 | 99.6 |
| | | 0.5 | | 65.4 | 2.3 | 70.4 | 0.0 | | 91.4 | 99.3 | 98.8 | 100.0 |
| | | 0.8 | | 56.6 | 1.5 | 65.5 | 0.0 | | 88.7 | 97.4 | 99.7 | 100.0 |
| | | 1 | | 39.0 | 12.8 | 40.2 | 0.0 | | 6.1 | 5.3 | 99.9 | 100.0 |
| $10^{-5}$ | 0.999 | 0 | 94.2 | 5.0 | 0.1 | 6.4 | 7.7 | 3.2 | 18.4 | 88.9 | 10.3 | 86.5 |
| | | 0.5 | | 74.1 | 1.4 | 42.3 | 0.4 | | 10.1 | 77.0 | 55.2 | 98.8 |
| | | 0.8 | | 37.4 | 3.8 | 13.2 | 0.7 | | 6.9 | 45.9 | 86.9 | 98.7 |
| | | 1 | | 48.0 | 19.1 | 2.7 | 0.0 | | 0.5 | 4.4 | 95.2 | 100.0 |
| | 0.9 | 0 | 94.2 | 89.2 | 18.9 | 69.6 | 26.8 | 69.9 | 74.0 | 91.1 | 66.8 | 82.8 |
| | | 0.5 | | 91.9 | 25.0 | 75.9 | 38.9 | | 70.8 | 86.9 | 82.0 | 99.2 |
| | | 0.8 | | 92.4 | 30.4 | 75.9 | 32.1 | | 60.8 | 80.2 | 87.8 | 99.8 |
| | | 1 | | 69.7 | 8.8 | 73.9 | 8.4 | | 13.8 | 4.4 | 92.6 | 100.0 |
| | 0.5 | 0 | 88.6 | 87.6 | 64.0 | 77.7 | 51.4 | 91.5 | 93.1 | 95.6 | 85.5 | 90.9 |
| | | 0.5 | | 85.9 | 57.5 | 80.0 | 57.0 | | 90.4 | 95.5 | 93.9 | 100.0 |
| | | 0.8 | | 84.9 | 46.8 | 78.2 | 50.4 | | 89.2 | 91.6 | 96.8 | 100.0 |
| | | 1 | | 80.1 | 33.5 | 77.6 | 13.3 | | 38.6 | 5.5 | 98.1 | 100.0 |
| $10^{-6}$ | 0.999 | 0 | 94.2 | 87.5 | 3.7 | 73.8 | 14.2 | 3.2 | 8.3 | 23.0 | 15.8 | 20.7 |
| | | 0.5 | | 95.3 | 49.3 | 83.3 | 19.2 | | 5.4 | 13.7 | 27.5 | 76.9 |
| | | 0.8 | | 94.0 | 31.0 | 71.4 | 7.7 | | 3.9 | 7.2 | 38.7 | 93.7 |
| | | 1 | | 88.1 | 43.0 | 50.6 | 2.1 | | 0.3 | 1.1 | 44.7 | 97.5 |
| | 0.9 | 0 | 94.2 | 93.6 | 88.2 | 73.2 | 70.7 | 69.9 | 69.6 | 75.6 | 56.9 | 66.9 |
| | | 0.5 | | 94.3 | 92.2 | 73.1 | 76.4 | | 70.3 | 69.7 | 67.4 | 88.6 |
| | | 0.8 | | 92.7 | 89.8 | 75.7 | 80.3 | | 65.3 | 63.6 | 71.8 | 94.7 |
| | | 1 | | 95.6 | 56.2 | 72.8 | 72.3 | | 60.2 | 7.8 | 71.9 | 97.9 |
| | 0.5 | 0 | 88.6 | 87.3 | 87.0 | 80.0 | 76.6 | 91.5 | 90.3 | 93.8 | 85.1 | 88.3 |
| | | 0.5 | | 87.7 | 84.5 | 80.4 | 80.8 | | 93.4 | 91.5 | 90.3 | 97.6 |
| | | 0.8 | | 90.6 | 83.2 | 83.1 | 80.5 | | 92.3 | 87.9 | 90.1 | 98.6 |
| | | 1 | | 89.0 | 77.3 | 79.8 | 70.2 | | 92.4 | 23.0 | 91.9 | 99.7 |

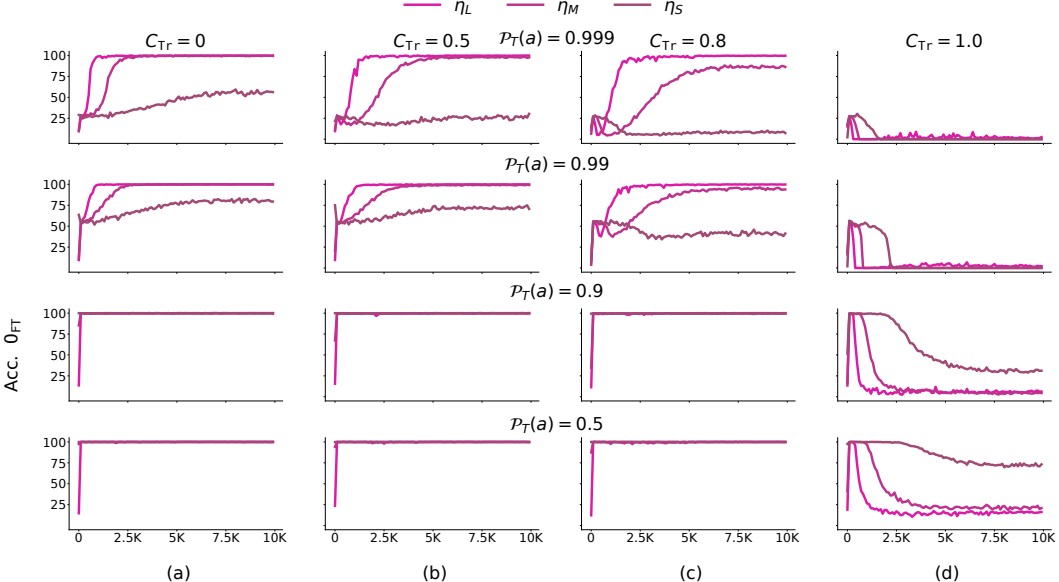

Figure 60: **Counter Task,** $n_{iters} = 200K$, $C_{\texttt{Te}} = 0$: Effect of learning rate (LR) on fine-tuning pre-trained models with weakly and strongly relevant capabilities and using different values of $C_{\texttt{Tr}}$ for fine-tuning. **Observation:** In the presence of strongly relevant capability, training with $\eta_S$ yields good performance on the fine-tuning dataset. The convergence time to learn the fine-tuning task increases with an increase in $C_{\texttt{Tr}}$.

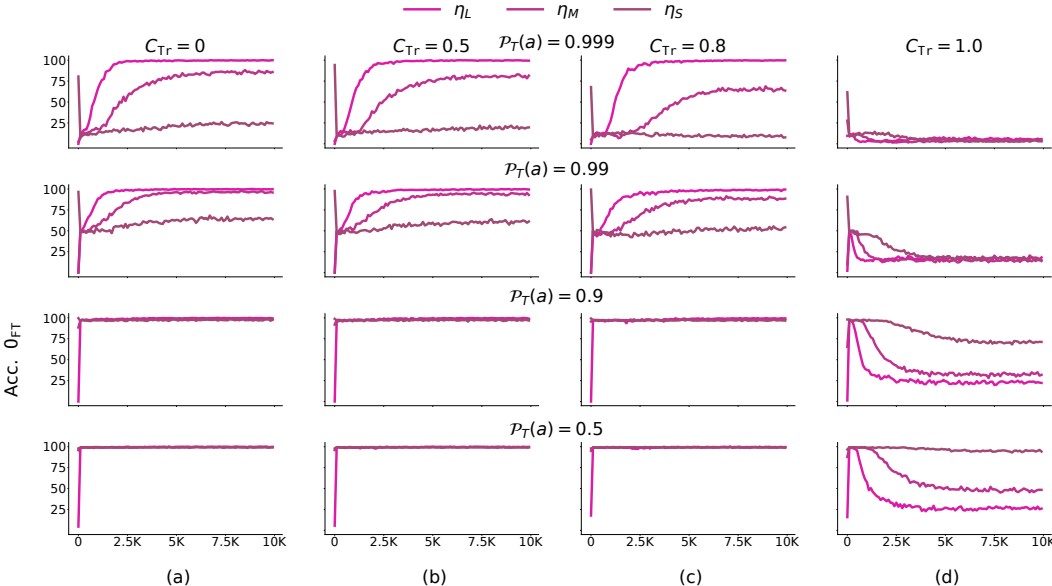

Figure 61: **Index of Occurrence Task,** $n_{iters} = 200K$, $C_{\texttt{Te}} = 0$. The settings are consistent with Fig. 60

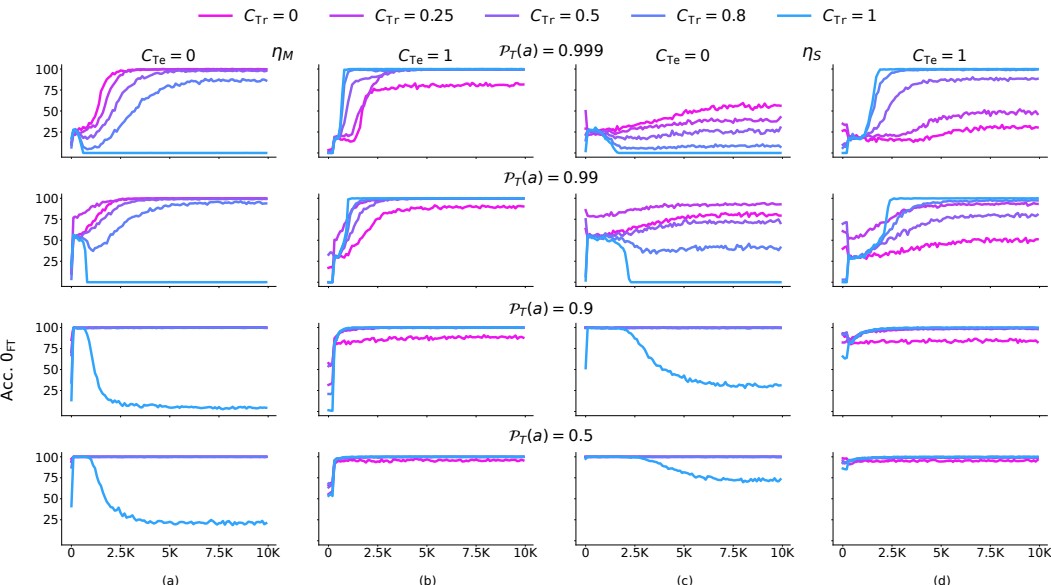

Figure 62: **Counter Task,** $n_{iters} = 200K$**:** Effect of the presence of strongly or weakly relevant pretuning capability on fine-tuning performance on using $\eta_M$ and $\eta_S$. Test sets with and without the spurious correlations are used for evaluation. **Observation:** The convergence time for learning the fine-tuning task in the absence of strongly relevant capability is higher as compared to when the strongly relevant is present in the model. The time further increases if spurious correlations are present in the fine-tuning set. However, in the presence of spurious correlations, the convergence time to learn the spurious correlation is small and is possible even on using the learning rate $\eta_S$. Using $\eta_S$ is unable to yield learning of the fine-tuning task if a weakly relevant capability is present in the model.

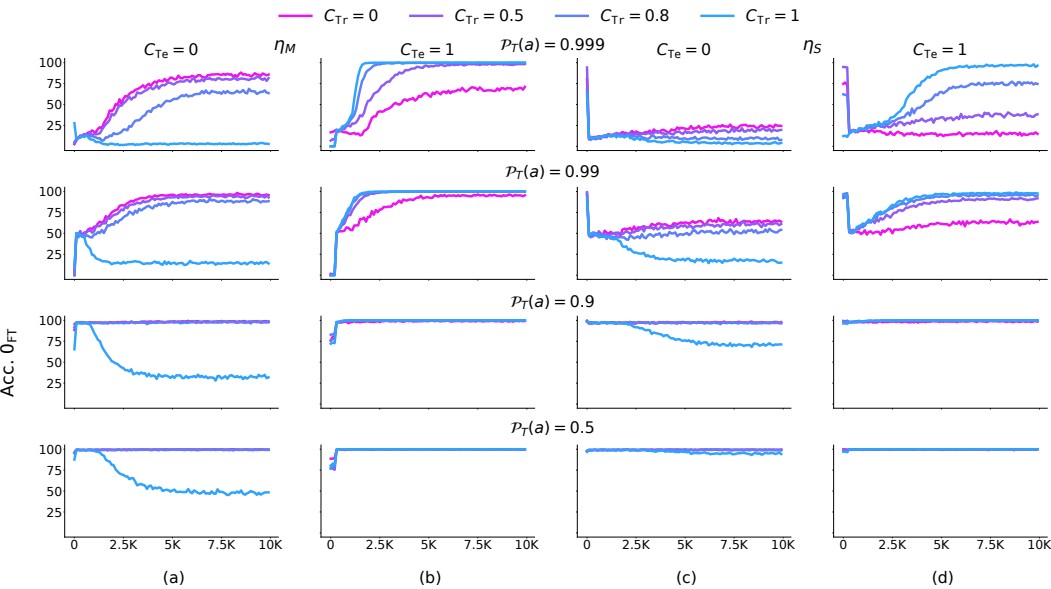

Figure 63: **Index of Occurrence Task,** $n_{iters} = 200K$**:** The settings are consistent with Fig. 62.

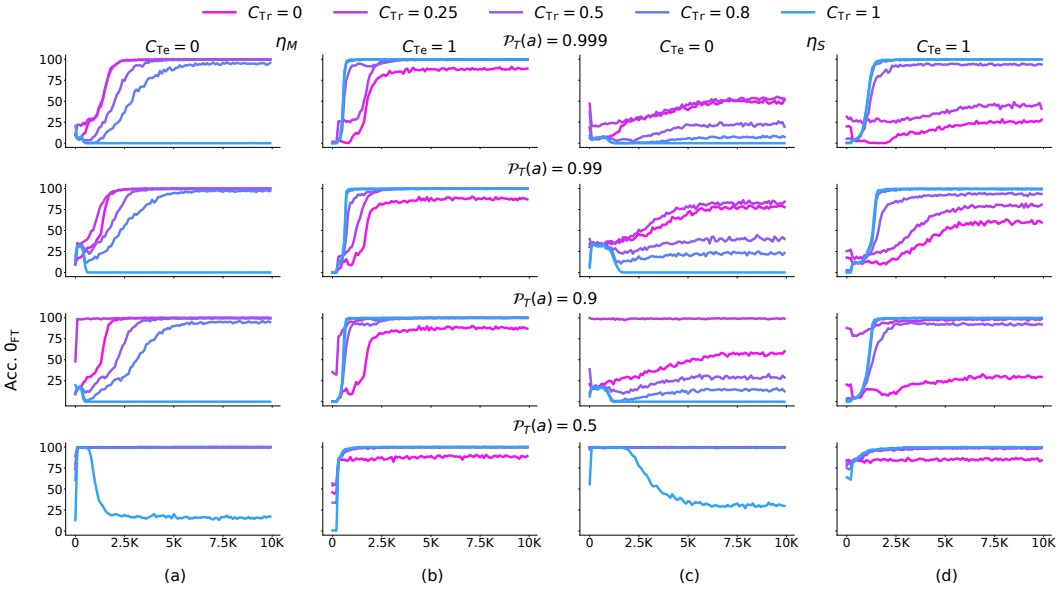

Figure 64: **Counter Task,** $n_{iters} = 50K$**:** Effect of the presence of strongly or weakly relevant pretuning capability on fine-tuning performance on using $\eta_M$ and $\eta_S$. Test sets with and without the spurious correlations are used for evaluation. The observations are consistent with Fig. 62.

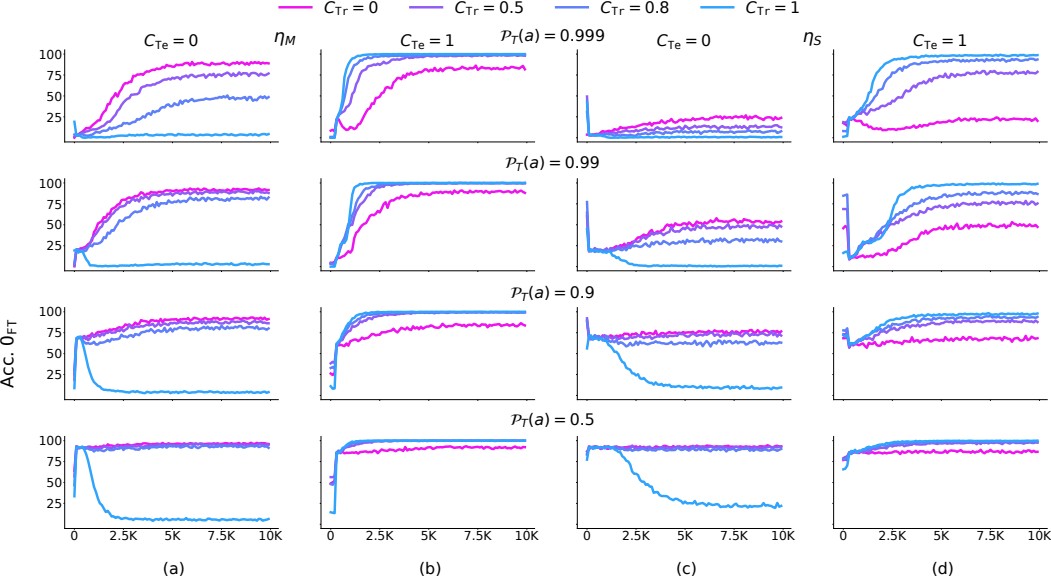

Figure 65: **Index of Occurrence Task,** $n_{iters} = 50K$**:** The settings are consistent with Fig. 64.

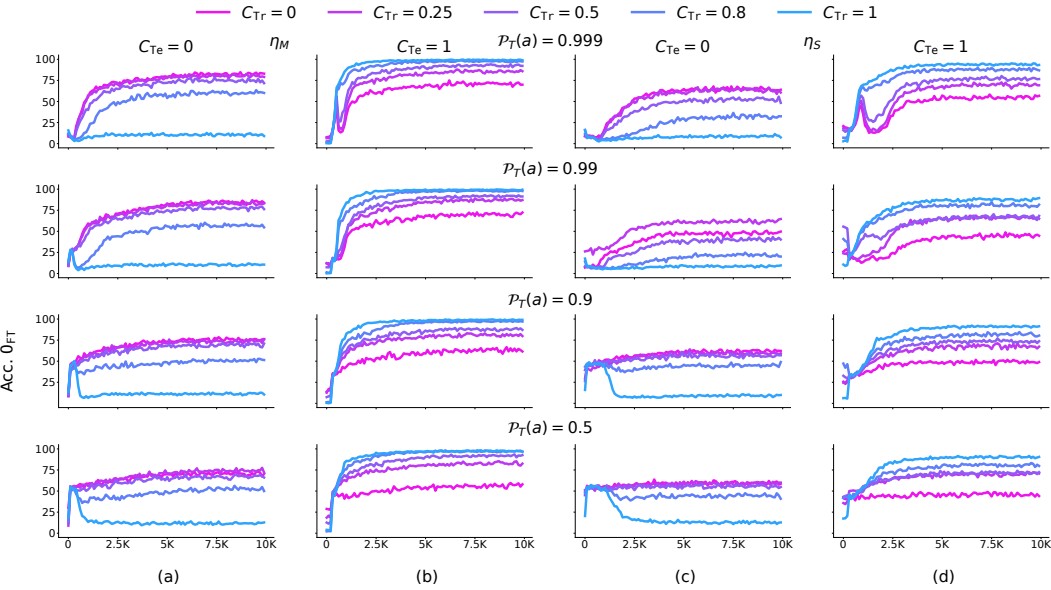

Figure 66: **Counter Task,** $n_{iters} = 10K$: Effect of the presence of strongly or weakly relevant pretuning capability on fine-tuning performance on using $\eta_M$ and $\eta_S$. Test sets with and without the spurious correlations are used for evaluation. The observations are consistent with Fig. 62.

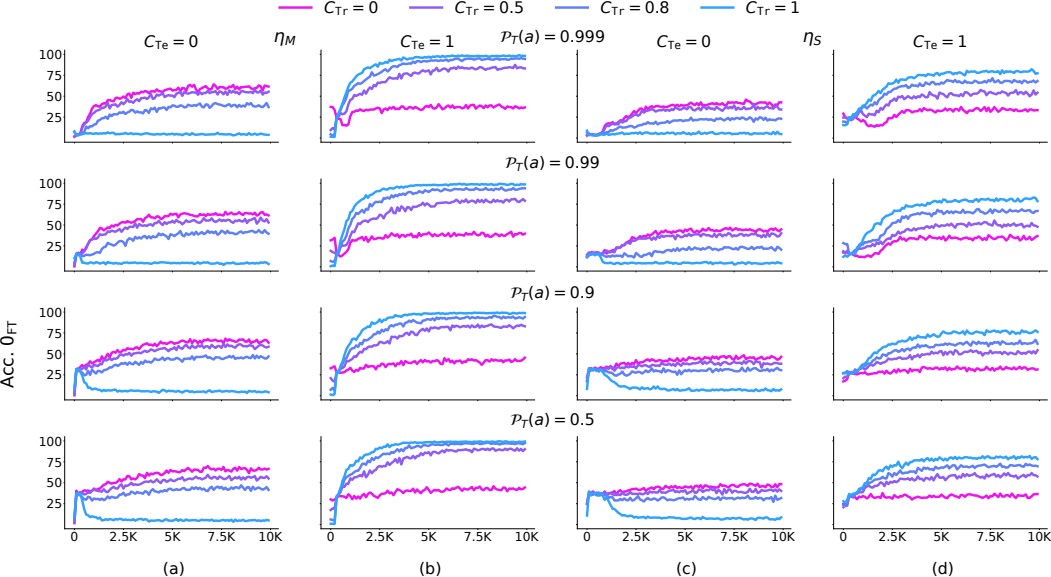

Figure 67: **Index of Occurrence Task,** $n_{iters} = 10K$: The settings are consistent with Fig. 66.

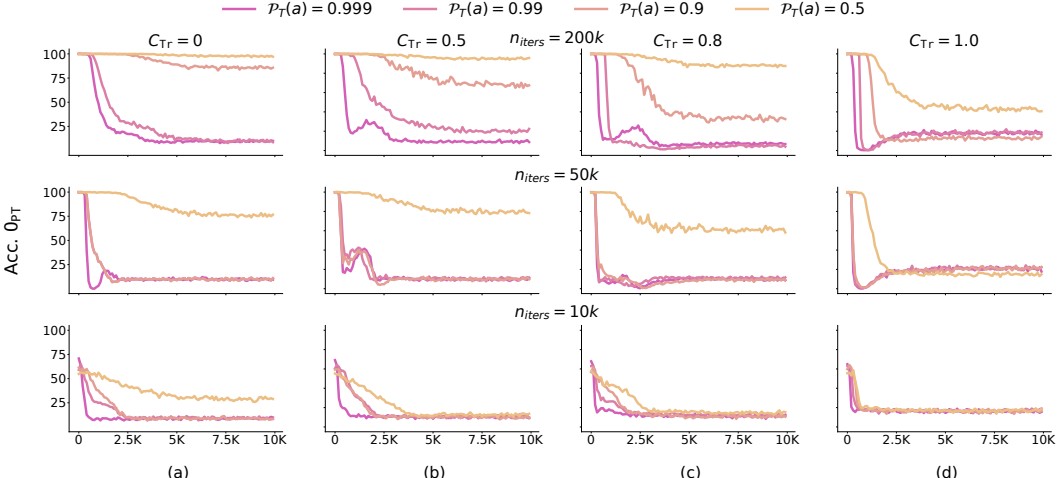

Figure 68: **Counter Task,** $\eta_M$, $C_{\text{Te}} = 0$ **:** Learning of the wrapper in presence of different fraction of spuriously correlated data, values of $\mathcal{P}_{\text{T}}(\text{a})$ during pre-training, and training iterations. **Observation:** Using a higher fraction of spuriously correlated data in the fine-tuning set (higher value of $C_{\text{Tr}}$) leads to faster degradation in the pre-training accuracy. Further this degradation is even faster in presence of weakly relevant capability.

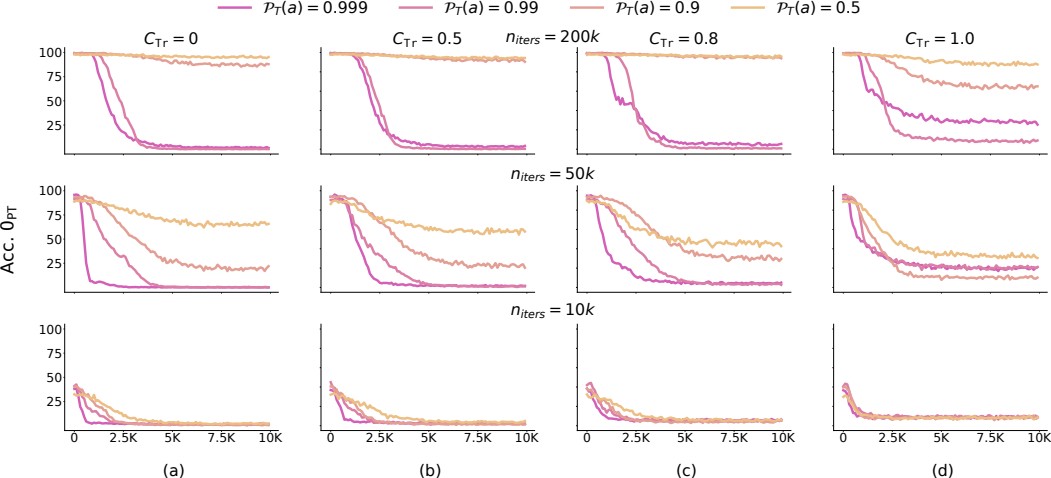

Figure 69: **Index of Occurrence Task, Medium LR,** $C_{\text{Te}} = 0$**:** The settings are consistent with Fig. 68.

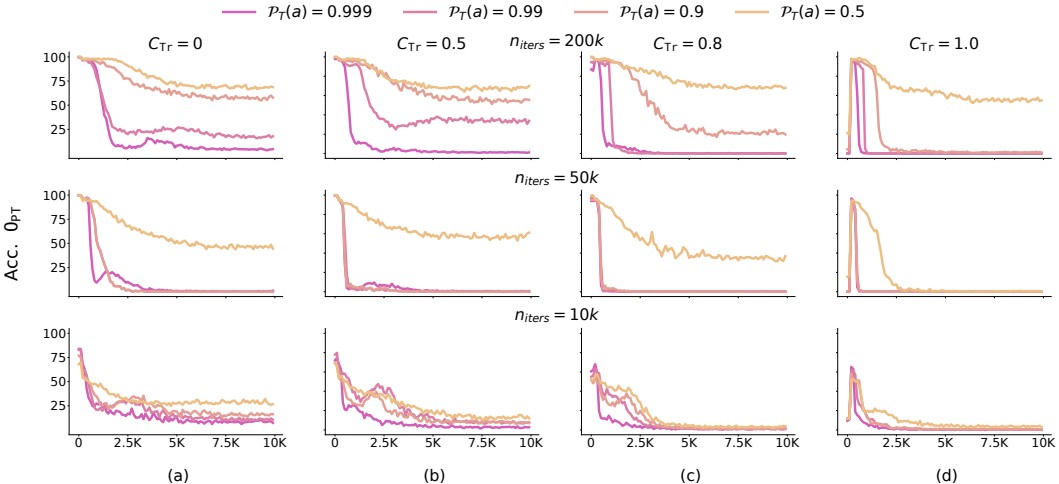

Figure 70: **Counter Task,** $\eta_M$**,** $C_{\text{Te}} = 1$ **:** Learning of the wrapper in presence of different fraction of spuriously correlated data, values of $\mathcal{P}_{\text{T}}(\text{a})$ during pre-training, and training iterations. **Observation:** Using a higher fraction of spuriously correlated data in the fine-tuning set (higher value of $C_{\text{Tr}}$) leads to faster degradation in the pre-training accuracy. Further this degradation is even faster in presence of weakly relevant capability.

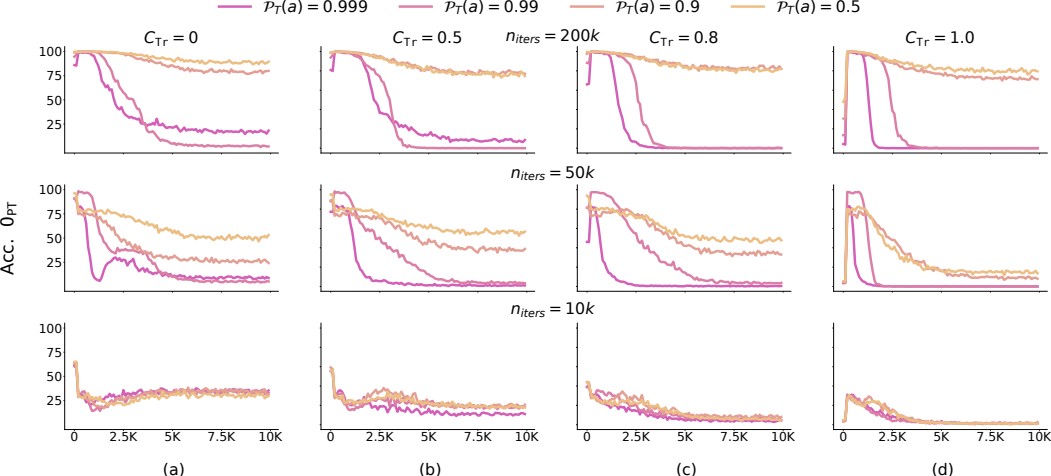

Figure 71: **Index of Occurrence Task, Medium LR,** $C_{\text{Te}} = 1$ **:** The settings are consistent with Fig. 70.

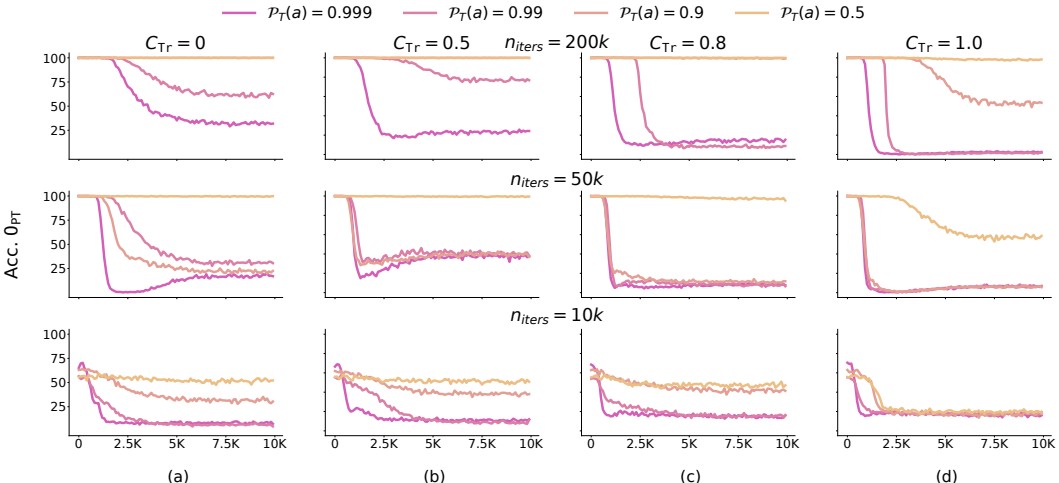

Figure 72: **Counter Task,** $\eta_S$, $C_{\text{Te}} = 0$ **:** Learning of the wrapper in presence of different fraction of spuriously correlated data, values of $\mathcal{P}_{\text{T}}\left(\text{a}\right)$ during pre-training, and training iterations. **Observation:** The observations are consistent with Fig. 68

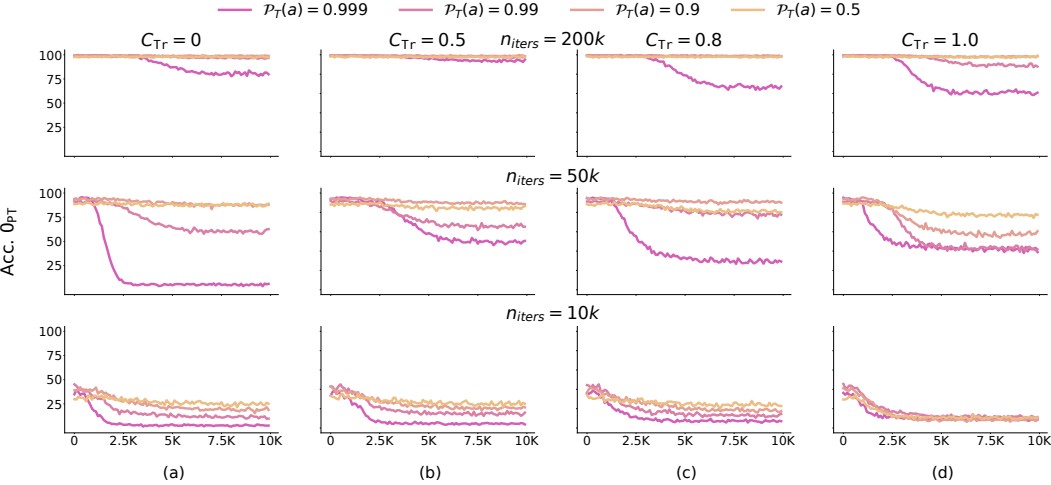

Figure 73: **Index of Occurrence Task,** $\eta_S$, $C_{\text{Te}} = 0$ **:** The settings are consistent with Fig. 72

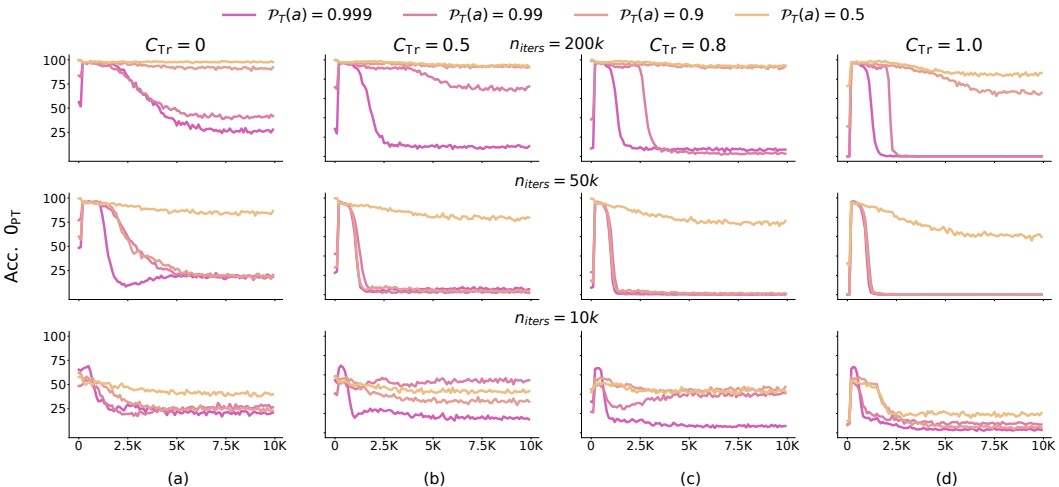

Figure 74: **Counter Task,** $\eta_S$, $C_{\text{Te}} = 1$ **:** Learning of the wrapper in presence of different fraction of spuriously correlated data, values of $\mathcal{P}_{\text{T}}$ (a) during pre-training, and training iterations. **Observation:** The settings are consistent with Fig. 70

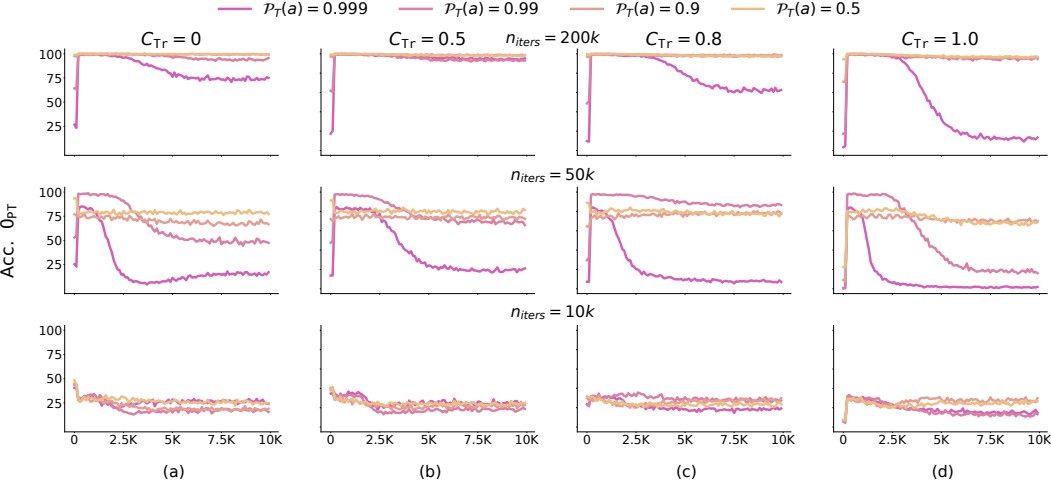

Figure 75: **Index of Occurrence Task,** $\eta_S$, $C_{\text{Te}} = 1$ **:** The settings are consistent with Fig. 74

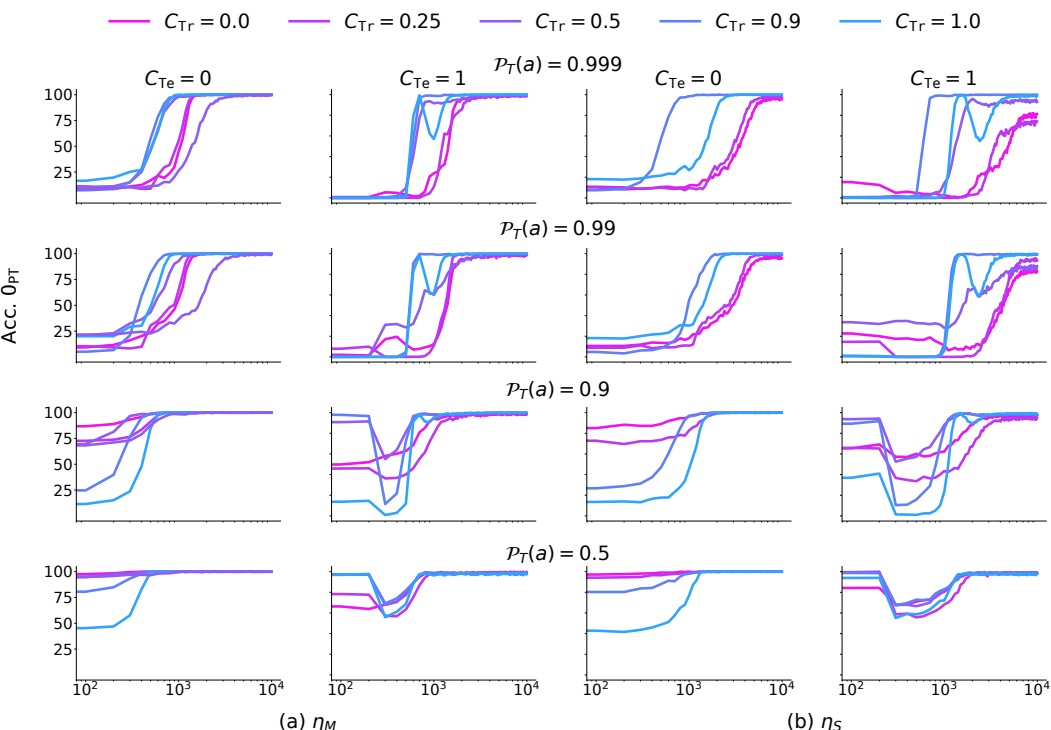

Figure 76: **Counter Task,** $n_{iters} = 200K$**: Reverse Fine-tuning** on weakly and strongly relevant capability fine-tuned models. Medium and small learning rates are used for reverse fine-tuning in the presence of different degrees of spuriously correlated data-points present in the train-set. The fine-tuned model was fine-tuned using Large LR. **Observation:** When the model possesses weakly relevant capability, the convergence time is lower for models fine-tuned on dataset with spurious correlations. If the model possesses strongly relevant capability, this difference is less. The "revival" of pre-training capability is observed for all values of $C_{\text{Tr}}$. Even though fine-tuning was done using a large learning rate of $10^{-4}$, capability revival is possible even on using a small learning rate.

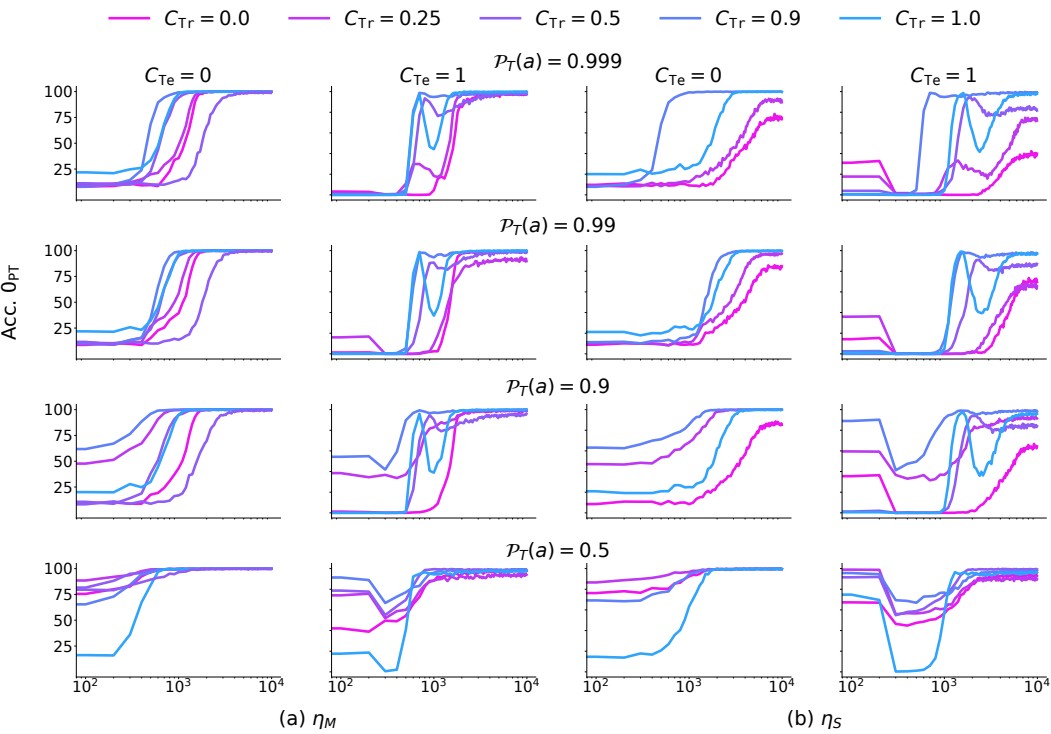

Figure 77: **Counter Task, $n_{iters} = 50K$: Reverse Fine-tuning** on weakly and strongly relevant capability fine-tuned models. $\eta_M$ and $\eta_S$ are used for capability reverse fine-tuning in the presence of different fraction of spuriously correlated data-points present in the train-set. The fine-tuned model was fine-tuned using Large LR. **Observation:** The observations are consistent with Fig. 76

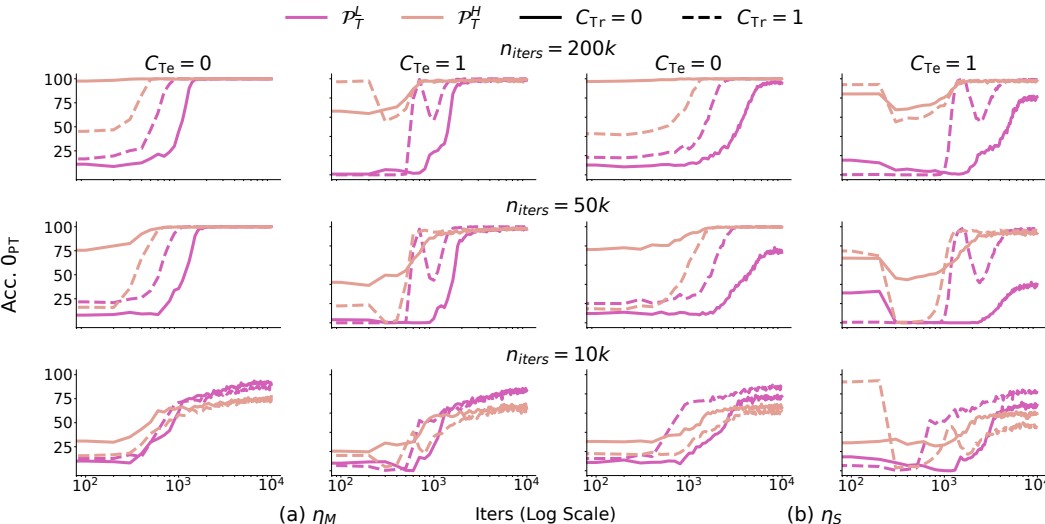

Figure 78: **Counter Task: Reverse fine-tuning analysis for different pre-training iterations. Observation:** Capability Revival is seen for models pre-trained with different number of iterations.

## H.1 PRUNING ANALYSIS

In this section, we present detailed results on pruning analysis of the PCFG setup on both counting and index of occurrence tasks. We provide an exhaustive evaluation in Fig. 79, 82, 84, 86 and 88 for the Counter task and Fig. 80, 83, 85, 87 and 89 for the index of occurrence task.

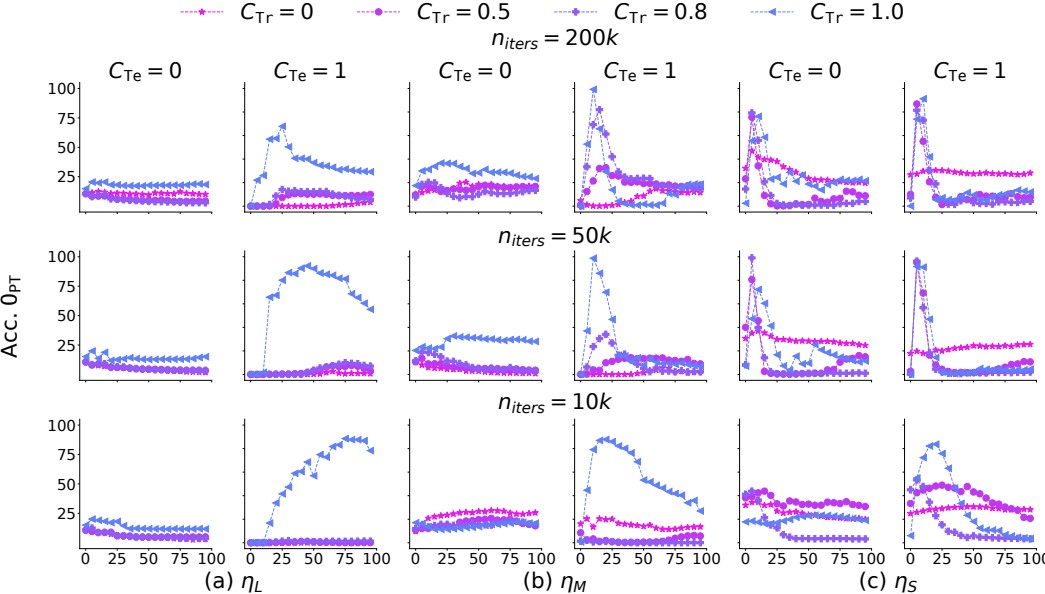

Figure 79: **Counter task,** $\mathcal{P}_{\mathrm{T}}(\mathsf{a}) = 0.999$**, Pruning Analysis:** Revival of pre-training capability analysis for different learning rates, pre-training iterations and different values of $C_{\mathrm{Tr}}$. **Observation:** (a) On using $\eta_L$ the model learns the wrapper only when fine-tuning on $C_{\mathrm{Te}} = 1$. This wrapper is learned only on fine-tuning set with the spurious correlations . (b) On using $\eta_M$ the model learns the wrapper on fine-tuning with smaller values of $C_{\mathrm{Te}}$ as well. However, still this wrapper is learned only on fine-tuning set with the spurious correlations. (c) On using $\eta_S$ the model learns the wrapper for all values of $C_{\mathrm{Te}}$ and for all the data samples.

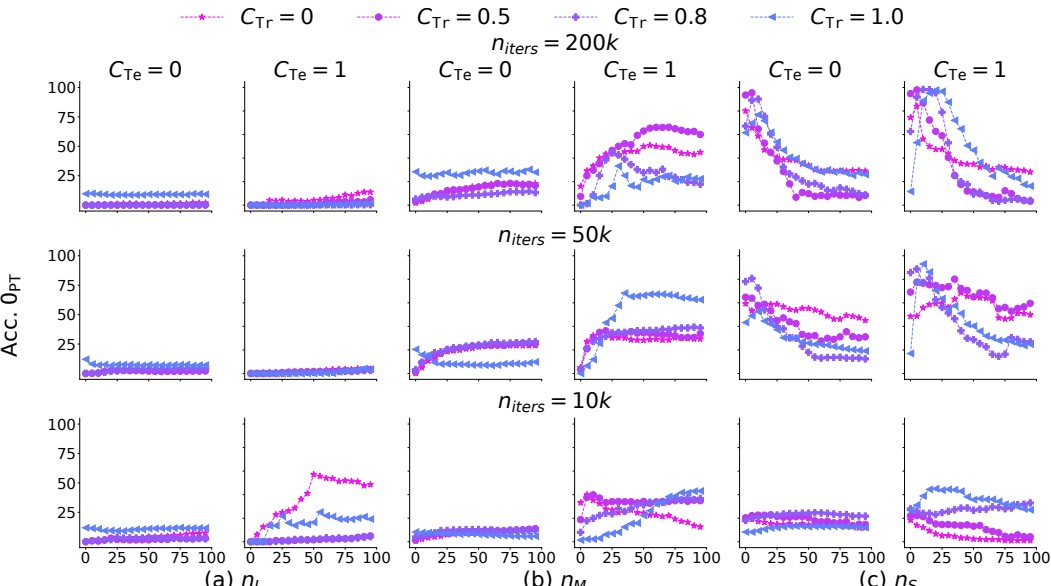

Figure 80: **Index of Occurrence task,** $\mathcal{P}_{\mathrm{T}}(\mathtt{a}) = 0.999$**, Pruning Analysis:** The settings are consistent with Fig. 79

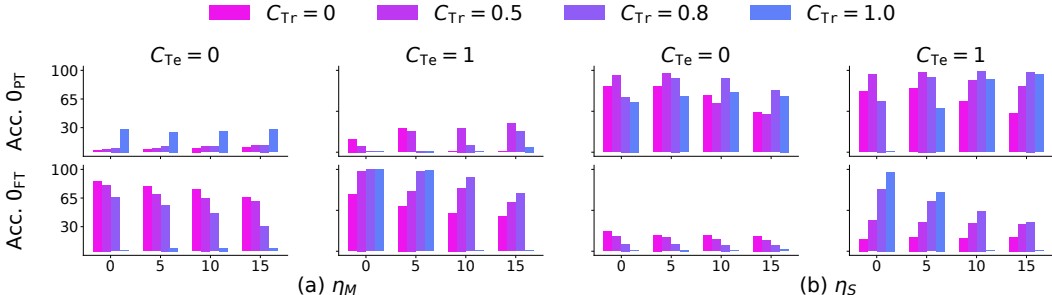

Figure 81: **Index of Occurrence task,** $\mathcal{P}_{\mathrm{T}}(\mathtt{a}) = 0.999$**, Pruning Analysis:** The settings are consistent with Fig. 7

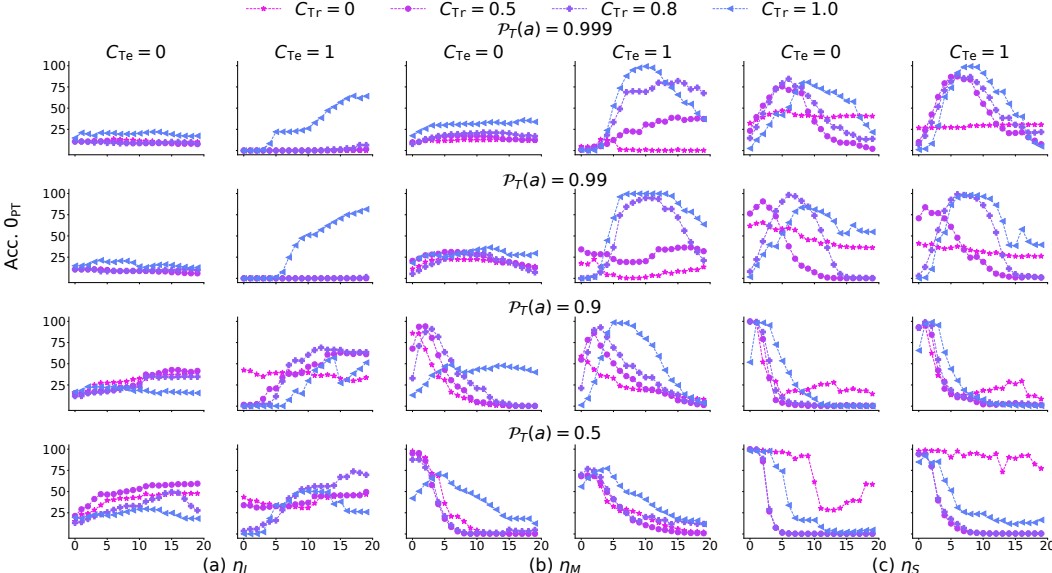

Figure 82: **Counter task,** $n_{iters} = 200K$**, Pruning Analysis:** Revival of pre-training capability analysis for different learning rates, weakly and strongly relevant capability fine-tuned models, and different values of $C_{Tr}$. **Observation:** Learning of the wrapper is possible for weakly as well as strongly relevant capability pre-trained models.

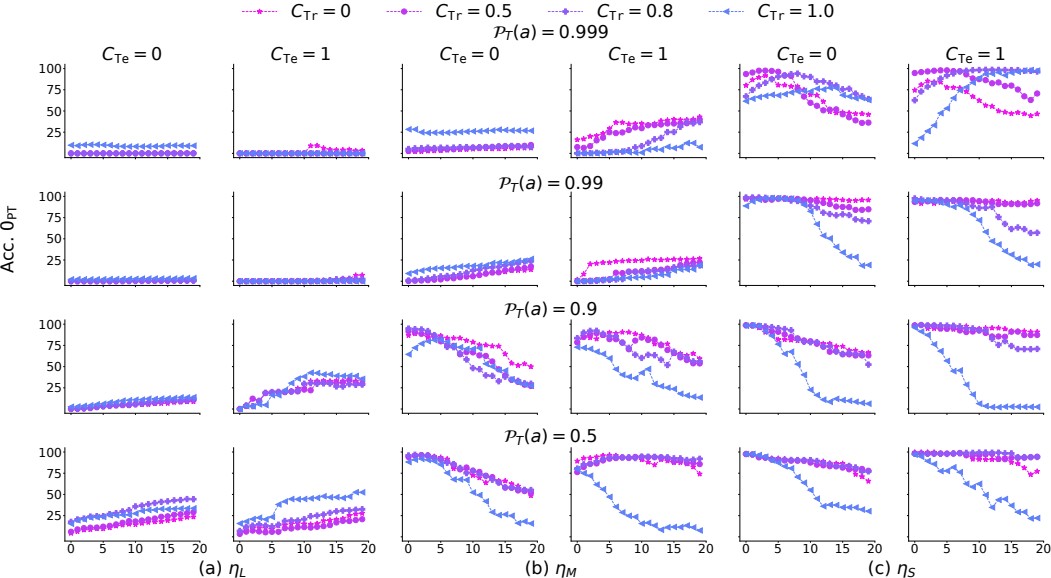

Figure 83: **Index of Occurrence task,** $n_{iters} = 200K$**, Pruning Analysis:** The settings are consistent with Fig. 82

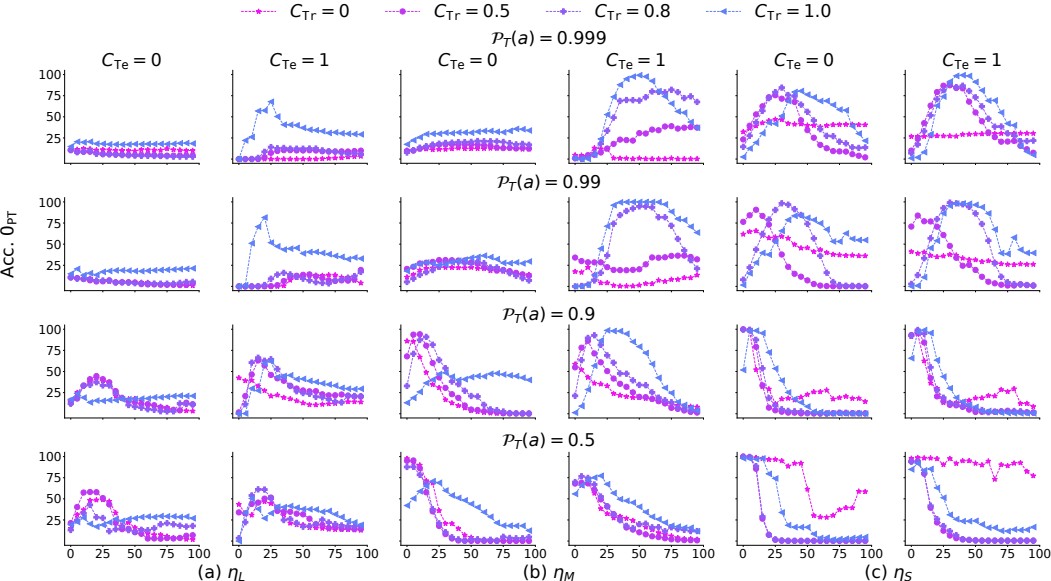

Figure 84: **Counter task,** $n_{iters} = 200K$ **Pruning Analysis:** Revival of pre-training capability analysis for different learning rates, weakly and strongly relevant capability fine-tuned models and different values of $C_{Tr}$. Here larger number of neurons are pruned as compared to Fig. 82.

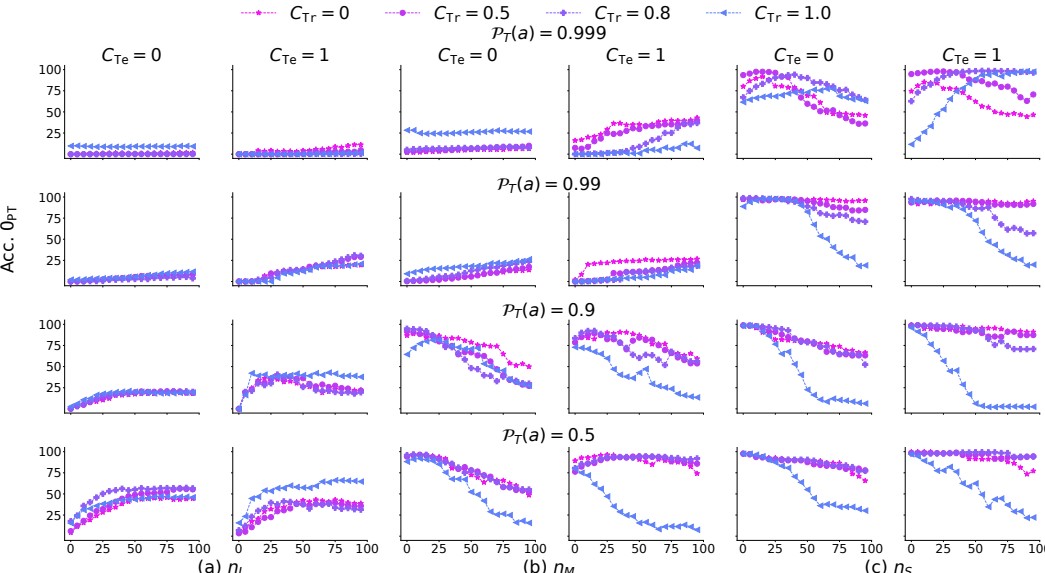

Figure 85: **Index of Occurrence task,** $n_{iters} = 200K$**, Pruning Analysis:** Here larger number of neurons are pruned. The settings are consistent with Fig. 84

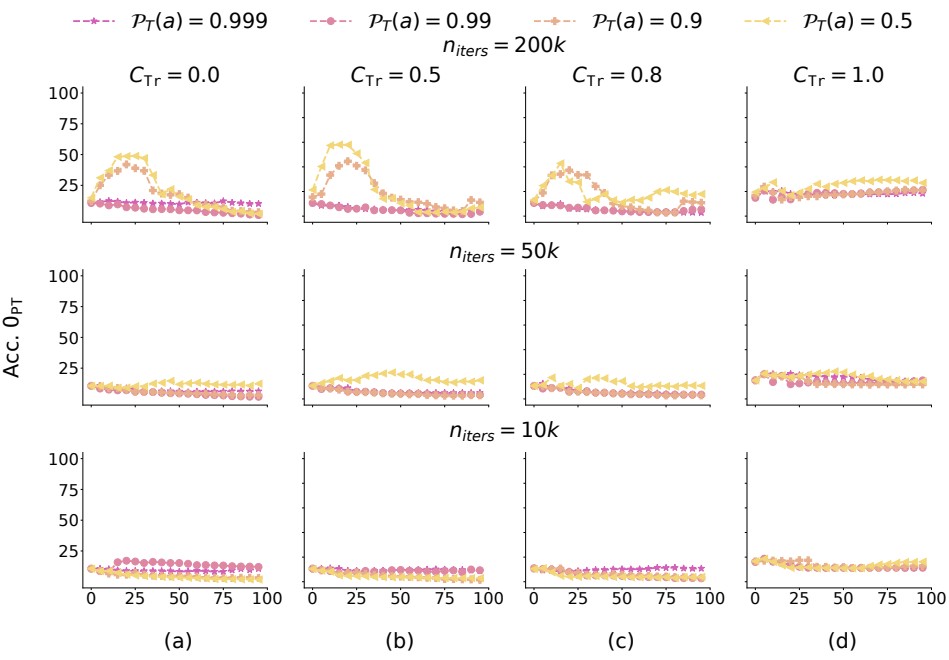

Figure 86: **Counter task,** $\eta_L$, $C_{\text{Te}} = 0$, **Pruning analysis:** Effect of strongly and weakly relevant capabilities for different number of pre-training iterations and different values of $C_{\text{Tr}}$ **Observation:** Fine-tuning a model with strongly relevant capability leads to learning of an "inhibitor" on its pre-training capability, i.e., a wrapper that disallows use of the pretraining capability. Revival of the pre-training capability is partly possible on pruning, if the model has strongly relevant capability and it was fine-tuned on dataset without spurious correlations. The inhibitor is mainly learned for the $200K$ iteration pre-trained model.

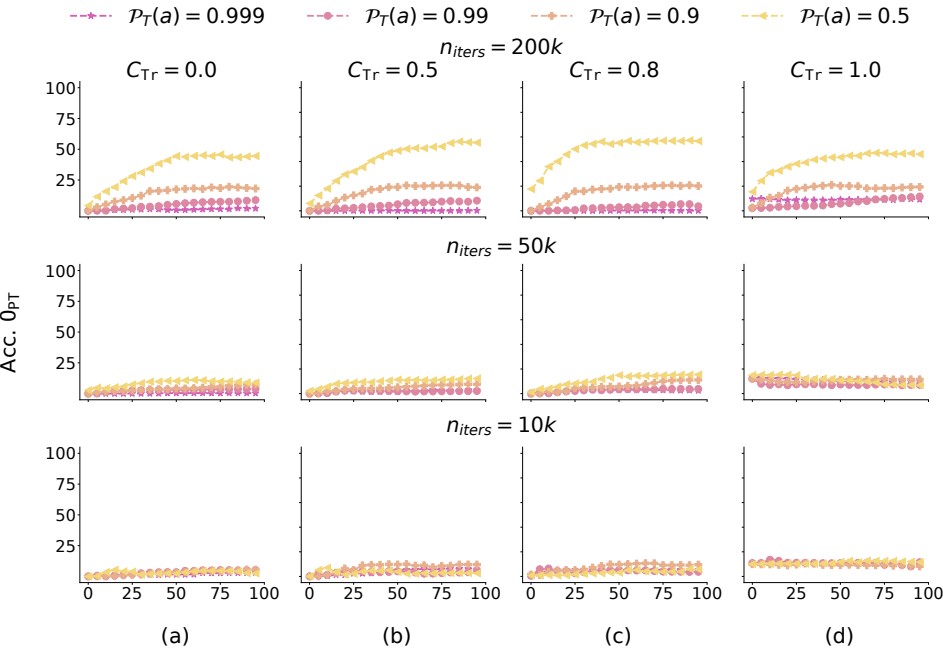

Figure 87: **Index of Occurrence task,** $\eta_L$, $C_{\text{Te}} = 0$, **Pruning analysis:** The settings are consistent with Fig. 86

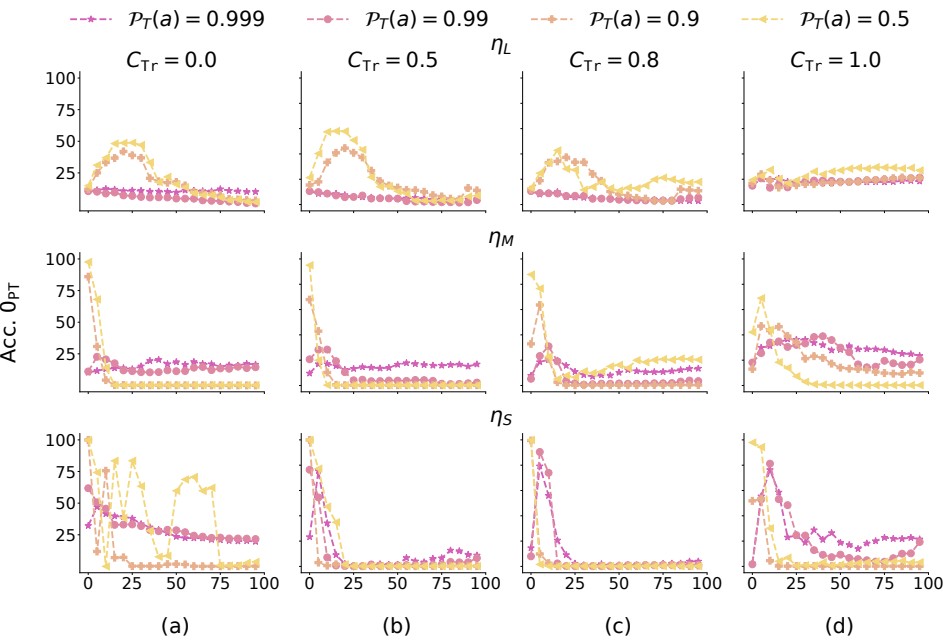

Figure 88: **Counter task,** $n_{iters} = 200K$**,** $C_{\text{Te}} = 0$**, Pruning analysis:** Effect of the strongly and weakly relevant capabilities for different number of pre-training iterations and different values of fraction of spurious correlations present in the fine-tuning dataset. **Observation:** Learning of the inhibitor is observed on using $\eta_L$.

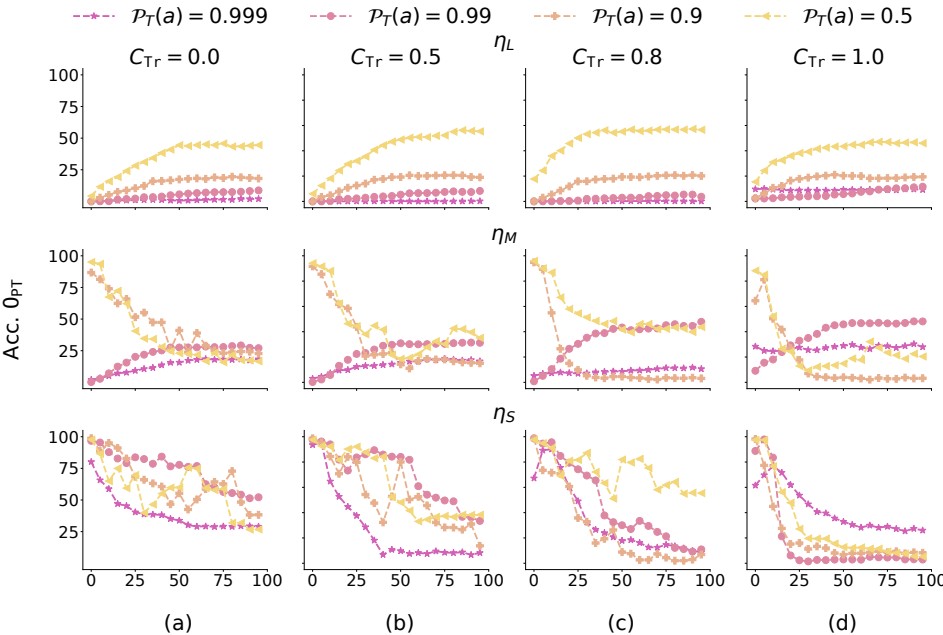

Figure 89: **Index of Occurrence task,** $n_{iters} = 200K$**,** $C_{\text{Te}} = 0$ **Pruning analysis:** the settings are consistent with Fig. 88

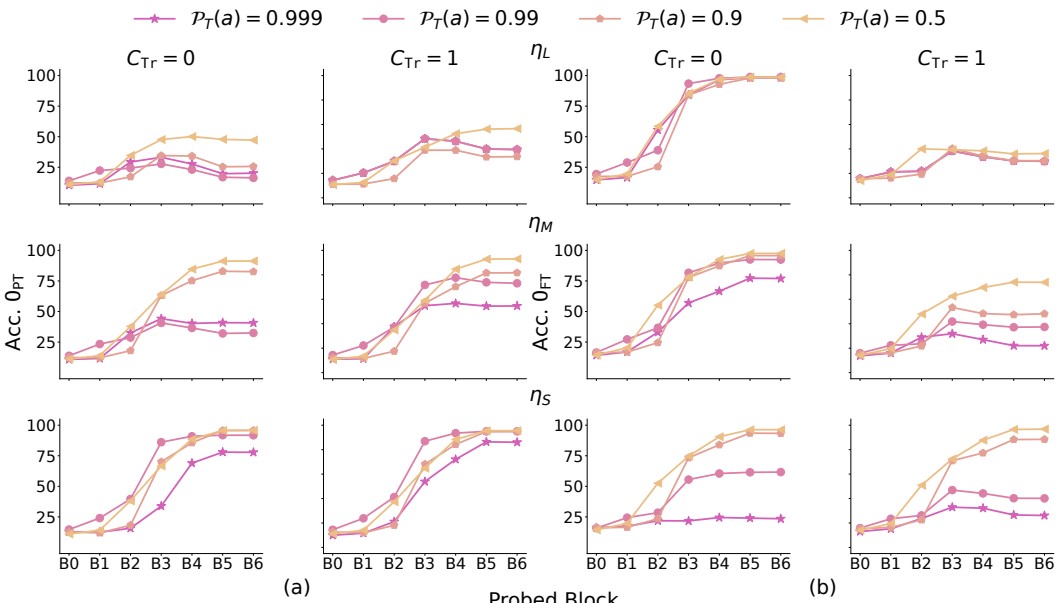

Figure 90: **Counter task,** $n_{iters} = 200K$ **,** $C_{\text{Te}} = 0$**, Probing analysis:** The effect of different values of learning rate, weakly and strongly relevant capabilities is shown. **Observation:** Using $\eta_L$ hampers the pre-training capability to count a especially when the probed model has a weakly relevant capability. The performance on the pre-training task of counting a's continues to be high, especially with $\eta_S$. The accuracy of counting b's shows that fine-tuning capability is learned on using $\eta_L$. On using $\eta_S$, models with weakly relevant capabilities are not able to learn the fine-tuning capability well.

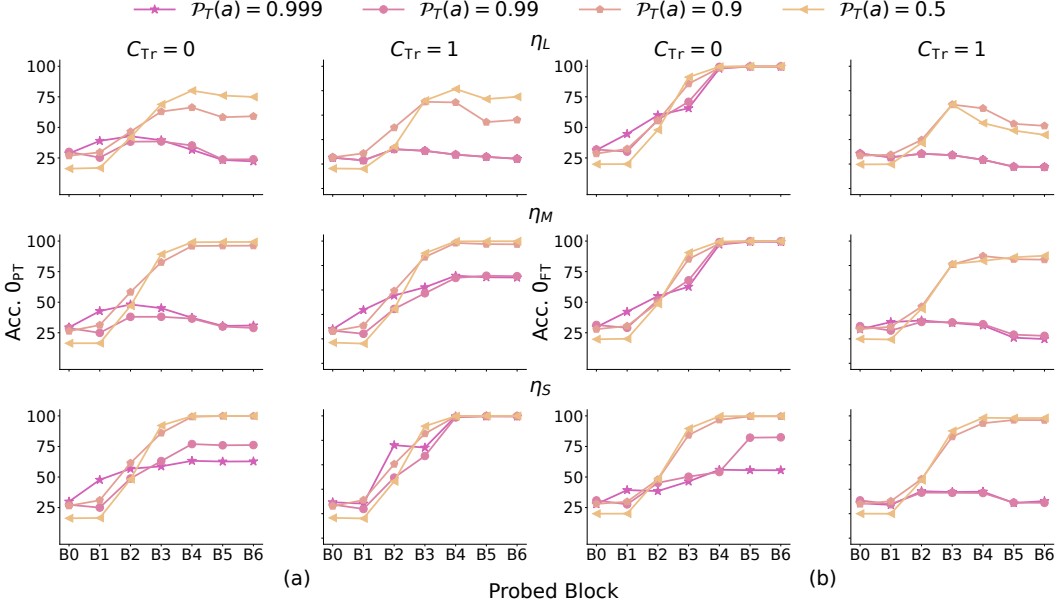

Figure 91: **Index of Occurrence task,** $n_{iters} = 200K$ **,** $C_{\text{Te}} = 0$**, Probing analysis:** The settings are consistent with Fig. 90

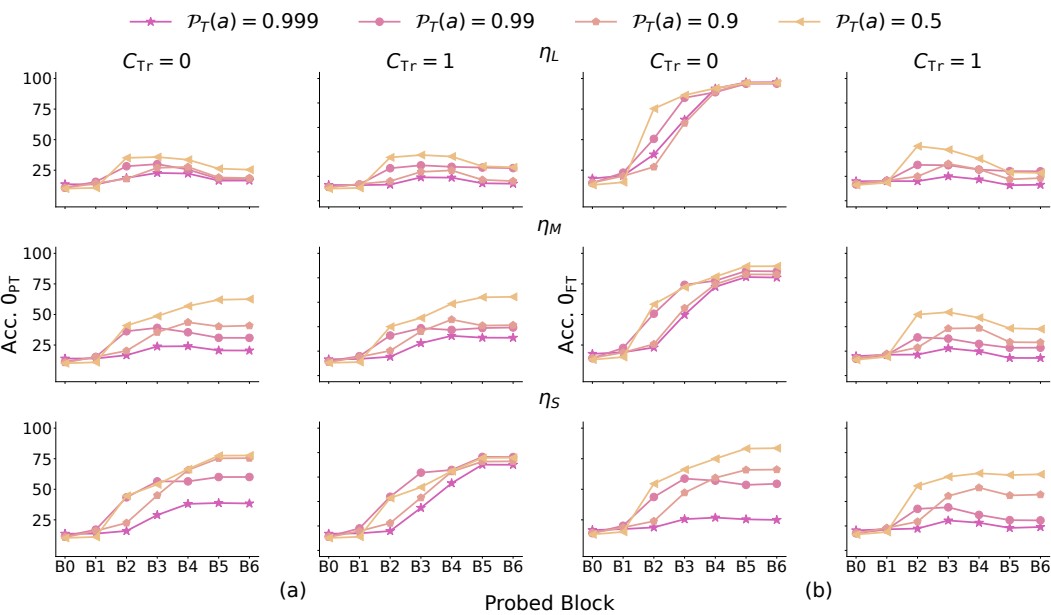

Figure 92: **Counter task,** $n_{iters} = 50K$, $C_{\text{Te}} = 0$, **Probing analysis:** The observations are consistent with Fig. 90

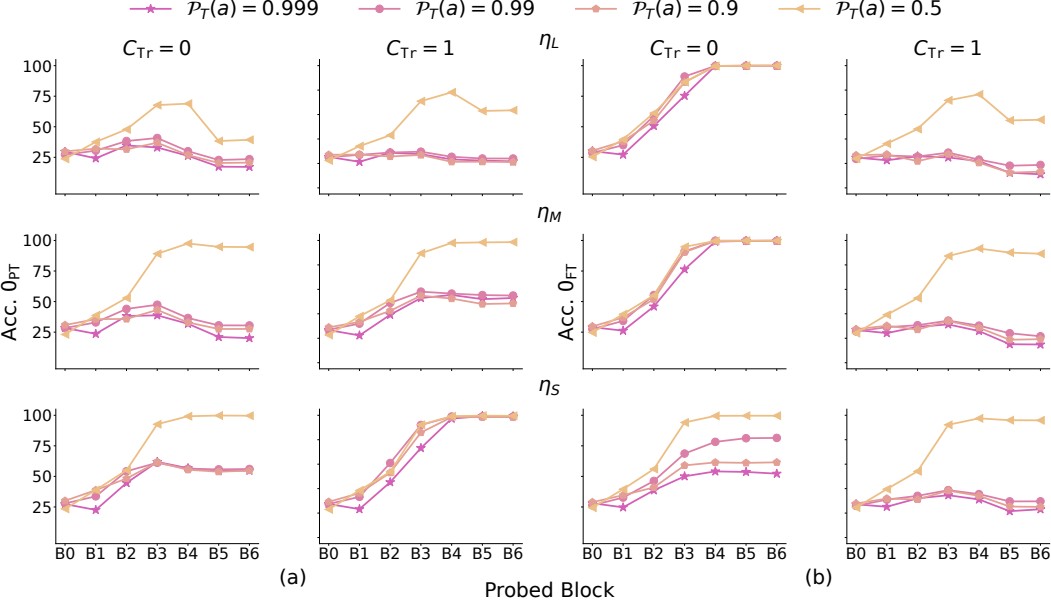

Figure 93: **Index of Occurrence task,** $n_{iters} = 50K$, $C_{\text{Te}} = 0$, **Probing analysis:** The settings are consistent with Fig. 92

## H.2 PROBING ANALYSIS

In this section, we present detailed results on probing analysis of the PCFG setup on both counting and index of occurrence tasks. We provide an exhaustive evaluation in Fig. 92, 95, 97 for the Counter task and Fig. 93, 96, 98 for the index of occurrence task.

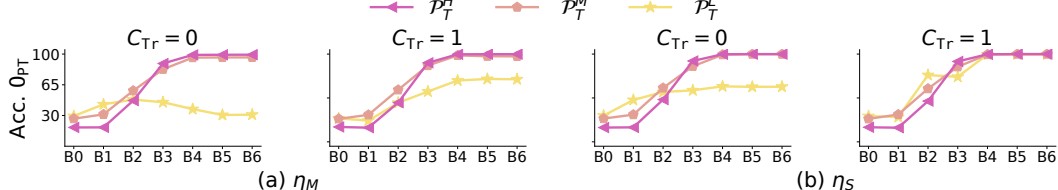

Figure 94: **Index of Occurrence task,** $n_{iters} = 200K$ **,** $C_{\text{Te}} = 0$**, Probing analysis:** The settings are consistent with Fig. 8

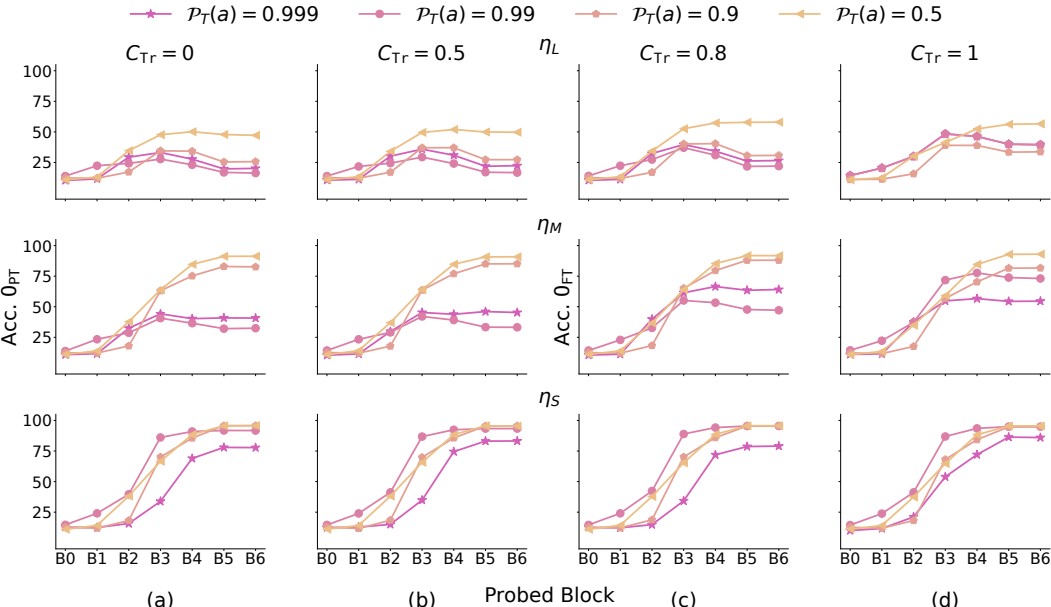

Figure 95: **Counter task,** $n_{iters} = 200K$ **,** $C_{\text{Te}} = 0$**, Probing analysis. Observation:** With an increase in $C_{\text{Tr}}$, the accuracy on counting a's also increases for both weakly as well as strongly relevant capability models.

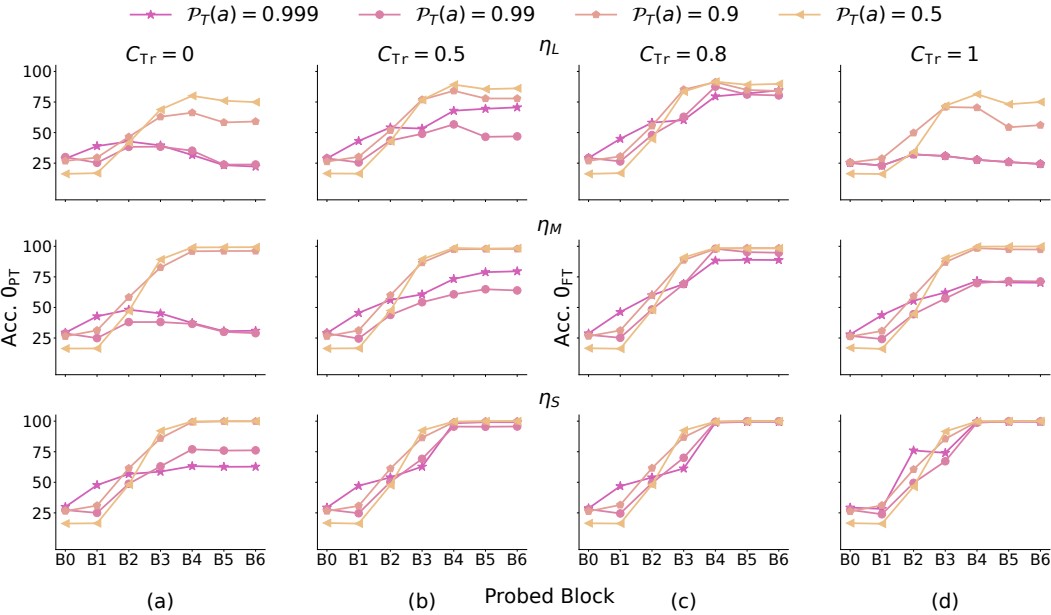

Figure 96: **Index of Occurrence task,** $n_{iters} = 200K$ **,** $C_{\text{Te}} = 0$**, Probing analysis:** The settings are consistent with Fig. 95

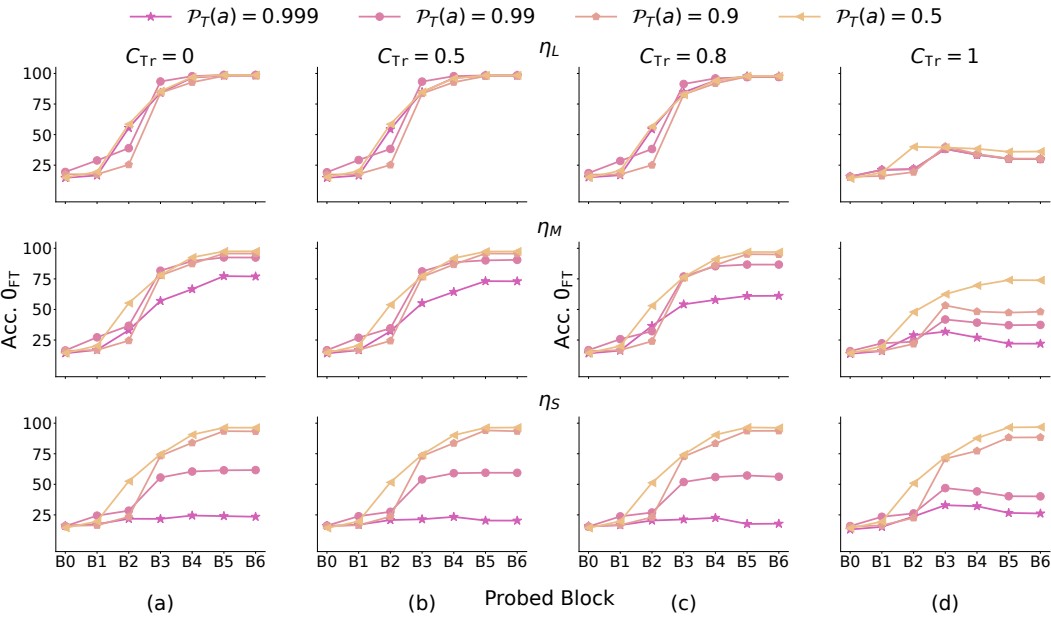

Figure 97: **Counter task,** $n_{iters} = 50K$ **,** $C_{\text{Te}} = 0$**, Probing analysis. Observation:** With an increase in $C_{\text{Tr}}$, the accuracy on counting b's also decreases for both weakly as well as strongly relevant capability models.

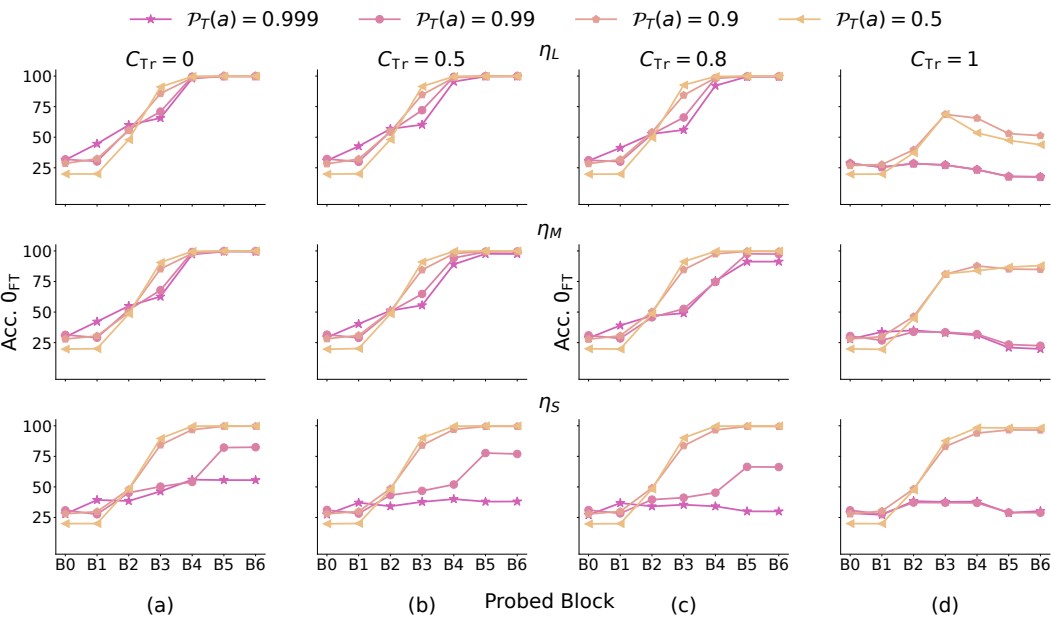

Figure 98: **Index of Occurrence task,** $n_{iters} = 50K$ **,** $C_{\text{Te}} = 0$**, Probing analysis:** The settings are consistent with Fig. 97

