# OpenReview forum: "Mechanistically analyzing the effects of fine-tuning on procedurally defined tasks"
_ICLR.cc/2024/Conference — ICLR 2024 poster_

### Official Review · Reviewer_GZdd · 2023-10-18

**Soundness:** 3 good
**Presentation:** 2 fair
**Contribution:** 3 good
**Rating:** 6
**Confidence:** 4

**Summary:**

The paper analyzed the changes of a pretrained model during fine-tuning, especially how the capabilities are learned or lost during fine-tuning.

**Strengths:**

1. I am really interested in this topic as while fine-tuning is important for unleashing LLM's capabilities learned during pretraining, there is little literature on understanding the dynamics during this process. This is also important for how to mitigate forgetting during LLM fine-tuning.

2. The authors proposed some novel synthetic tasks to probe the evolving of capabilities  during LM fine-tuning.

**Weaknesses:**

1. The presentation of this paper is extremely unclear. I have tried my best to understand the first 5.5 pages with many unclear notations but cannot understand the remaining 2.5 pages. I list some (but not all) unclear points below:
* What are $R_{FT, TR}$ and $R_{FT, TE}$ in Section 4? What do the three elements mean in them?
* What does the characters $L, M, H$ mean in $P_C^L$?
* In Section 4, it seems that $O_{pre}$ has contained $b$, why do you still fine-tune the model on counting $O_{ft}b$?
* "The probability of embedding a spurious correlation in the train/test fine-tuning dataset," This sentence is unclear and has grammatical error.
* What does different lines mean in Figure 5?

Unclear presentations such as listed above largely downgraded the quality of this paper and I cannot fully understand some of the paper's main claims. I hope the author can thoroughly revise their paper on its presentations.

2. Some related works are missing such as [1][2].

[1] Two-stage LLM Fine-tuning with Less Specialization and More Generalization

[2] Data distributional properties drive emergent incontext learning in transformers


------post rebuttal--------

I sincerely thank the authors for clarifications and revisions. The paper looks much better now with the necessary clarifications on the notations.

Overall, I think the paper provides some insightful analysis on the fine-tuning dynamics of transformers. However, the presentation and some of definitions can still be made clearer, for example, in Definition 1, $X_D$ is declared but never used or explained, is it the same thing as $X_C$?

Some definitions and notations should also be made more forma such as the relationship between $P^{FT, E}$ and $P^{FT}$.
I suggest the authors to further revise the paper if it would be accepted.

The reason that I thought the two papers would be related for this work is that they also studied the relationship between pretraining and finetuning, and to some extends validates that the spurious correlation is utilized during fine-tuning. In [1], the model easily "forgets" its capabilities obtained from pretraining as it wrapped these capabilities through spurious correlations. However, these capabilities are not actually lost and can be preserved if we remove this wrapper by removing the soft prompt. In [2], the authors also studies how the model lost its general capabilities during fine-tuning on a single dataset, which can also be related to spurious correlation.

I have increased my score to 6.

**Questions:**

Please see the first point in Weaknesses.

---

> ### Author Response · Authors · 2023-11-21
>
> We thank the reviewer for their feedback and respond to specific comments below.
>
> > The presentation of this paper is extremely unclear. I have tried my best to understand the first 5.5 pages with many unclear notations but cannot understand the remaining 2.5 pages. I list some (but not all) unclear points below:
>
> We sincerely apologize for any confusions caused by the presentation of our submitted draft and emphasize that we have significantly improved the writeup to address your concerns. Amongst several changes, we highlight a few below.
>
> - **Notations.** We have merged any notations in the main text that can be unified under a single variable and have added a table of notations in the appendix for ease of reference for the reader (see Table 2 in Appendix A).
> - **Pseudocodes and examples.** We have provided algorithms specifying in detail how models with different capabilities are constructed in the setup with models compiled via Tracr or trained on PCFGs (see Appendix B). We have also added examples of samples drawn from the tasks studied in our work in the appendix.
> - **Refocus on a single task's description.** We have reduced our focus to a single task in the main paper to enable better focus on the main results. Results on TinyStories have also been provided in more detail now (see Section 5.1)
>
> Overall, we thank the reviewer for this comment that has prompted us to improve the presentation of our work. *We would be deeply grateful if the reviewer can take a look at Section 5 again.*
>
> -------
>
> > What are $R_{FT,TR}$ and $R_{FT,TE}$ in Section 4? What do the three elements mean in them?
>
> Thank you for the question. We use these variables to model the probability that a spurious correlation is added to a randomly sampled datapoint in the train (Tr) or test (Te) subset of the fine-tuning dataset. Thus, $R_{FT, Tr}$ denotes the expected proportion of fine-tuning (FT) training (Tr) samples that include a spurious correlation. For example, if $R_{FT, Tr}=1$, all these samples would have a spurious correlation. *We note that in improving our work’s presentation, we have simplified these notations to $C_{Tr}$ and $C_{Te}$ now.*
>
> -------
>
> > What do the characters $L, M, H$ mean in $P_{T}^{L}$?
>
> The superscript in these variables denotes whether the probability of sampling the target token for fine-tuning tasks is low (L), medium (M), or high (H) in the pretraining data. If this probability is low (e.g., 0.001), the model will not learn a relevant capability that can be exploited for the downstream task; however, if the probability is sufficiently high (e.g., 0.5), the model will essentially learn capabilities strongly relevant to the downstream task in the fine-tuning process. Overall, these sampling priors allow us to control the extent to which the pretraining capabilities are relevant for fine-tuning. We do emphasize that we discussed these notations in the paper in Section 4.1, paragraph 2; however, we acknowledge that their presentation could be improved---*we have accordingly made changes to introduce them earlier in Section 4, repeat them later in the section in more detail, and also summarize in Table 2 (summary of all notations) in Appendix A.* We hope this helps address reviewer’s concerns.
>
> -------
>
> > In Section 4, it seems that $O_{pre}$ has contained $O_{FT}=b$, why do you still fine-tune the model on counting $b$?
>
> To model the notion of a strongly or weakly relevant capability during fine-tuning in our PCFG setup, we include samples where the task is to be performed on the target token for fine-tuning ($O_{FT}$) in the pretraining data itself. When the probability such samples are seen during pretraining is sufficiently high, we get a model that has a capability relevant to the downstream task.
>
> -------
>
> > "The probability of embedding a spurious correlation in the train/test fine-tuning dataset," This sentence is unclear and has grammatical error.
>
> We thank the reviewer for bringing this to our attention! The sentence was supposed to end in a period and has been rectified in the updated manuscript now. The updated sentence reads: “The probability a datapoint sampled from the train or test fine-tuning dataset contains a spurious correlation is denoted $C_{Tr}$ and $C_{Te}$, respectively.”
>
> -------
>
> > What does different lines mean in Figure 5?
>
> We sincerely apologize for the confusion! The figure was missing a legend and has been fixed now; please see the updated figure (now Figure 6). In brief, *different colored lines in the plot denote different sampling prior* of token $O_{FT} = b$ in the pretraining data.
>
> -------

---

> > ### Author Response · Authors · 2023-11-21
> >
> > > Some related works are missing such as [1][2]. [1] Two-stage LLM Fine-tuning with Less Specialization and More Generalization [2] Data distributional properties drive emergent incontext learning in transformers
> >
> > We thank the reviewers for bringing these works to our attention. While we were aware of them, we were unsure of their relevance to the goal of our paper. For example, reference [1] focuses on the negative impact of fine-tuning on in-context learning capabilities of an LLM, developing a protocol for mitigating this negative impact. Meanwhile, reference [2] focuses on developing a better understanding of in-context learning in transformers. In contrast, we emphasize our work focuses on mechanistically assessing how the pretraining capabilities of a model are altered by fine-tuning on a downstream task where the model’s pretraining capabilities may or may not be relevant. *Given the reviewer’s comment though, we have added citations to these references.*
> >
> > -------
> >
> > **Summary:** We thank the reviewer for their comments and hope our revisions to the writeup and the added references address their concerns. We hope that they can consider increasing their score to support the acceptance of our work.

---

### Official Review · Reviewer_wEP2 · 2023-10-31

**Soundness:** 2 fair
**Presentation:** 2 fair
**Contribution:** 3 good
**Rating:** 6
**Confidence:** 4

**Summary:**

This work explores how fine-tuning impacts the capabilities of large pre-trained models. It uses mechanistic interpretability tools and the technique of reverse fine-tuning to study how fine-tuning affects these capabilities and finds that fine-tuning often adds a minimal transformation, called a "wrapper," on top of the existing capabilities when the learning rate is sufficiently small. They demonstrate this in a synthetic setting with PCFG's initialized with Tracr solutions and in a more realistic setting with the TinyStories dataset.

**Strengths:**

1. The authors address the critical problem of understanding how fine-tuning impacts models, and specifically try to offer mechanistic insight via a compelling section of techniques (probing, pruning, reverse fine-tuning).
2. I like the TinyStories experiment, where the authors find that the pretraining capability can be recovered via fine-tuning uniquely when the learning rate is small enough, as evidenced by the control model. Importantly, I find it really interesting how in the high learning rate deletion setting, the capability loss for fine-tuning the control and deleted model match each other. I best understood this from Figure 79 on page 53 in the Appendix, which might benefit from being in the main paper.
3. The authors are incredibly comprehensive over hyperparameters, giving many alternatives in the appendix.

**Weaknesses:**

My overall analysis is that the authors haven't provided a clear definition of what it means for a capability to have been lost or recoverable

1. [Experiment 2] I believe the wrapper formalization is inconsistently used throughout the paper. Starting at Definition 2, a wrapper $g$ is composed with $\mathcal{C}$ and is defined as a map acting on the output of a capability ($g$ is unquantified here, and I believe the range should be the domain here). This definition is independent of a model and how it is parameterized. However, in the experiments, the wrapper definition is always a subset of the weights to be pruned. I believe this is not captured as a function of the capability/model output, but rather the function parameterization. As such, probing can not be used as evidence that the model learns a wrapper over the capability.
    - This is not simply fixed by altering the definition to be a change over weights; there is always the wrapper of adding the optimal FT weights and subtracting the current PT weights, which would produce a strongly relevant capability but (I think) would go against the spirit of a wrapper. I believe it is non-trivial to find a definition of wrapper that is 1) covered by pruning and 2) captures the spirit of being a lightweight modification of the model.
    - Is there an example of a fine-tuning capability that can not be obtained by a wrapper under the current or a revised definition? Having this delineation is important to proving something that isn't true by construction.

2. [Experiment 3] In the abstract and introduction, revival via reverse fine-tuning is discussed as being sample-efficient. Experiment 3 is intended to demonstrate this phenomenon experimentally. However, there is no discussion of sample efficiency here or later in the paper. Specifically, to show that the pretrained capability is actually forgotten, there would need to be a comparison to how much time it takes to learn the pretrained capability, which doesn't exist since the capabilities are directly compiled via Tracr. Even this experiment would be confounded by the fact that the fine-tuned model can be seen as simply a good initialization for learning the pretrained capability from scratch. This sample efficiency analysis is critical since in theory, any capability can be learnt by training for long enough.

3. [Experiment 4] The linked Figure 8 for the probing analysis does not concern probing. Does this mean to refer to Figure 5? If so, what new information does this section provide that is not captured in Experiment 2, which refers to Figure 5? The analysis and takeaway in this section do not seem grounded in data in the current state.

4. I found the results in this paper quite difficult to parse. None of these items individually changed my score, but it did make it much harder/time-consuming to parse the message of the results.
    - In Figure 4, (i) through (iv) do not agree on the figure vs the text.
    - Figure 5 does not have a legend and I do not know what it is measuring.
    - Figure 6 does not have an x-axis and I do not know what it is measuring.
    - The plots are out of order, for example, Fig 7 is referenced before Fig 5. This led to some confusion while reading experiments.
    - Plots such as Figures 6, 7, and 8 test for three controlled variables at once and take a lot of effort to parse.
    - Is there a reason the chosen $P(O_{PT})$ is changed from Figure 7b to Figure 8b?
    - The reader is required to parse and remember a lot of notation consistently across the paper. It would be helpful to be more verbose about the notation definitions, especially in plot titles and captions.

**Questions:**

All questions are addressed in weaknesses.

---

> ### Author Response · Authors · 2023-11-21
>
> We thank the reviewer for their positive comments on our paper’s motivations, extensivity of our experiments, and the mechanistic insights offered by our results on the impact of fine-tuning on a model’s pretraining capabilities! We respond to specific comments below.
>
> -------
>
> > I like the TinyStories experiment, where the authors find that the pretraining capability can be recovered via fine-tuning uniquely when the learning rate is small enough, as evidenced by the control model. Importantly, I find it really interesting how in the high learning rate deletion setting, the capability loss for fine-tuning the control and deleted model match each other. I best understood this from Figure 79 on page 53 in the Appendix, which might benefit from being in the main paper.
>
> Thank you for this suggestion! We have pulled Figure 79 to the main paper now (see Figure 11 (right)). To augment this figure, we have also added the following results to the main paper now.
>
> 1. **Probing TinyStories models:** In Figure 11 (left), we present a probing analysis of models fine-tuned on the TinyStories dataset via different fine-tuning protocols to delete the capability to produce stories with a certain feature (specifically, Twists). *We find that the model continues to represent the Twists feature in its intermediate outputs*, despite being actively fine-tuned to not produce it.
>
> 2. **Generation scores during reverse fine-tuning:** In Table 1, we report the percentage of stories with the Twist feature generated by a model undergoing reverse fine-tuning. To develop this result, we fine-tuned a GPT-3.5 model via the OpenAI API to infer whether a story contains a certain feature from the possible set of features in TinyStories, including Twists. We then run stories generated by our models undergoing reverse fine-tuning through this GPT-3.5 model. As shown in Table 1, *we see that often 30–300 gradient steps are sufficient for the model to again start producing stories with Twists during reverse fine-tuning.*
>
> *We also highlight that we have significantly expanded the results on TinyStories to thoroughly corroborate our claims in a more realistic setting (see Appendix F).* Specifically, we now include results on probing for different features and dynamics of loss and generation scores during reverse fine-tuning in several different setups. All results show that the model retains its pretraining capabilities during fine-tuning and information relevant to the pretraining task can be linearly probed from its intermediate outputs.
>
> -------
>
> > [Experiment 2] I believe the wrapper formalization is inconsistently used throughout the paper. Starting at Definition 2, a wrapper g is composed with C and is defined as a map acting on the output of a capability (g is unquantified here, and I believe the range should be the domain here). This definition is independent of a model and how it is parameterized. However, in the experiments, the wrapper definition is always a subset of the weights to be pruned. I believe this is not captured as a function of the capability/model output, but rather the function parameterization. As such, probing can not be used as evidence that the model learns a wrapper over the capability.
>
> Thank you for this comment! We believe there is a misunderstanding here and would like to clarify the same. *Please note that the notion of a wrapper was not defined in Definition 2*---that definition solely focuses on defining “capability relevance”, a term we operationalize later on in the work to understand the effects of a pretraining capability on fine-tuning. The notion of a “wrapper” is in fact instantiated (though never formally defined) in the paragraph after Definition 2. Quoting from the paper, we write (note that the following is an updated version, though the submitted version used a similar phrasing): “when a weakly relevant pretraining capability is available, we empirically observe that we can *often* identify specific components in the latter half of the model (e.g., neurons or layers) that seem to implement the transform $g$ in Def. 2.” **That is, the notion of a wrapper is entirely an empirical observation, not a formal definition.** Importantly, it is not defined as part of Definition 2.
>
> For completeness, we explain why we use the term at all: As noted above, we *empirically* find that pruning a few neurons from the deeper layers or probing the activations from those layers is sufficient to infer the correct outputs for a pretraining capability. This *indicates* that the pretraining capabilities are still present in the model and argue that fine-tuning does not quite alter the pretraining capabilities of a model, but “wraps” them into a set of outputs that yield low loss on the downstream dataset. We emphasize however that this is an entirely empirical claim, but one that we justify via experiments in almost 100 different setups (spanning 60 pages of results).
>
> -------

---

> ### Author Response · Authors · 2023-11-21
>
> > In the abstract and introduction, revival via reverse fine-tuning is discussed as being sample-efficient. Experiment 3 is intended to demonstrate this phenomenon experimentally. However, there is no discussion of sample efficiency here or later in the paper. Specifically, to show that the pretrained capability is actually forgotten, there would need to be a comparison to how much time it takes to learn the pretrained capability, which doesn't exist since the capabilities are directly compiled via Tracr. Even this experiment would be confounded by the fact that the fine-tuned model can be seen as simply a good initialization for learning the pretrained capability from scratch. This sample efficiency analysis is critical since in theory, any capability can be learnt by training for long enough.
>
> Thank you for highlighting this! Indeed, we argue earlier in the paper that reverse fine-tuning is sample-efficient. In the experimental results, we try to empirically corroborate this claim by providing the progress of test accuracy as a function of training iterations and demonstrating that the model quickly reaches perfect test accuracy, regardless of whether it was originally fine-tuned with or without a spurious correlation. However, we agree with the reviewer that a claim of sample efficiency requires inclusion of a baseline to compare with the sample complexity of learning the original task. To address this, we have now done the following.
>
> 1. **Added a baseline to PCFG / Tracr results:** We have added a baseline method called Scratch+FT, wherein we initialize the model to parameters pretrained to count $O_{FT}$ and then fine-tune the model to count $O_{PT}$. This baseline is akin to our overall reverse fine-tuning pipeline without a pretraining to count $O_{PT}$ step; importantly, it is designed to capture the note made by the reviewer that an irrelevant capability can still serve as a good initialization. Results are shown in Appendix E.3 and Figure 9 in the main paper. *We see that in noticeably fewer iterations than the baseline, reverse fine-tuning reaches perfect performance.*
>
> 2. **Further validated results on TinyStories:** We have added results at different steps of reverse fine-tuning in the TinyStories setup in Table 1. In this table, the “Not in PT” baseline is similar to the Scratch+FT baseline from point 1 above. We report results of a fine-tuned GPT-3.5 classifier that predicts whether a story generated by the given model contains a Twist, i.e., the feature the model was fine-tuned to *not* generate stories for. We again see that in much fewer steps than the baseline, *all analyzed fine-tuning protocols are able to perform well on the task of generating stories with Twists.* Interestingly, we find sample efficiency is very high in this more realistic setup (**often 30–300 gradient steps are enough to achieve high performance on the pretraining task!**)
>
> -------
>
> > My overall analysis is that the authors haven't provided a clear definition of what it means for a capability to have been lost or recoverable
>
> Thank you for raising this point! We agree that the term “recovered” was never defined formally. We have tried to clarify this updating the paper by adding the following: “If we intervene on the model by either removing its neurons or training it to forget the fine-tuning task for a few iterations, we find the model starts to perform well on the pretraining task again. In such cases, we say the pretraining capability has been 'recovered'.” We have also removed all occurrences of the phrase “capability is not lost” and instead note at these places paraphrases of the following sentence: “performance on the pretraining task continues to be high”.
>
> -------

---

> ### Author Response · Authors · 2023-11-21
>
> > The linked Figure 8 for the probing analysis does not concern probing. Does this mean to refer to Figure 5? If so, what new information does this section provide that is not captured in Experiment 2, which refers to Figure 5? The analysis and takeaway in this section do not seem grounded in data in the current state.
>
> We sincerely apologize for the confusion here. The probing results were supposed to be discussed together with the pruning results in Experiment 2, but were discussed separately in an earlier version of the draft (in Experiment 4), which we forgot to remove during submission. We have uploaded a revised manuscript that addresses this issue now. *We would be extremely grateful if the reviewer can take a look at the results in Section 5.*
>
> -------
>
> > Note on typos and captions / legends in figures.
>
> We sincerely thank the reviewer for patiently describing the typos and relevant missing information in the figures. We have addressed noted concerns and updated the figures, with a thorough proofread of the entire paper.
>
> -------
> -------
>
> **Summary:** We thank the reviewer for their comments that have helped us expand our results in realistic settings and improve the overall presentation of our work. We hope these changes justifiably address the reviewer’s concerns and hope that they can consider increasing their score to support the acceptance of our work.

---

> > ### Comment · Reviewer_wEP2 · 2023-11-23
> >
> > I appreciate the work put in during the rebuttal procedure, these additional clarifications and experiments strengthen the paper, and I have raised my score from 5 to 6.

---

### Official Review · Reviewer_eQAm · 2023-11-05

**Soundness:** 4 excellent
**Presentation:** 3 good
**Contribution:** 3 good
**Rating:** 8
**Confidence:** 4

**Summary:**

This work applies mechanistic interpretability methods on finetuned language models as an attempt to understand how the finetuning process alters pretrained models. The authors conclude that finetuning with smaller learning rate allows the finetuned model to learn a "wrapper" on top of the existing pretrained model to perform specific finetuned tasks.

**Strengths:**

I appreciate how the experiment is setup: gradully stepping from a controllable but synthetic setting to a less controllable but realistic setting. The gradual stepping from controllable to uncontrollable settings and the meticulous linking of claim consistency between each experiment makes tracing the arguments and their evidence easy.  The conclusions drawn are well supported by the empirical results.

I also appreciate how the notion of capability is defined in this work. An exact definition conveys what the authors consider as "capabilities" and their coarse dichotomy into strong and weak relevances.

**Weaknesses:**

The main (potential) weakness of this work is the conclusion drawn. I personally dislike the usage of "lack of novelty" as a justification to reject a paper. That being said, there is frankly not much to be learned about finetuning from this work. Yes, small finetuning rate would only change the model a little. Yes, the change would be potentially reverted if training on the original pretraining task. Yes, a larger finetuning rate would cause collateral damage, causing existing capabilities to be lost. These are all well-established knowledge about finetuning where plenty of papers have explored. Again, this is not to discredit this entire work. The authors should consider highlighting the newer findings that readers may not already know, perhaps how the distinction between strong and weak relevance would benefit the finetuning of a new capability.

The authors motivate the importance of understanding of the finetuning process by mentioning safety and jailbreaking attacks in the introduction and conclusion. Unfortunately no related experiments are shown.

Last but not least, the authors should include a paragraph in related works discussing on how semi-controllable language models (e.g. transformers training on PCFG and Tracr) have been utilized in model interpretability previously, to help contexualize the usage of mechanistic interpretability techniques (e.g. weight pruning). The authors should also include the baseline of finetuning with a portion of pretrained data mix in, since that would be the most common finetuning technique. Conclusions drawn on such finetuned procedures would be more applicable, compared to purely finetuning on the new domain.

**Questions:**

(see weaknesses)


Overall this is a good solid and interesting paper. I would like to see it being accepted.

---

> ### Author Response · Authors · 2023-11-21
>
> We thank the reviewer for their positive feedback! We are glad they found our setup well designed and appreciated the consistency of our claims and arguments with the extensive experiments conducted in the paper. We address specific comments below.
>
> -------
>
> > The main (potential) weakness of this work is the conclusion drawn. I personally dislike the usage of "lack of novelty" as a justification to reject a paper. That being said, there is frankly not much to be learned about finetuning from this work. .... These are all well-established knowledge about finetuning where plenty of papers have explored. Again, this is not to discredit this entire work. The authors should consider highlighting the newer findings ....
>
>
> Thank you for this comment! However, we *respectfully* disagree with the reviewer that the results presented in our work on the effects and limitations of fine-tuning are well-recognized or well-established in the literature. Arguably, certain ML subcommunities [0] have made *intuitive claims* that are related to the precise conclusions of our work; however, *we are unaware of any specific work that rigorously demonstrates results similar to ours on a broad set of well-constructed tasks and a wide range of hyperparameters* (also see Section 2, Related Work for a discussion of prior literature). This is best exemplified by the fact that fine-tuning protocols continue to be used in applications such as alignment of language models, in essence creating an expectation that mere fine-tuning with some well-designed objective can replace undesirable capabilities of a model with more desirable ones. To our knowledge, our work is the first to provide *concrete, mechanistic* evidence that shows pretraining capabilities continue to persist in the model after fine-tuning. We further summarize our contributions below and note that we have updated the text to emphasize them better.
>
> - **Reverse fine-tuning.** Our proposed methodology of reverse fine-tuning (reFT), to the best of our knowledge, has not been utilized in past work to demonstrate the inability of fine-tuning to alter pretraining capabilities. Incidentally, a few contemporary works [1,2,3,4] were released after the ICLR deadline that also propose a version of our reFT pipeline and show that models fine-tuned via protocols like RLHF can, in a few gradient steps (e.g., 5), be made to produce undesirable outputs. These works echo the claim made in our paper’s abstract: a practitioner via downstream fine-tuning can unintentionally remove safety wrappers. *We emphasize that the release of papers in the last month echoing similar results as ours further indicates that our paper’s contributions have not been established or recognized by the community yet.*
>
> - **Mechanistic demonstration that provides concrete evidence that pretraining capabilities continue to persist in a fine-tuned model.** Performing an extensive mechanistic characterization, we demonstrate that the impression of removal of pretraining capabilities via fine-tuning the model is often inaccurate: the fine-tuned model continues to retain features relevant to the pretraining task, these features can be inferred via mere linear probes and localized via network pruning, and they enable fast re-learning of the pretraining task. We further show that the reason a fine-tuned model seemingly shows loss of pretraining capabilities is because it learns minimal wrappers in later layers that alter the intermediate outputs generated by the pretrained capabilities; removing this wrapper revives the pretraining capability. *To our knowledge, these results and the notion of a wrapper have not been introduced in prior work on understanding fine-tuning.*
>
> - **Modeling capability relevance.** By modeling the notion of capability relevance, we produce a precise distinction between when behavioral evaluations will seemingly indicate that the model has lost its pretraining capability versus not: fine-tuning a model with a weakly relevant pretraining capability will yield the impression that pretraining capabilities have been removed, meanwhile fine-tuning a model with strongly relevant pretraining capabilities will not. *To our knowledge, the notion of capability relevance does not exist in prior work and provides a novel axis for thinking about the effects of fine-tuning.* While Lovering et al.’s [5] notion of pretraining as an inductive bias is related, we emphasize their main focus is to assess when a model can sample-efficiently learn a downstream task. Meanwhile, we characterize how the pretraining capabilities themselves are altered via fine-tuning.
>
> [0] https://www.alignmentforum.org/posts/FZL4ftXvcuKmmobmj/causal-confusion-as-an-argument-against-the-scaling
>
> [1] https://arxiv.org/abs/2310.02949
>
> [2] https://openreview.net/forum?id=tmsqb6WpLz
>
> [3] https://arxiv.org/abs/2310.03693v1
>
> [4] https://openreview.net/forum?id=hkQOYyUChL
>
> [5] https://openreview.net/forum?id=mNtmhaDkAr
>
>
> -------

---

> ### Author Response · Authors · 2023-11-21
>
> > The authors motivate the importance of understanding of the finetuning process by mentioning safety and jailbreaking attacks in the introduction and conclusion. Unfortunately no related experiments are shown.
>
> Thank you for mentioning this! Indeed, our paper is *partly* motivated by the fact that fine-tuning plays a critical role in addressing safety concerns of pretrained models. **We have now added further experiments** to further verify our claims in setups designed with this motivation, as explained below.
>
> 1. **Fine-tuning for safety.** We assess a fine-tuning pipeline aimed to capture the idea that when a model produces an undesirable output after pretraining, we may train it to instead produce outputs corresponding to another capability that is deemed safe or desirable. We call this pipeline randFT and analyze it on two setups–-the PCFG counter task and TinyStories. In both scenarios, **we find our claims continue to hold**: (i) via probing, we find we can infer intermediate outputs from the fine-tuned model that correspond to the pretrained capabilities and that the model was actively trained to produce an incorrect output for, and (ii) via reverse fine-tuning, we see that in a few iterations of training, the model is able to accurately perform the pretraining task again.
>
> - *TinyStories setup:* We use a model pretrained on the task of producing stories containing a feature specified by a prompt (e.g., Twists, Bad-ending, Foreshadowing). The model is then fine-tuned to, given a prompt that requires generating stories with the feature Twists, produce a story with a different predefined feature. That is, given the same inputs as seen during pretraining, the output produced after fine-tuning should be consistently altered "to a more desired one". Results are reported in Appendix F.
> - *PCFG setup:* We use a model pretrained on the task of counting a token $O_{PT}$ and fine-tune it on the downstream task where the task specification still indicates the model should count $O_{PT}$, but the answer corresponds to the count of $O_{FT}$; i.e., given the same input, the output produced after fine-tuning should be consistently altered to a more desired one. We then (i) probe the model to infer the count of $O_{PT}$ or (ii) reverse fine-tune it to relearn the ability to count $O_{PT}$. Results are reported in Appendix E.4.
>
> 2. **Jailbreaking PCFG models.** We develop a jail-breakable setup by defining multiple task tokens in the PCFG counters task. Specifically, instead of just a single task token that prompts the model to count an operand (e.g., the number of $O_{PT}$’s in a string), we now use three such task tokens—called $T_{NJ}$, $T_{J_1}$, $T_{J_2}$—during the model’s pretraining. We randFT (see point 1. above) these models to count a different target operand token ($O_{FT}$), but use only one of the task tokens (specifically, $T_{NJ}$). Results are reported in Appendix E.2. Behaviorally (Figure 24), we see that the model upon prompting to count the token $O_{PT}$ with the task token $T_{NJ}$ instead counts the number of $O_{FT}$. This seemingly indicates fine-tuning has altered the capability to count the pretraining token. However, if the task tokens $T_{J_1}$ or $T_{J_2}$ are used, we find the model outputs the count of $O_{PT}$. This indicates the pretraining capability is still present in the model. Our minimal setup thus captures the essence of jailbreaking, whereby a small prompt change can elicit an undesirable capability. **Performing linear probing of these models (Figure 25), we see the intermediate outputs of the model *with any of the task tokens* produce similar results in the earlier layers.** We argue this indicates that the use of task tokens $T_{J_1}/T_{J_2}$ ensures the model does not use the wrapper on the intermediate outputs to produce the count of $O_{FT}$, but the relevant features corresponding to the pretraining capability are produced in all scenarios!
>
> -------

---

> ### Author Response · Authors · 2023-11-21
>
> > Last but not least, the authors should include a paragraph in related works discussing on how semi-controllable language models (e.g. transformers training on PCFG and Tracr) have been utilized in model interpretability previously, to help contexualize the usage of mechanistic interpretability techniques (e.g. weight pruning).
>
> We agree this discussion is valuable! We note we had included a paragraph highlighting the use of semi-controllable systems in past work to develop a better understanding of language models (see Paragraph 3 in Section 2, Related work). However, to further address your comments, *we have significantly expanded the discussion in this paragraph* by including more papers that were recently released. We report the updated paragraph below.
>
> ```
> Model interpretability via synthetic tasks. Several recent works have focused on mechanistically understanding how Transformers learn synthetic language generation tasks, such as learning formal grammars and board games (Allen-Zhu & Li, 2023c; Zhao et al., 2023; Li et al., 2023; Nanda et al., 2023; Liu et al., 2022a; Valvoda et al., 2022; Liu et al., 2023a; Zhou et al., 2023). The goal of such papers, including ours, is not necessarily to provide accurate explanations for the success of LLMs, but to develop concrete hypotheses that can be used to develop grounded experiments or tools for understanding their behavior. For example, in a recent work, Allen-Zhu & Li (2023a;b) use a synthetically designed setup to develop hypotheses for how ``knowledge'' about an entity is stored in a pretrained model, showing such knowledge can often be manipulated via relatively simple linear transformations. Similarly, Okawa et al. (2023) use a procedurally defined multimodal dataset to hypothesize and demonstrate emergent capabilities seen in neural networks (Wei et al., 2022) may be driven by the compositional nature of real world data. In another work, Zhou et al. (2023) utilize Tracr compiled Transformers to hypothesize and demonstrate that if primitive operations involved in a formal algorithm can be implemented by a model, stepwise inference is sufficient to enable length generalization. Similarly, Feng et al. (2023) use context-free grammars to demonstrate stepwise inference allows Transformers to solve problems that require dynamic programming.
> ```
>
>
>
> -------
>
> > The authors should also include the baseline of finetuning with a portion of pretrained data mix in, since that would be the most common finetuning technique. Conclusions drawn on such finetuned procedures would be more applicable, compared to purely finetuning on the new domain.
>
> Thank you for this suggestion! We agree that mixing in the pretraining data during fine-tuning can help us further demonstrate the validity of our claims in an often used setup in practical scenarios. To this end, we have now performed a behavioral and probing analysis of our PCFG counter task by fine-tuning the pretrained model on a “mixed dataset” that involves samples for both learning to count a target fine-tuning token *and* samples for learning to count tokens seen during pretraining. Results are shown in Appendix E.1. **We find our claims from the zero mixing setup, i.e., when the pretraining data is not seen at all, directly transfer.** Specifically, we assess three degrees of mixing: (i) 50% pretrained data + 50% fine-tuning data (high mixing); (ii) 10% pretraining data + 90% fine-tuning data (medium mixing); and (iii) 0.1% pretraining data + 99.9% fine-tuning data (low mixing). We find that when a strongly relevant capability is present (high sampling prior of $O_{FT}$ in pretraining), the model performs well on the fine-tuning task regardless of a spurious correlation; if only a weakly relevant capability is present, the model performs well only if the spurious correlation is present in a sample; and probing intermediate representations shows the model encodes the count of pretraining token in both scenarios. The primary difference with our prior, zero-mixing results is that, if enough mixing is performed, the fine-tuned model now sees reduced degradation in performance on the pretraining task—*this is expected since the pretraining data is actively presented to the model during fine-tuning now.*
>
> -------
> -------
>
>
> **Summary:** We thank the reviewer for their valuable feedback that has helped us better contextualize our contributions and expand our results to further relevant settings that the community may find interesting. We hope these changes justifiably address the reviewer’s concerns and hope that they will consider increasing their score to support the acceptance of our work.

---

> ### Comment · Reviewer_eQAm · 2023-11-22
>
> I truly appreciated the authors for their dedication in composing the rebuttal. The authors' response covered all the concerns mentioned in the review (some experiments far exceeding expectations). Perhaps one last thing the authors should consider clarifying is the relevance of the classification between strong and weak relevance. I have no other concerns and thus have increased the score from 5 to 8.

---

### Meta-Review · Area_Chair_qarH · 2023-12-12

**Metareview:**

The authors systematically study the impact of fine-tuning on large language models via applying mechanistic interpretable methods. The experimental setting and analysis are thorough and the results highlight many interesting facts on fine-tuning, e.g., finetuning with smaller learning rate allows the finetuned model to learn a "wrapper" on top of the existing pretrained model to perform specific finetuned tasks.

**Justification For Why Not Higher Score:**

Definition of some key concepts like capability and the overall conclusion could have been described more clearly.

**Justification For Why Not Lower Score:**

The paper has sufficient novelty for a publication.

---

### Decision · Program_Chairs · 2024-01-16

Accept (poster)